# Multi-step ahead daily inflow forecasting using ERA-Interim reanalysis dataset based on gradient boosting regression trees

Shengli Liao[1], Zhanwei Liu[1], Benxi Liu[1], Chuntian Cheng[1], Xinfeng Jin[1], Zhipeng Zhao[1]

[1]Institute of Hydropower System and Hydroinformatics, Dalian University of Technology, Dalian 116024, China

*Correspondence to*: Zhanwei Liu (337891617@qq.com)

**Abstract.** Inflow forecasting plays an essential role in reservoir management and operation. The impacts of climate change and human activities make accurate inflow prediction increasingly difficult, especially for longer lead times. In this study, a new hybrid inflow forecast framework with ERA-Interim reanalysis dataset as input, adopting gradient boosting regression trees (GBRT) and the maximal information coefficient (MIC) is developed for multi-step ahead daily inflow forecasting. Firstly, the ERA-Interim reanalysis dataset provides more information for the framework to discover inflow for longer lead times. Secondly, MIC can identify effective feature subset from massive features that significantly affects inflow so that the framework can reduce computational burden, distinguish key attributes with unimportant ones and provide a concise understanding of inflow. Lastly, the GBRT is a prediction model in the form of an ensemble of decision trees and has a strong ability to capture nonlinear relationships between input and output at longer lead times more fully. The Xiaowan hydropower station located in Yunnan Province, China is selected as the study area. Six evaluation criteria, the mean absolute error (MAE), the root mean squared error (RMSE), the Pearson correlation coefficient (CORR), Kling–Gupta efficiency scores (KGE), the percent bias in flow duration curve high-segment volume (BHV) and the Index of Agreement (IA) are used to evaluate the established models using historical daily inflow data (1/1/2017-31/12/2018). Performance of the presented framework is compared to that of artificial neural networks (ANN), support vector regression (SVR) and multiple linear regression (MLR) models. The results indicate that the reanalysis data enhances the accuracy of inflow forecasting for all lead times studied (1-10 days) and the developed method generally performs better than other models, especially for the extreme values and longer lead times (4-10 days).

**Keywords** Inflow forecasting, Gradient boosting, Regression trees, Maximal information coefficient, ERA-Interim

## 1 Introduction

Reliable and accurate inflow forecasting 1-10 days in advance is significant for efficient utilization of water resources, reservoir operation and flood control, especially in areas with concentrated rainfall. Rainfall in southern China is usually concentrated for several days at a time due to strong convective weather, for example, typhoons. Low accuracy inflow predictions can easily cause the failure of power stations to make reasonable power generation plans 7-10 days ahead of disaster events and lead to unnecessary water abandonment and even substantial economic losses. Fig. 1 shows the losses of electric quantity due to discarded water (LEQDW) in Yunnan and Sichuan Provinces, China from 2011 to 2016 (Sohu, 2017;

in-en, 2018). The total amount of LEQDW in Yunnan and Sichuan Provinces increased from 1.5 billion kWh to 45.6 billion kWh from 2011 to 2016, with an average annual growth rate of 98.0%. In recent years, due to the increased number of hydropower stations and installed hydropower capacities, the problem of discarding water caused by inaccurate inflow forecasting is becoming increasingly serious, which has also produced a negative impact on the development of hydropower in China.

The main challenge in inflow forecasting caused by climate change and human activities at present are low accuracy, especially for longer lead times (Badrzadeh et al., 2013; El-Shafie et al., 2007). Meanwhile, due to streamflow variation by reason of climate change and human activities, inflow forecasting model often needs to be rebuilt and the model parameters need to be recalibrated according to the actual inflow and meteorological data within one or two years. To address these problems, a variety of models and approaches have been developed. These approaches can be divided into three categories: statistical methods (Valipour et al., 2013), physical methods (Duan et al., 1992; Wang et al., 2011; Robertson et al., 2013), and machine learning methods (Chau et al., 2005; Liu et al., 2015; Rajaee et al., 2019; Zhang et al., 2018; Yaseen et al., 2019; Fotovatikhah et al., 2018; Mosavi et al., 2018; Chau, 2017). Each method has its own conditions and scope of application. Statistical methods are usually based on historical inflow records and mainly include the autoregressive model, the autoregressive moving average (ARMA) model and the autoregressive integrated moving average (ARIMA) model (Lin et al., 2006). Statistical methods assume that the inflow series is stationary and the relationship between input and output is simple. However, real inflow series is complex, nonlinear and chaotic (Dhanya and Kumar, 2011), making it difficult to obtain high-accuracy predictions using statistical models. Physical methods which have clear mechanisms are implemented using theories of inflow generation and confluence. These methods can reflect the characteristics of catchment but are very strict with initial conditions and input data (Bennett et al., 2016). Meanwhile, these methods used for flood forecasting have a shorter lead time and cannot be used to acquire long-term forecasting results due to input uncertainty. Machine learning methods, having a strong ability to handle the nonlinear relationship between input and output and recently shown excellent performance in inflow prediction, are widely used for medium and long-term inflow forecasts. In particular, several studies had shown that artificial neural networks (ANN) (Rasouli et al., 2012; Cheng et al., 2015; El-Shafie and Noureldin, 2011) and support vector regression (SVR) (Tongal and Booij, 2018; Luo et al., 2019; Moazenzadeh et al., 2018) are the two powerful models for inflow predicting. However, these models still have some inherent disadvantages. For example, ANN is prone to being trapped by local minima and both ANN and SVR suffer from over-fitting problems and reduced generalizing performance. In recent years, gradient boosting regression trees (GBRT) (Fienen et al., 2018; Friedman, 2001), a nonparametric machine learning method based on a boosting strategy and decision trees, was developed and had been used in traffic (Zhan et al., 2019) and environmental (Wei et al., 2019) field and proved to alleviate these problems mentioned above. Thus, GBRT is selected for daily inflow prediction with a lead times of 1-10 days in this paper. Compared with ANN and SVR, GBRT also has two other advantages. Firstly, GBRT can rank features according to their contribution to model

scores, which is of great significance for reducing the complexity of the model. Secondly, GBRT is a white box model and can be easily interpreted. To the best of our knowledge, GBRT has not been used for daily inflow prediction with a lead times of 1-10 days before. For comparison purposes, ANN, SVR and multiple linear regression (MLR) have been employed to forecast daily inflow and are considered as bench mark models in this study.

In addition to forecasting models, a vital reason why many approaches cannot attain higher accuracy for inflow predictions is that inflow is influenced by various factors (Yang et al., 2019), such as rainfall, temperature, humidity, etc. Thus, it is very difficult to select appropriate features for inflow forecasting. Current feature selection methods for inflow forecasting mainly include two methodologies. The first method is the model-free method (Bowden et al., 2005; Snieder et al., 2019) which employs a measure of the correlation coefficient criterion ( He et al., 2011; Badrzadeh et al., 2013; Siqueira et al., 2018; Pal et al., 2013) to characterise the correlation between a potential model input and the output variable. The second method is the model-based method (Snieder et al., 2019) which usually utilizes the model and search strategies to determine optimal input subset. Common search strategies include forward selection, backward elimination et al (May et al., 2011). The correlation coefficient has limited ability for capturing nonlinear relationships and exhaustive search tend to need the higher computation burden. In order to select effective inputs accurately and quickly, the maximal information coefficient (MIC) (Reshef et al., 2011) is used to select input factors for inflow forecasting. MIC is a robust measure of the degree of correlation between two variables and has attracted a lot attention from academia (Zhao et al., 2013; Ge et al., 2016; Lyu et al., 2017; Sun et al., 2018). In addition, sufficient potential input factors are the prerequisite for obtaining reliable and accurate prediction results and it is not enough to use only antecedent inflow series as the input of the model. To enhance the accuracy of inflow forecasting and acquire a longer lead time, increasing amounts of meteorological forecasting data have been used for inflow forecasting (Lima et al., 2017; Fan et al., 2015; Rasouli et al., 2012). However, with extended lead times, the errors of forecast data continuously increase because the variables obtained by numerical weather prediction (NWP) system are also affected by complex factors (Mehr et al., 2019). Moreover, with the continuous improvement of forecasting systems, it is difficult to obtain consistent and long series of forecasting data (Verkade et al., 2013). To mitigate these problems, the reanalysis data generated by ERA-Interim (European Centre for Medium-Range Weather Forecasts (ECMWF) Re-Analysis Interim) (Dee et al., 2011), which was proved to be one of the best methods for reanalysis of data describing atmospheric circulation and elements (Kishore et al., 2011), has been used as an input. The reanalysis data which has less error than observed data and forecast data is the result of assimilating observed data with forecast data. ERA-Interim shows the results of a global climate reanalysis from 1979 to date, which are produced by a fixed version of NWP system. The fixed version ensures that there are no spurious trends caused by an evolving NWP system. Therefore, meteorological reanalysis data satisfies the need for long sequences of consistent data and has been used for the prediction of wind speeds (Stopa and Cheung, 2014) and solar radiation (Ghimire et al., 2019; Linares-Rodríguez et al., 2011).

This study aims to provide a reliable inflow forecasting framework with longer lead times for daily inflow forecasting. The framework adopts the ERA-Interim reanalysis dataset as the input which ensured ample information is supplied to depict inflow. MIC is used to select appropriate features to avoid over-fitting and waste of computing resources caused by feature redundancy. GBRT, which is robust to outliers and has strong non-linear fitting ability, is used as the prediction model to improve inflow forecasting accuracy of longer lead times.

This paper is organized as follows: Section 2 describes a case study and collected data. Section 3 introduces the theory and process of methods used, including MIC and GBRT. Section 4 shows the results and discussion of the data, followed by the conclusions in Section 5.

## 2 Data

### 2.1 Study area and collected data

The Xiaowan Hydropower Station in the lower reaches of the Lancang River is chosen as the study site (Fig. 2). The Xiaowan Hydropower Station is the main controlling hydropower station in the Lancang River and it is very meaningful to adopt the Xiaowan Hydropower Station as the case study. The Lancang River is approximately 2000 km long and has a drainage area of 113300 km$^2$ above the Xiaowan Hydropower Station. The Lancang River which is also known as the Mekong River originates in the Tibetan Plateau and runs through China, Myanmar, Laos, Thailand, Cambodia, and Vietnam. The major source of water flowing into the Lancang River in China comes from melting snow on the Tibetan plateau ( Commission Mekong River, 2005).

We collected ERA-Interim reanalysis dataset, observed daily inflow and rainfall data for Xiaowan for 8 years (January 2011 to December 2018). Fig. 3 depicts the daily inflow series. The data from January 2011 to December 2014 (1461 days, approximately 50% of the whole dataset), from January 2015 to December 2016 (731 days, approximately 25% of the whole dataset) and from January 2017 to December 2018 (730 days, approximately 25% of the whole dataset) are used as training, validation and testing set, respectively. The reanalysis dataset can be downloaded from *https://apps.ecmwf.int/datasets/data/ interim-full-daily/levtype=sfc/* and is provided every 12 hours on a spatial grid size of 0.25° × 0.25°. Based on the expert knowledge and on the basis of available literature, the near-surface 26 variables (Table A1) from the reanalysis data are considered as potential selected predictors for inflow forecasting, which include the total precipitation (tp), the 2 meter temperature (t2m), the total column water (tcw), etc. More details about ERA-Interim dataset are presented in the Appendix A.

### 2.2 Feature scaling and feature selection

Feature scaling is necessary for machine learning methods and all features are scaled to the range between 0 and 1 before taking part in the calculation, as follows:

$$x_{scale} = \frac{x_{original} - x_{min}}{x_{max} - x_{min}} \tag{1}$$

where $x_{scale}$ and $x_{original}$ indicate the scaled and original data, respectively. $x_{max}$ and $x_{min}$ represent the maximum and

minimum of inflow series, respectively.

Reasonable selection of input variables can reduce the computational burden and improve the prediction accuracy of the

model by removing redundant feature information and reducing the dimensions of the features. If too many features are

selected, model will become very complex, which will cause trouble when adjusting parameters, resulting in over-fitting and

difficult convergence. Moreover, natural patterns in the data will be blurred by noise (Zhao et al., 2013). On the other hand,

if irrelevant features are chosen, there will add noise into the model and also hinder the learning process. MIC is employed to

select inputs from candidate predictors from reanalysis data. The lagged inflow and rainfall series are identified by partial

autocorrelation function (PACF) and cross-correlation function (CCF). The corresponding 95% confidence interval is used

to identify significant correlations. Furthermore, when correlation coefficient slowly declines and cannot fall into confidence

interval, a trial and error procedure is used to determine the optimum lag, i.e., starting from one-lag and then modifying the

external inputs by successively adding one more lagged time series into inputs (Amiri 2015; Shoaib et al., 2015).

## 3 Methodology

### 3.1 Feature selection via maximal information coefficient

The calculation of MIC is based on concepts of the mutual information (MI) (Kinney and Atwal, 2014). For a random

variable X, such as observed inflow, the entropy of X is defined as

$$H(X) = -\sum_{x \in X} p(x) \log p(x) \tag{2}$$

where $p(x)$ is the probability density function of X = x. Furthermore, for another random variable Y, such as observed

rainfall, the conditional entropy of X given Y may be evaluated from the following expression

$$H(X|Y) = -\sum_{x \in X} \sum_{y \in Y} p(x,y) \log p(x|y) \tag{3}$$

where $H(X|Y)$ is the uncertainty of X given knowledge, $p(x,y)$ and $p(x|y)$ are the joint probability density and the

conditional probability of X = x and Y = y, respectively. The reduction of the original uncertainty of X, due to the knowledge

of Y, is called the MI (Amorocho and Espildora, 1973; Chapman, 1986), defined by

$$MI(X,Y) = H(X) - H(X|Y) = \sum_{x \in X} \sum_{y \in Y} p(x,y) \log \frac{p(x,y)}{p(x)p(y)} \tag{4}$$

The calculation of MIC is divided into three steps. Consider given a dataset D, including variable X and Y with a sample size

$n$. Firstly, drawing scatter plots of X and Y and drawing grids for partitioning which is called an $x$-by-$y$ grid. Let D|G denote

the distribution of D divided by one of x-by-y grids as G. $MI^{*}(D, x, y) = \max MI(D \mid G)$, where $MI(D \mid G)$ is the mutual information of D|G. Secondly, characteristic matrix is defined as

$$M(D)_{x,y} = \frac{MI^{*}(D, x, y)}{\log(\min(x, y))} \tag{5}$$

Lastly, MIC is introduced as the maximum value of characteristic matrix, that is, $MIC(D) = \max_{xy < B(n)} M(D)_{x,y}$, where B(n) is the upper bound of the grid size which is a function of sample size, defined $B = n^{0.6}$.

We perform feature selection from ERA-Interim reanalysis dataset in two steps via MIC. First, compute MIC of each reanalysis variables and observed inflow. Then, sort features based on MIC in a descending order and determine the optimum inputs by using a trail-and-error procedure, i.e. starting from the top one feature and then modifying the external inputs by successively adding one more feature into model inputs. The selected $k$ features from reanalysis data are used as part of input to the model.

**3.2 Gradient boosting regression trees**

GBRT is an ensemble model which mainly includes two algorithms: decision tree algorithm and the boosting algorithm. The decision tree robust to outliers is used as a primitive model and boosting algorithm as integration rule is used to improve inflow forecasting accuracy.

**3.3.1 The decision tree**

The decision tree in this paper refers to decision tree learning used in computer science, which is one of the predictive modelling approaches used in machine learning. A decision tree consists of branch nodes (the tree structure) and leaf nodes (the tree output).

Supposing a training dataset is given in a feature space with $N$ features and each feature with $n$ samples, $\{(X_1, y_1), (X_2, y_2), \ldots, (X_n, y_n)\}$ $(X_i = (x_1, x_2, \ldots, x_N), i = 1, 2, \ldots, n)$. In the input space where the training set is located, each region is recursively divided into two subregions and the output value of each subregion is used to construct a binary decision tree. The top-down cyclic branch learning of the decision tree adopts a greedy algorithm where each branch node only cares about its own objective function. By traversing all features and all segmentation points of each feature, the best feature $j$ and segmentation points $s$ can be found by minimizing squared loss:

$$\min_{j,s} \left[ \min_{c_1} \sum_{X_i \in R_1(j,s)} (y_i - c_1)^2 + \min_{c_2} \sum_{X_i \in R_2(j,s)} (y_i - c_2)^2 \right] \tag{6}$$

where

$$\begin{aligned} R_1(j,s) &= \left\{ X_i \mid x_i^{(j)} \leqslant s, i = 1, 2, \cdots, N \right\} \\ R_2(j,s) &= \left\{ X_i \mid x_i^{(j)} > s, i = 1, 2, \cdots, N \right\} \end{aligned} \tag{7}$$

$$c_m = \frac{1}{N_m} \sum_{X_i \in R_m(j,s)} y_i \quad (m = 1, 2) \tag{8}$$

$y_i$ is the observed value and $R_1(j, s)$ and $R_2(j, s)$ are the results of partitioning. $c_1$ and $c_2$ are output values of $R_1(j, s)$ and $R_2(j, s)$, respectively. Fig. 4 shows an example of a decision tree model with a max depth and number of leaf nodes of 3 and 5, respectively. If the threshold of loss is set as the stopping condition of the decision tree, it will easily lead to over-fitting problems. Hence, we set the following parameters to alleviate the over-fitting problem of the decision tree model: the maximum depth of the tree, the minimum number of samples required to split an internal node, the minimum number of samples required to be at a leaf node and the number of leaf nodes. These parameters are also the ones used for optimization when using the decision tree.

### 3.3.2 The boosting algorithm

The idea of gradient boosting originated in the observation by Breiman (Breiman, 1997) and can be interpreted as an optimization algorithm based on a suitable cost function. Explicit regression gradient boosting algorithms are subsequently developed (Friedman, 2001; Mason et al., 2000). The boosting algorithm used is described here. Supposing a training dataset with $n$ sample $\{(X_1,y_1),(X_2,y_2),\cdots,(X_n,y_n)\}$, a squared loss function is used to train the decision tree:

$$L(y, f(X)) = \sum_{i=1}^{n} (y - f(X_i))^2 \tag{9}$$

The core of the GBRT algorithm is the iterative process of training the decision with a residual method. The iterative training process of GBRT with $M$ decision trees is as follows:

1) Initialization $f_0(x) = \arg\min_c \sum_{i=1}^{n} L(y_i, c)$ .

2) For $m$-th (m=1, 2, ..., $M$) decision trees:

a) Operating $i$-th (i=1, 2, ..., $n$) sample points. Using the negative gradient of the loss function to replace the residual in the current model $r_{mi} = -\left[ \dfrac{\partial L(y_i, f(x_i))}{\partial f(x_i)} \right]_{f(x)=f_{m-1}(x)}$ .

b) Fitting a regression tree with $\{(x_i, r_{mi})\}$. The $i$-th regression tree with $R_{mt}$ (t = 1, 2, ..., T) as its corresponding leaf node region is obtained, where $t$ is the number of leaf nodes of regression.

c) For each leaf region $t = 1, 2, ..., T$, and the best fitting value is calculated by $c_{mt} = \arg\min_c \sum_{x_i \in R_{mt}} L(y_i, f_{m-1}(x_i) + c)$ .

d) The fitting results are updated by adding the obtained fitting values to the previous ones using

$$f_{mt}(x_i) = f_{m-1}(x_i) + \sum_{t=1}^{T} c_{mt} I \quad (x_i \in R_{mt}) .$$

3) Finally, a strong learning method is obtained $\hat{f}(x_i) = f_M(x_i) = \sum_{m=1}^{M} \sum_{t=1}^{T} c_{mt} I \quad (x_i \in R_{mt})$ .

According to the above introduction to GBRT, the parameters of the GBRT can be divided into two categories: boosting parameters and tree parameters. The boosting parameters include the learning rate and the number of weak learners (*learning_rate* and *n_estimators*). The learning rate setting is used for reducing the gradient step. The learning rate influences the overall time of training, and the smaller the value is, the more iterations are required for training. There are four tree parameters: *max_leaf_nodes*, *min_samples_leaf*, *min_samples_split* and *max_depth*. Hence, GBRT has six parameters control model complexity (Fienen et al., 2018), which we adjusted for tuning by using a trial-and-error procedure.

### 3.3 Evaluation criteria of the models

It is critical to carefully define the meaning of performance and to evaluate the performance on the basis of the forecasting and fitted values of the model compared with historical data. The root mean squared error (RMSE) and mean absolute error (MAE) are the most commonly used criteria to assess model performance and are calculated using Eq. (10) and Eq. (11), respectively.

$$RMSE = \sqrt{\frac{1}{n}\sum_{i=1}^{n}(\hat{Q}_i - Q_i)^2} \tag{10}$$

$$MAE = \frac{1}{n}\sum_{i=1}^{n}\left|\hat{Q}_i - Q_i\right| \tag{11}$$

where $\hat{Q}_i$ and $Q_i$ are the inflow estimation and observed value at time $i$, respectively and $n$ is the number of samples. The RMSE is more sensitive to extremes in sample sets and thus it is used to evaluate the model's ability to simulate flood peaks. The Pearson correlation coefficient (CORR) is a measure of the strength of the association between observed inflow series and forecasted inflow series; it is calculated according to Eq. (12).

$$CORR = \frac{\sum_{i=1}^{n}(Q_i - \bar{Q})(\hat{Q}_i - \bar{\hat{Q}})}{\sqrt{\sum_{i=1}^{n}(Q_i - \bar{Q})^2}\sqrt{\sum_{i=1}^{n}(\hat{Q}_i - \bar{\hat{Q}})^2}} \tag{12}$$

where $\bar{\hat{Q}}$ is the mean of the estimation series. The range of the CORR is between 0 and 1 and values close to 1 demonstrate a perfect estimation result.

Kling–Gupta efficiency scores (KGE) (Knoben et al., 2019) is also a widely used evaluation index. It can be provided as following Eq. (13) and (14).

$$KGE = 1 - \sqrt{(CORR - 1)^2 + (\frac{\hat{\sigma}}{\sigma} - 1)^2 + (\frac{\bar{\hat{Q}}}{\bar{Q}} - 1)^2} \tag{13}$$

$$\hat{\sigma} = \sqrt{\frac{1}{n}\sum_{i=1}^{n}(\hat{Q}_i - \bar{\hat{Q}})^2}, \ \sigma = \sqrt{\frac{1}{n}\sum_{i=1}^{n}(Q_i - \bar{Q})^2} \tag{14}$$

where $\sigma$ is the standard deviation of the observed values, $\hat{\sigma}$ is the standard deviation of the inflow estimation, $\mu$ is the mean of the observed series and $\hat{\mu}$ is the mean of the inflow estimation series.

The percent bias in flow duration curve high-segment volume (BHV) (Yilmaz et al., 2008; Vogel and Fennessey, 1994) is presented to estimate prediction performance of extreme value for model. It can be provided as following Eq. (15).

$$BHV = \frac{\sum_{h=1}^{H}(\hat{Q}_h - Q_h)}{\sum_{h=1}^{H}Q_h} \times 100 \tag{15}$$

where h = 1, 2, . . ., H are the inflow indices for inflows with exceedance probabilities lower than 0.02. In this paper, the inflow threshold of exceedance probabilities equalling 0.02 is 1722 m³/s.

The Index of Agreement (IA) (Willmott, 1981) plays a significant role in evaluating the degree of the agreement between observed series and inflow estimation series. Similar to CORR, its range is between 0 (no agreement at all) and 1 (perfect fit). It is given by:

$$IA = 1 - \frac{\sum_{i=1}^{n}(\hat{Q}_i - Q_i)^2}{\sum_{i=1}^{n}(|\hat{Q}_i - \overline{Q}| + |Q_i - \overline{Q}|)^2} \tag{16}$$

### 3.4 Overview of framework

Fig.5 illustrates the overall structure of framework presented. This structure consists of two major models: GBRT and GBRT-MIC.

In GBRT, we measure the relevance of different lags observed inflow and rainfall with observed inflow at the time of forecast via partial autocorrelation function (PACF) and cross-correlation function (CCF) (Badrzadeh et al., 2013) and select appropriate lags as predictors of model by hypothesis test and trial-and-error procedures. Then, data pre-processing and feature scaling are carried out for selected predictors. Next, dividing the dataset into training set, validation set, and testing set according to the length of each data set specified in advance (in Section 2.2). A grid search algorithm, which is an exhaustive search all candidate parameter combination method, is guided to optimization model parameters by evaluation of validation set for each lead time (Chicco and Davide, 2017). Lastly, prediction results are evaluated based on testing set. Compared with GBRT, GBRT-MIC adds reanalysis data which are selected via MIC (in Section 3.1) as the input of the model. Moreover, GBRT-MIC also calculates the importance of features according to the prediction results and ranks the features (Louppe, 2014).

It is difficult to perform multi-step forecasting by the reason of accumulation of errors, reduced accuracy, and increased uncertainty. The current state of multi-step ahead forecasting is reviewed, there are mainly two strategies that you can use for multi-step forecasting for single-output, namely, Static (Direct) multi-step forecast and Recursive multi-step forecast

(Bontempi et al., 2013; Taieb et al., 2012). Recursive forecast strategy is biased when the underlying model is nonlinear and is sensitive to the estimation error, since estimated values, instead of actual ones, are more and more used when we get further in the future (Bontempi et al., 2012). Thus, the Static multi-step forecasting strategy is employed in this paper. Since the Static strategy does not use any approximated values to compute the forecasts, it is not prone to any accumulation of errors. The model structures of GBRT and GBRT-MIC are as follows:

$$\hat{Q}_{t+T}^{I} = f(\vec{\theta}_t^{I}; Q_t, Q_{t-1}, ..., Q_{t+1-p}, R_t, R_{t-1}, ..., R_{t+1-q}) \quad (T = 1, 2, ..., 10) \tag{17}$$

$$\hat{Q}_{t+T}^{II} = f(\vec{\theta}_t^{II}; Q_t, Q_{t-1}, ..., Q_{t+1-p}, R_t, R_{t-1}, ..., R_{t+1-q}, E_t^1, E_t^2, ..., E_t^k) \quad (T = 1, 2, ..., 10) \tag{18}$$

where $\hat{Q}_{t+T}^{I}$ and $\hat{Q}_{t+T}^{II}$ are the forecasted value of GBRT and GBRT-MIC at the lead time $T$ of current time $t$, respectively. $\vec{\theta}_t^{I}$ and $\vec{\theta}_t^{II}$ are parameters of GBRT and GBRT-MIC at the lead time $T$ of current time $t$, respectively. $p$ and $q$ are lags of observed inflow and rainfall determined via PACF and CCF, respectively. $E_t$ is the features from reanalysis data at the current time $t$ and $k$ is the number of features from reanalysis data determined via MIC.

## 4. Experimental results and discussion

In order to compare with GBRT-MIC, the ANN-MIC, SVR-MIC and MLR-MIC, obtained by replacing GBRT in the framework with ANN, SVR and MLR, respectively, are also employed for inflow forecasting with lead times of 1-10 days. As mentioned previously, six indices, i.e. the MAE, RMSE, CORR, KGE, BHV and IA, are calculated to evaluate the performance of models based on the testing set. We also explored the feature importance based on the GBRT-MIC model (Louppe, 2014). All computations of this paper are performed on a ThinkPad P1 workstation containing an Intel Core i7-9850H CPU with 2.60 GHz and 16.0 GB of RAM, using the version 3.7.10 of Python (Python Software Foundation, 2020), which is powerful, fast and open, and scikit-learn package (Pedregosa et al., 2011).

### 4.1 Feature selection

Fig. 6 shows the PACF, CCF and the corresponding 95% confidence interval from lag 1 to lag 12. The PACF shows significant autocorrelation at lag one and lag four, respectively (Fig. 6(a)), and thus, inflow series one and four-day lag are selected as the inputs of model. CCF between inflow and rainfall gradually decreases as increasing the time lag (Fig. 6(b)) and cannot fall into 95% confidence interval. Therefore, a trial-and-error procedure is used to determine optimal selection of lagged rainfall series. 13 input structures are tried (Table 1) and the trial results are shown in Fig. A1. The results indicate that 7th input structure obtains best performance. Accordingly, rainfall series from one to six-day lag are selected as the inputs of model. As mentioned previously, based to MIC between inflow and the reanalysis variable (Table A1), a trial-and-error procedure is used to determine optimal input subset. 26 input structures are tried (Table 2) and the trial results are

shown in Fig. A2. The results show that 8th input structure obtains best performance and thus the No.1 to 8 predictors in Table A1 are selected as the model input.

Finally, a total of 16 variables including 8 observed variables and 8 reanalysis variables are selected as the model inputs (Table 3). As shown in Table 3, No. 9 to 18 are reanalysis variables and the range of MIC of the reanalysis variables selected is 0.643 to 0.847. Furthermore, No. 9 and No. 13 to 16 are variables related to temperature. Soil temperature level 3 (No. 9) is the temperature of the soil in layer 3 (28-100 cm, the surface is at 0 cm). The temperature of the snow layer (No. 13) gives the temperature of the snow layer from the ground to the snow-air interface. No. 10 to 12 are variables related to the water content of the atmosphere. 2 meter dewpoint temperature (No. 10) is a measure of the humidity of the air. Combined with temperature and pressure, it can be used to calculate the relative humidity. The total column water vapor (No. 11) is only the total amount of water vapor, which is a fraction of the total column water. Total column water (No. 12) is the sum of water vapor, liquid water, cloud ice, rain and snow in a column extending from the surface of the Earth to the top of the atmosphere. Volumetric soil water layer 1 (No. 19) is the volume of water in soil layer 1. In summary, all the selected predictors are interpretable and have a good physical connection with inflow.

## 4.2 Hyperparameter optimization

For machine learning methods, hyperparameters are a king of parameters that are set before training and cannot be directly learned from the regular training process. In order to improve the performance of models, it is imperative to tune the hyperparameters of models. Grid search is employed to tune the hyperparameters of GBRT, GBRT-MIC, ANN-MIC and SVR-MIC.

Reviewing to the basis of available literature (Badrzadeh et al., 2013; Rasouli et al., 2012), an optimizer in the family of quasi-Newton methods, namely *L-BFGS* is used as the training algorithm of ANN and the number of hidden layers is fixed to 3. Another two parameters, namely activation function and the number of nodes of the hidden layer need to be adjusted. A range of 2-20 neurons and four commonly used activation functions (Table 4) are selected by grid search. To alleviate the influence of random initialization of weights, 50 ANN-MIC models are trained for each parameter combination. Optimal activation function and the number of nodes of the hidden layer are determined by selecting the minimal MAE of the validation set for each lead time. The results of the trials show *tanh* and *logistic* function are two more robust activation function (Fig. 7) and ANN with fewer nodes is inclined to obtain lower error. The optimal parameter combination for each lead time is listed in Table 5. It can be seen that the optimal number of nodes is 2, 3 or 4 and the optimal activation function is either *tanh* or *logistic* function.

For SVR, according to Lin et al. (2006) and Dibike et al. (2001), radial basis function (RBF) outperforms other kernel functions for runoff modelling and thus RBF is used as the kernel function in this study. There are three parameters need to be adjusted. Firstly, an appropriate tuning range of parameter is determined by a trial-and-error procedure. And then, to reach

at an optimal choice of these parameters, the MAE is used to optimize the parameters by grid search. Optimal tuning parameters of SVR are shown in Table 5.

As mentioned earlier, for GBRT, there are six parameters need to be adjusted. In order to obtain an optimal parameter combination as soon as possible, we optimize all parameters in two steps. Firstly, *n_estimators* and *learning_rate* are fixed to 100 and 0.1, respectively. The *max_leaf_nodes*, *min_samples_leaf*, *max_depth* and *min_samples_split*, four tuning parameters generate 40000 models at each lead time. Secondly, after the tree parameters are determined, *learning_rate* is modified to 0.01 and *n_estimators* is determined by grid search. To accommodate the computational burden, all models are distributed among about 12 central processing units (CPUs) and total wall time for the runs is about 7 hours for GBRT_MIC and GBRT. Table 6 lists optimal tuning parameters of GBRT and GBRT-MIC.

**4.3 Inputs comparison**

Fig. 8 illustrates performance indices of GBRT and GBRT-MIC on the testing set (2017/01/01-2018/12/31) at lead times of 1-10 days. It is obvious that the reanalysis data selected by MIC makes a great improvement on the GBRT forecasting at both short and long lead times. In particular, for the longer lead times prediction of GBRT-MIC is significantly outperform GBRT. For Fig. 8(a), the MAE of GBRT-MIC decreases from 175 to 172, a decrease of 1.74% for two-day ahead forecasting and decreases from 273 to 237, a decrease increasing to 13.18% for ten-day ahead forecasting compared with GBRT. For Fig. 8(b), the RMSE of GBRT-MIC achieves 1.4% and 10.6% reduction for two and ten-day ahead forecasting, respectively, compared with GBRT. For Fig. 8(c), 8(d) and 8(f), the CORR, KGE and IA of GBRT-MIC increase by 0.2%, 2.2%, 1.0% for two-day ahead forecasting and 3.4%, 7.8% and 2.2% for ten-day ahead forecasting, respectively. Fig. 8(e) compares the BHV of GBRT and GBRT-MIC which indicates reanalysis data can enhance forecasting of extreme values.

Fig. 9(a) shows the five-day ahead forecasted inflow of GBRT-MIC and GBRT versus the observed inflow in the testing set. The slopes of fitting curve of GBRT-MIC and GBRT are 0.89 and 0.81, respectively, which also demonstrates that GBRT-MIC can obtain more accurate inflow forecasting than GBRT. Fig. 9(b) illustrates the distribution of the forecast errors of GBRT and GBRT-MIC. The results show the prediction error of two models approximate to normal distribution. It demonstrates that the prediction error contains less information that is not extracted by the model and more errors of forecasted inflow concentrate at 0 around by GBRT-MIC than GBRT. Fig. 9(c) provides forecasted inflow time series (from the testing set) of GBRT-MIC and GBRT at lead time of five-day. It can be seen that GBRT-MIC provides great performance compare to GBRT, especially for the extreme values. This reveals that the problem of inaccurate extreme value prediction arisen in areas with concentrated rainfall for the GBRT model could be mitigated by incorporating the reanalysis data identified by MIC.

**4.4 Model comparison**

GBRT-MIC, SVR-MIC, ANN-MIC with obtained optimal model parameters are employed for inflow forecasting of one to ten-day ahead. Summarized results for training and testing set are presented in Table 7 and Table 8, respectively. To avoid local minima problems, 50 ANN-MIC models are trained for each lead time and the median of the predictions of the 50 models gives the final prediction. It is clear from Table 7 that the GBRT-MIC are more efficient in the training set than other models at lead times of 1-10 days, which demonstrates that GBRT-MIC has a powerful fitting ability. Meanwhile, all machine learning models obtain better forecasted results than MLR-MIC which cannot capture nonlinear relationship. It should be noted that ANN-MIC has best performance for extreme values in terms of BHV in the training set.

As shown in Table 8, GBRT-MIC performs best for the testing set at lead times of 4-10 days in terms of six indices. At a lead time of ten days, the KGE of GBRT-MIC even reached 0.8317. At the lead times of 1-3 days, three machine learning models obtain approximate performance but outperform MLP-MIC. The machine learning models can acquire enough information to perform forecasting at the short lead time (1-3 days).

The performance indices of these four models in the testing set (2017-2018) at the lead times of 1-10 days are presented in Fig. 10. The results indicate the performance of these four models decreases (higher MAE, RMSE and BHV, and lower CORR, KGE and IA) as the lead time increases. As mentioned earlier, the four models perform equally well for one- to three-day ahead forecasting, whereas significant differences among their performances are found as the lead time exceeds three days. It clearly indicates that the GBRT produce much higher CORR, KGE and IA, and lower MAE, RMSE and BHV than the other three models for four to ten-day ahead forecasting except that the ANN-MIC perform nearly to GBRT-MIC for ten-day ahead forecasting. It should be noted that SVR performs worst according to BHV and KGE, which demonstrates that SVR cannot capture extreme values. On the contrary, GBRT-MIC significantly outperform other models in terms of BHV at lead times of 1-10 days, which indicates that GBRT-MIC is able to obtain extreme values among all models developed in this paper.

**4.5 Feature importance**

A benefit of using gradient boosting is that after the boosted trees are constructed, relative importance scores for each feature can be acquired to estimate the contribution of each feature to inflow forecasting. Fig. 11 shows the feature importance based on GBRT-MIC for lead times of one and ten days. The one-day lag observed time series ( $Q_{t-1}$ ) is more important for shorter lead times (Fig. 11(a)), which demonstrates that the historical observed values are essential to inflow forecasting at shorter lead times.

The features (e.g., $stl3_{t-10}$ and $d2m_{t-10}$) from the reanalysis data have a high relative importance at longer lead times (Fig. 11(b)). Based on the analysis of the concepts of $stl3_{t-10}$ and $tcw_{t-10}$ (Section 4.1), we infer that the temperature near the ground effects the inflow by affecting the melting of snow which is consistent with the fact that the Lancang River is a snow-

melt river. The ten-day lag observed time series ( $Q_{t-10}$ ) is also very important which indicate the long memory of inflow series (Salas 1993). Meanwhile, it is found that the reanalysis data provides important information for inflow forecasting at longer lead times.

**5. Conclusion**

In this study, GBRT-MIC is employed to make inflow forecasts for lead times of 1-10 days and ANN-MIC, SVR-MIC and MLR-MIC are developed to compare with GBRT-MIC. The reanalysis data selected by MIC, the antecedent inflow and the rainfall records selected by PACF and CCF are used as predictors to drive the models. These models are compared using six evaluation criteria, the MAE, RMSE, CORR, KGE, BHV and IA. It is shown that GBRT-MIC, ANN-MIC and SVR-MIC outperform MLR-MIC at lead times of 1-10 days, and GBRT-MIC performs best at lead times of 4-10 days, especially for

forecasting of extreme values.

According to comparison the forecasted results of GBRT and GBRT-MIC, we conclude that GBRT-MIC can be used for more accurate and reliable inflow forecasting at lead times of 1-10 days and reanalysis data selected by MIC makes a great improvement on the GBRT forecasting, especially for lead times of 4-10 days. In addition, the feature importance achieved by GBRT-MIC demonstrates that soil temperature, the total amount of water vapour in a column and dewpoint temperature

near the ground contribute to increase the prediction accuracy of inflow at longer lead times.

In summary, the developed framework that integrates GBRT and reanalysis data selected MIC and can well perform inflow forecasting at lead times of 1-10 days. The results of this study are of significance to assist power stations in making power generation plans 7-10 days in advance in order to reduce LEQDW and flood disasters.

Another direction of improving the results could be considering heuristic methods (for example, Grey Wolf algorithm) to

optimize model parameters, which could search for more wide range of hyper parameters and get optimization parameters more quickly.

**Acknowledgements**

This research is supported by National Natural Science Foundation of China (No. 51979023, No. U1765103) and the Liaoning province Natural Science Foundation of China (No. 20180550354). We are grateful for reanalysis data provided by

European Centre for Medium-Range Weather Forecasts.

**Code/Data availability**

Request for materials should be addressed to Zhanwei Liu.

**Appendix A**

The ERA-Interim is a reanalysis product of the global atmospheric forecasts at ECMWF which is produced through data assimilation system, called as the Integrated Forecast System (IFS). The system includes a 4-dimensional variational analysis (4D-Var) with a 12-hour analysis window. The spatial resolution of the data set is approximately 80 km (0.72°) on 60 levels in the vertical from the surface up to 0.1 hPa. (Berrisford et al., 2011). This reanalysis meteorological products of from 0.125° to 2.5° are generated by interpolation. This reanalysis meteorological products from the ERA-Interim such as rainfall, maximum and minimum temperatures, and wind speed at 0.25° (latitude) × 0.25° (longitude) spatial and 12-hour temporal

resolutions for the study period 2011-2018 are downloaded from the ECMWF webpage.

**Table A1.** Description and notations of the ECMWF Reanalysis Fields.

| Number | Variable | MIC | Description | Units |
|--------|----------|-------|-------------|-------|
| 1 | stl3 | 0.847 | Soil temperature level 3 | K |
| 2 | d2m | 0.781 | 2 metre dewpoint temperature | K |
| 3 | tcwv | 0.699 | Total column water vapour | kg m$^{-2}$ |
| 4 | tcw | 0.699 | Total column water | kg m$^{-2}$ |
| 5 | stl2 | 0.689 | Soil temperature level 2 | K |
| 6 | mn2t | 0.684 | Minimum temperature at 2 metres since previous post-processing | K |
| 7 | tsn | 0.664 | Temperature of snow layer | K |
| 8 | stl4 | 0.643 | Soil temperature level 4 | K |
| 9 | stl1 | 0.631 | Soil temperature level 1 | K |
| 10 | ro | 0.619 | Runoff | m |
| 11 | swvl1 | 0.614 | Volumetric soil water layer 1 | m$^3$ m$^{-3}$ |
| 12 | swvl2 | 0.610 | Volumetric soil water layer 2 | m$^3$ m$^{-3}$ |
| 13 | swvl3 | 0.610 | Volumetric soil water layer 3 | m$^3$ m$^{-3}$ |
| 14 | t2m | 0.571 | 2 metre temperature | K |
| 15 | swvl4 | 0.550 | Volumetric soil water layer 4 | m$^3$ m$^{-3}$ |
| 16 | mx2t | 0.539 | Maximum temperature at 2 metres since previous post-processing | K |
| 17 | sf | 0.470 | Snowfall | m of water equivalent |
| 18 | cp | 0.426 | Convective precipitation | m |
| 19 | tp | 0.416 | Total precipitation | m |
| 20 | rsn | 0.408 | Snow density | kg m$^{-3}$ |
| 21 | lsp | 0.358 | Large-scale precipitation | m |
| 22 | sd | 0.337 | Snow depth | m of water equivalent |
| 23 | smlt | 0.252 | Snowmelt | m of water equivalent |
| 24 | istl1 | 0.112 | Ice temperature layer 1 | K |
| 25 | istl3 | 0.109 | Ice temperature layer 3 | K |
| 26 | istl2 | 0.100 | Ice temperature layer 2 | K |

13 input structures from observed data are tried and 50 trials are performed for each input structure. The results (Fig. A1) show 7th input structure is the optimal input subset for GBRT.

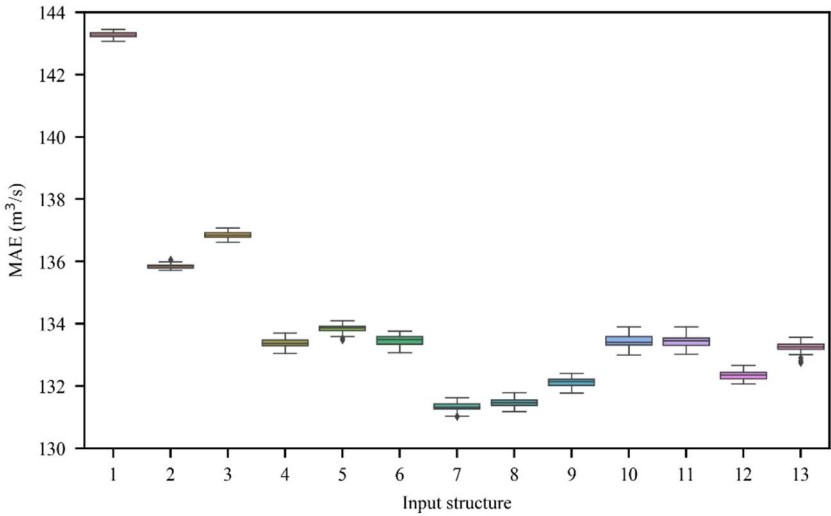

**Figure A1.** Trial results of 13 input structures from observed data.

26 input structures from reanalysis data are tried and 50 trials are performed for each input structure. The results (Fig. A2) show 8th input

structure is the optimal input subset for GBRT-MIC.

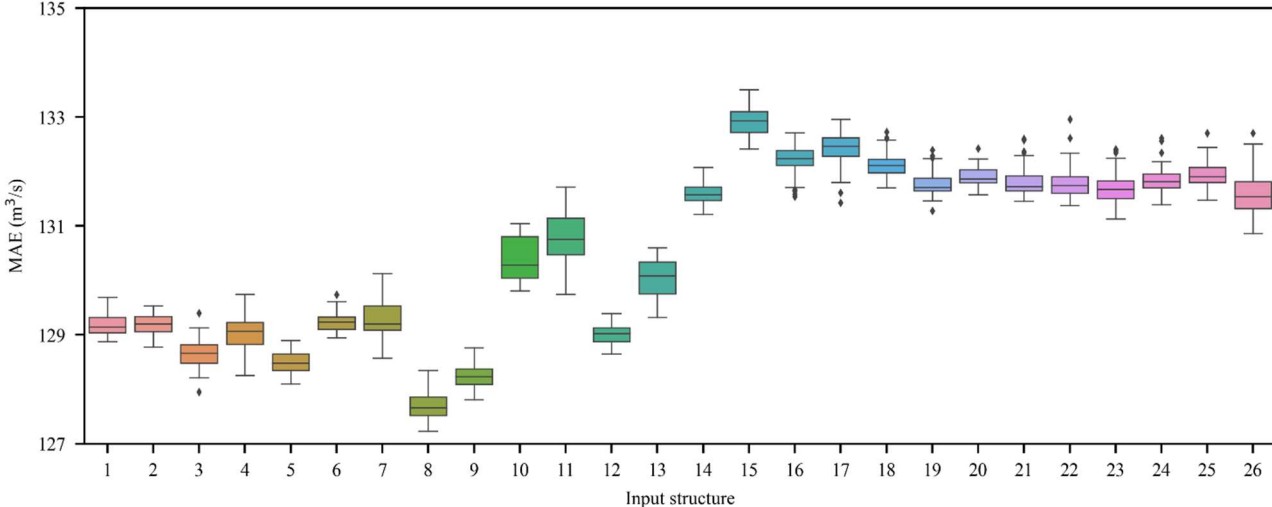

**Figure A2.** Trial results of 13 input structures from reanalysis data.

**Author contributions**

S.L. carried out the study design, the analysis and interpretation of data, and drafted the manuscript. Z.L. and B.L.

participated in the study design, data collection, analysis of data, and preparation of the manuscript. C.C. and Z.Z. carried out the experimental work and the data collection and interpretation. X.J. participated in the design and coordination of experimental work, and acquisition of data. All authors read and approved the final manuscript.

**Competing interests**

The authors declare that they have no conflict of interest.

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

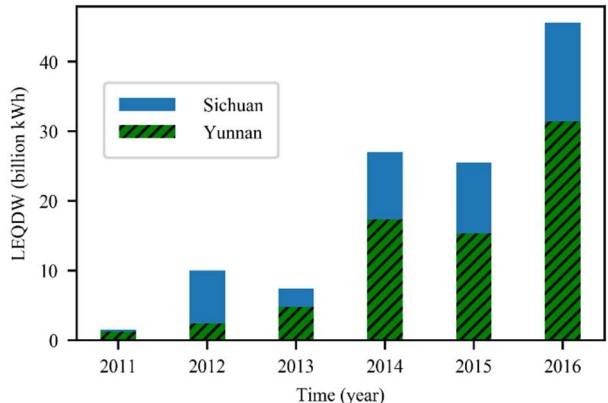

**Figure 1.** Losses of electric quantity due to discarded water (LEQDW) in the Sichuan and Yunnan province.

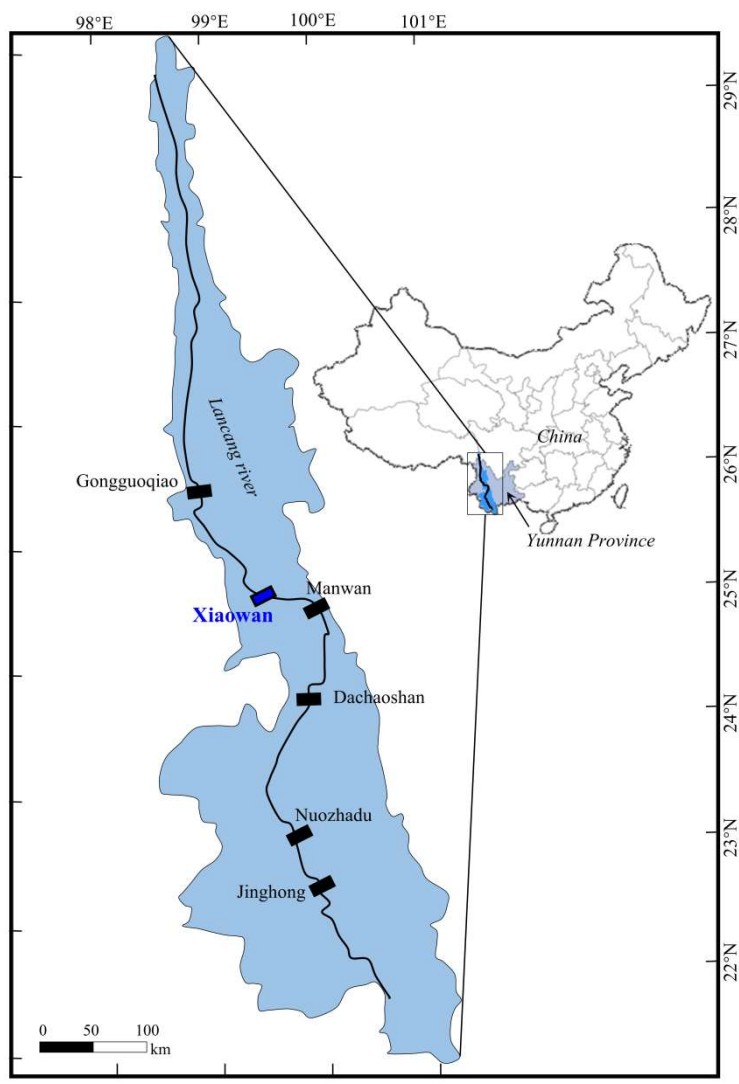

**Figure 2.** Location of the Xiaowan hydropower station.

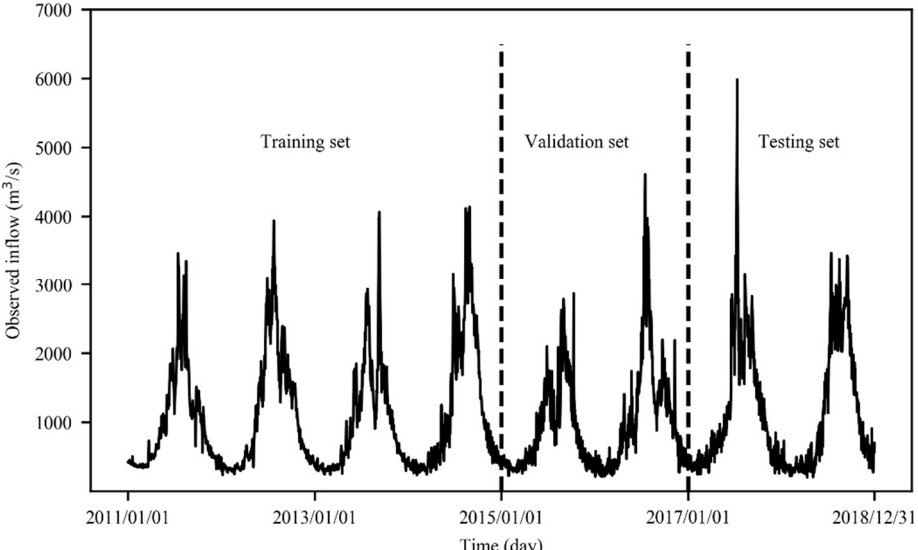

**Figure 3.** Daily inflow series of the Xiaowan hydropower station.

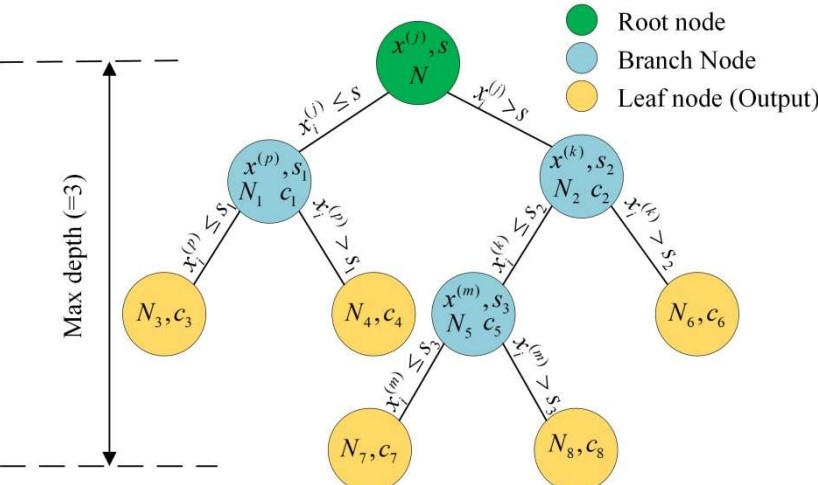

**Figure 4.** The structure of decision tree model.

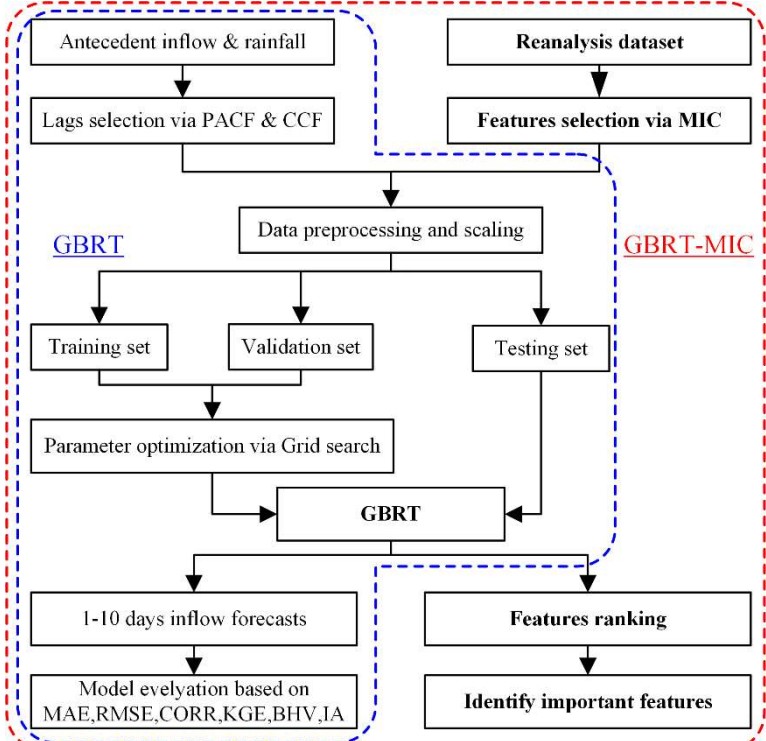

**Figure 5.** Overview of the framework.

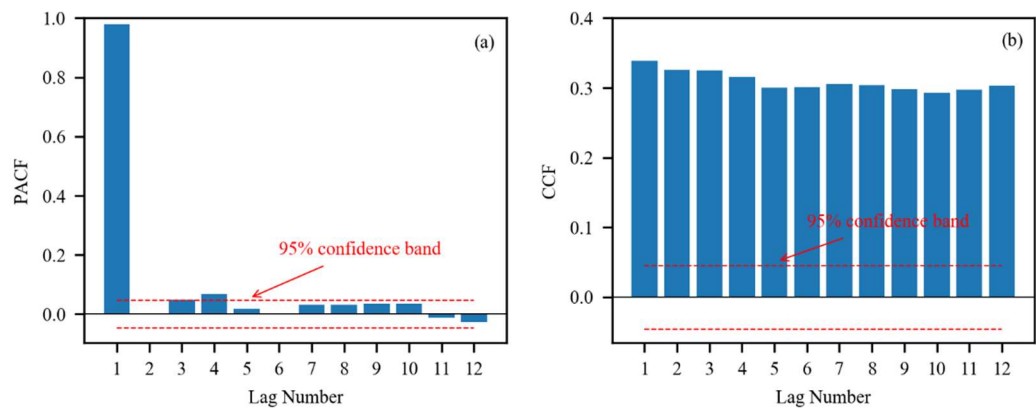

**Figure 6.** PACF plot of Xiaowan daily inflow and CCF of Xiaowan rainfall and inflow. (a) PACF (b) CCF.

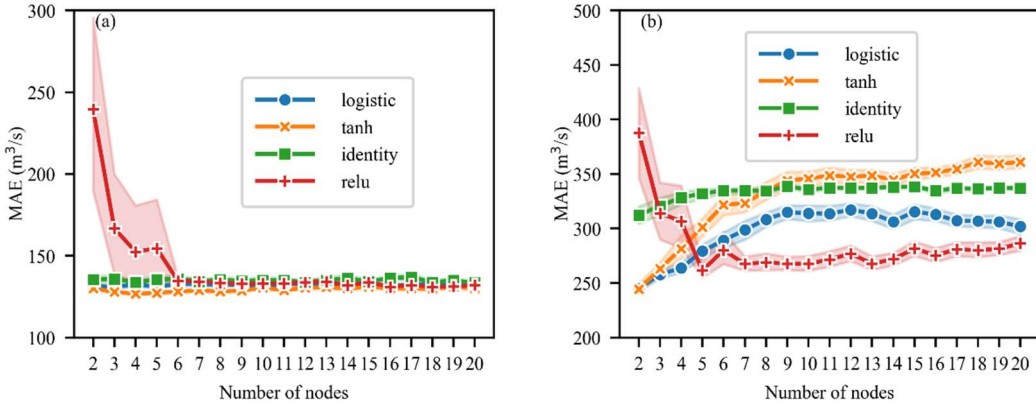

**Figure 7.** Sensitivity of the activation function and the number of nodes in the hidden layer on the MAE of ANN-MIC, the shadow part is 95% confidence interval obtained by bootstrap of 50 trials. (a) One-day ahead (b) Ten-day ahead.

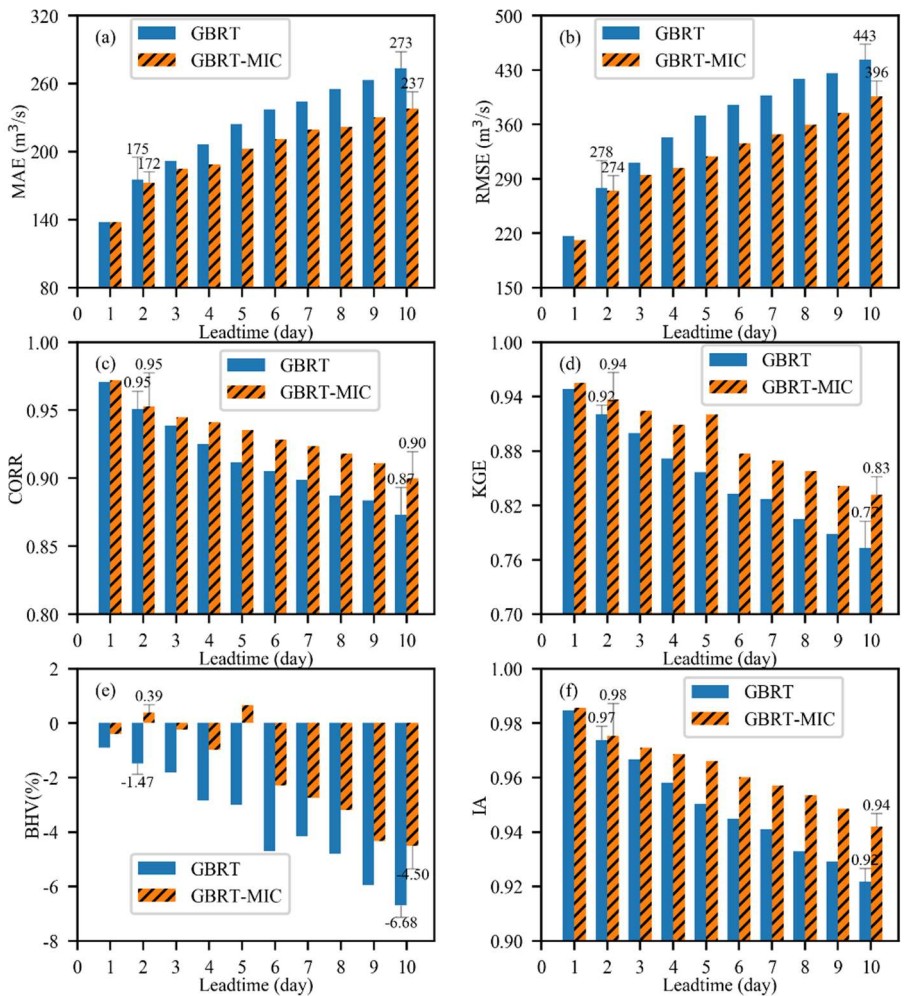

**Figure 8.** Performance of GBRT and GBRT-MIC for the testing set (2017-2018) in term of six indices. (a) MAE (b) RMSE (c) CORR (d) KGE (e) BHV (f) IA.

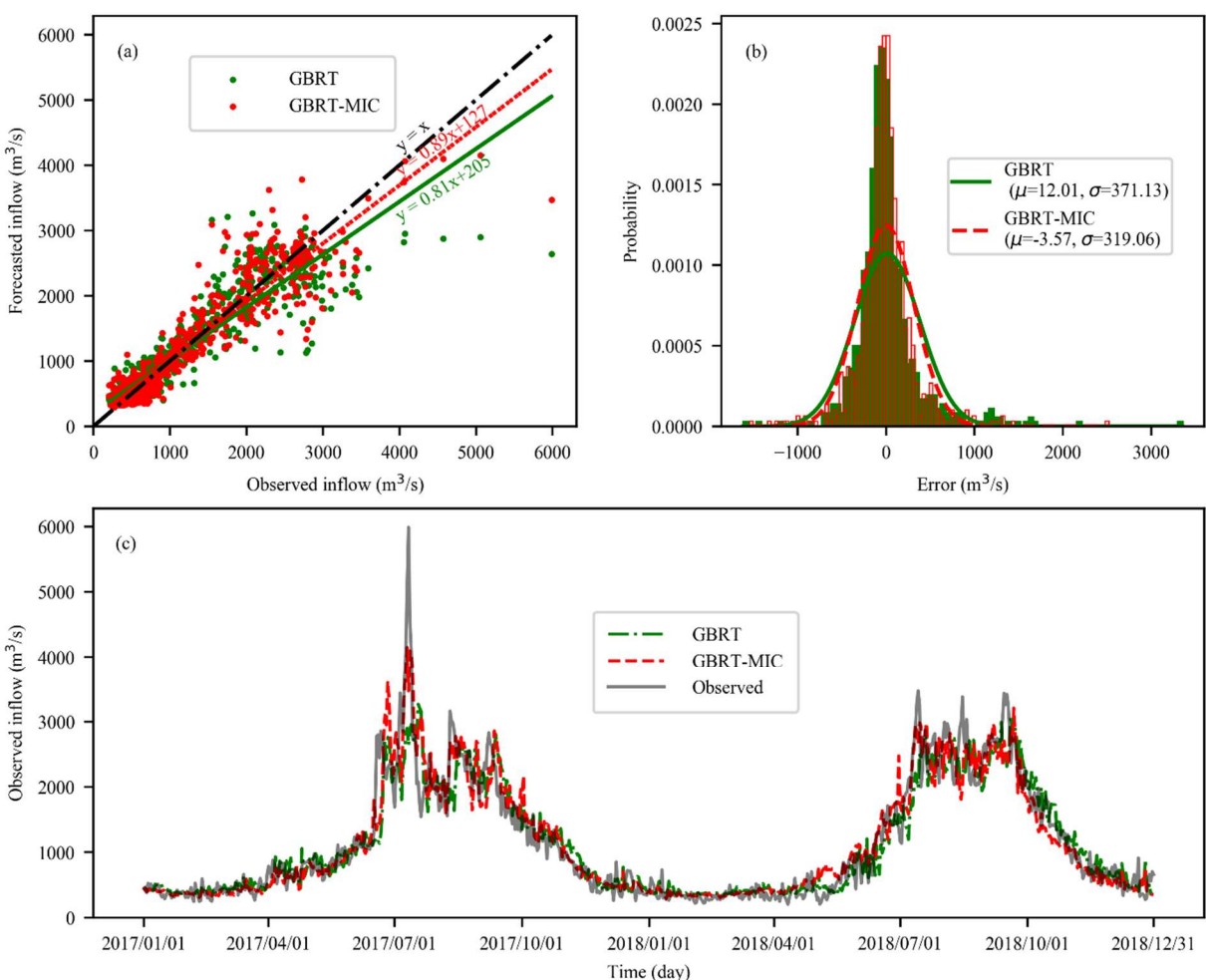

**Figure 9.** Five-day ahead inflow forecasts of the GBRT and GBRT-MIC for the testing set (2017-2018, 730 days). (a) Observed versus forecasted inflow. (b) The histogram of predicting error of testing set (c) Comparison of the observed and forecasted inflow.

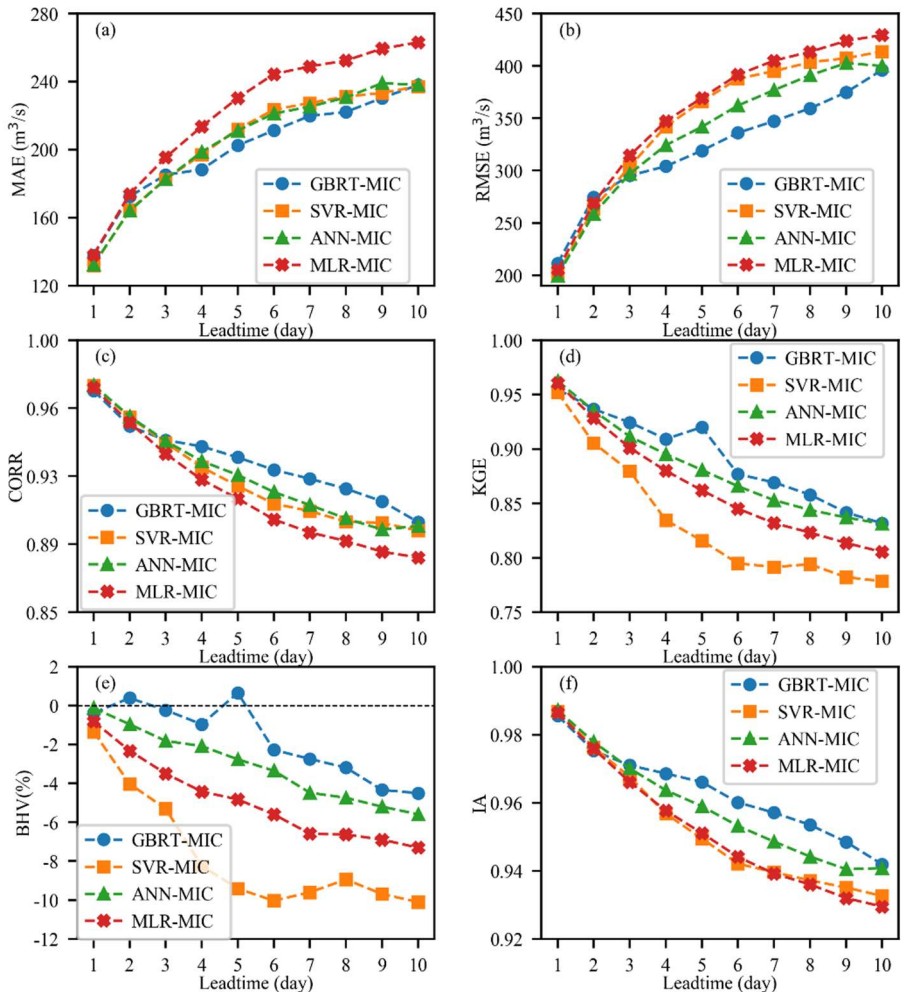

**Figure 10.** Performance of GBRT-MIC, SVR-MIC, ANN-MIC and MLR-MIC for the testing set (2017-2018) in term of six indices. (a) MAE (b) RMSE (c) CORR (d) KGE (e) BHV (f) IA.

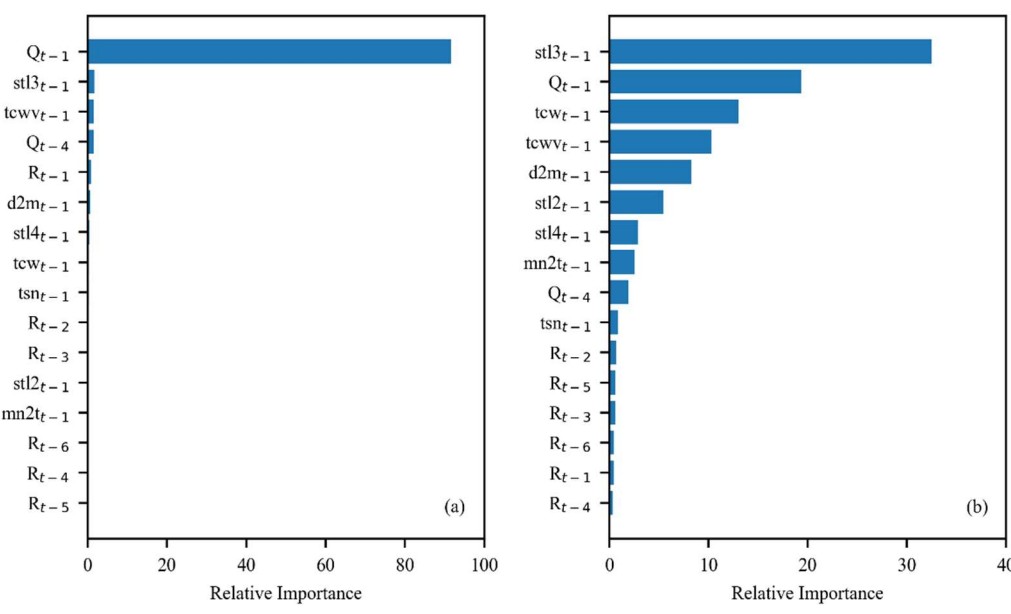

**Figure 11.** Feature importance obtained by GBRT-MIC. (a) One-day ahead (b) Ten-day ahead.

**Table 1.** The candidate inputs via PACF and CCF.

| Number | Input |
|--------|-------|
| 1 | $Q_{t-1}, Q_{t-4}$ |
| 2 | $Q_{t-1}, Q_{t-4}, R_{t-1}$ |
| 3 | $Q_{t-1}, Q_{t-4}, R_{t-1}, R_{t-2}$ |
| 4 | $Q_{t-1}, Q_{t-4}, R_{t-1}, R_{t-2}, R_{t-3}$ |
| 5 | $Q_{t-1}, Q_{t-4}, R_{t-1}, R_{t-2}, R_{t-3}, R_{t-4}$ |
| 6 | $Q_{t-1}, Q_{t-4}, R_{t-1}, R_{t-2}, R_{t-3}, R_{t-4}, R_{t-5}$ |
| 7 | $Q_{t-1}, Q_{t-4}, R_{t-1}, R_{t-2}, R_{t-3}, R_{t-4}, R_{t-5}, R_{t-6}$ |
| 8 | $Q_{t-1}, Q_{t-4}, R_{t-1}, R_{t-2}, R_{t-3}, R_{t-4}, R_{t-5}, R_{t-6}, R_{t-7}$ |
| 9 | $Q_{t-1}, Q_{t-4}, R_{t-1}, R_{t-2}, R_{t-3}, R_{t-4}, R_{t-5}, R_{t-6}, R_{t-7}, R_{t-8}$ |
| 10 | $Q_{t-1}, Q_{t-4}, R_{t-1}, R_{t-2}, R_{t-3}, R_{t-4}, R_{t-5}, R_{t-6}, R_{t-7}, R_{t-8}, R_{t-9}$ |
| 11 | $Q_{t-1}, Q_{t-4}, R_{t-1}, R_{t-2}, R_{t-3}, R_{t-4}, R_{t-5}, R_{t-6}, R_{t-7}, R_{t-8}, R_{t-9}, R_{t-10}$ |
| 12 | $Q_{t-1}, Q_{t-4}, R_{t-1}, R_{t-2}, R_{t-3}, R_{t-4}, R_{t-5}, R_{t-6}, R_{t-7}, R_{t-8}, R_{t-9}, R_{t-10}, R_{t-11}$ |
| 13 | $Q_{t-1}, Q_{t-4}, R_{t-1}, R_{t-2}, R_{t-3}, R_{t-4}, R_{t-5}, R_{t-6}, R_{t-7}, R_{t-8}, R_{t-9}, R_{t-10}, R_{t-11}, R_{t-12}$ |

**Table 2.** The candidate inputs from reanalysis data via MIC.

| Number | Input |
|---|---|
| 1 | $obs, stl3_{t-1}$ |
| 2 | $obs, stl3_{t-1}, d2m_{t-1}$ |
| 3 | $obs, stl3_{t-1}, d2m_{t-1}, tcwv_{t-1}$ |
| 4 | $obs, stl3_{t-1}, d2m_{t-1}, tcwv_{t-1}, tcw_{t-1}$ |
| 5 | $obs, stl3_{t-1}, d2m_{t-1}, tcwv_{t-1}, tcw_{t-1}, stl2_{t-1}$ |
| 6 | $obs, stl3_{t-1}, d2m_{t-1}, tcwv_{t-1}, tcw_{t-1}, stl2_{t-1}, mn2t_{t-1}$ |
| 7 | $obs, stl3_{t-1}, d2m_{t-1}, tcwv_{t-1}, tcw_{t-1}, stl2_{t-1}, mn2t_{t-1}, tsn_{t-1}$ |
| 8 | $obs, stl3_{t-1}, d2m_{t-1}, tcwv_{t-1}, tcw_{t-1}, stl2_{t-1}, mn2t_{t-1}, tsn_{t-1}, stl4_{t-1}$ |
| 9 | $obs, stl3_{t-1}, d2m_{t-1}, tcwv_{t-1}, tcw_{t-1}, stl2_{t-1}, mn2t_{t-1}, tsn_{t-1}, stl4_{t-1}, stl1_{t-1}$ |
| 10 | $obs, stl3_{t-1}, d2m_{t-1}, tcwv_{t-1}, tcw_{t-1}, stl2_{t-1}, mn2t_{t-1}, tsn_{t-1}, stl4_{t-1}, stl1_{t-1}, ro_{t-1}$ |
| 11 | $obs, stl3_{t-1}, d2m_{t-1}, tcwv_{t-1}, tcw_{t-1}, stl2_{t-1}, mn2t_{t-1}, tsn_{t-1}, stl4_{t-1}, stl1_{t-1}, ro_{t-1}, swvl1_{t-1}$ |
| 12 | $obs, stl3_{t-1}, d2m_{t-1}, tcwv_{t-1}, tcw_{t-1}, stl2_{t-1}, mn2t_{t-1}, tsn_{t-1}, stl4_{t-1}, stl1_{t-1}, ro_{t-1}, swvl1_{t-1}, swvl2_{t-1}$ |
| 13 | $obs, stl3_{t-1}, d2m_{t-1}, tcwv_{t-1}, tcw_{t-1}, stl2_{t-1}, mn2t_{t-1}, tsn_{t-1}, stl4_{t-1}, stl1_{t-1}, ro_{t-1}, swvl1_{t-1}, swvl2_{t-1}, swvl3_{t-1}$ |
| 14 | $obs, stl3_{t-1}, d2m_{t-1}, tcwv_{t-1}, tcw_{t-1}, stl2_{t-1}, mn2t_{t-1}, tsn_{t-1}, stl4_{t-1}, stl1_{t-1}, ro_{t-1}, swvl1_{t-1}, swvl2_{t-1}, swvl3_{t-1}, t2m_{t-1}$ |
| 15 | $obs, stl3_{t-1}, d2m_{t-1}, tcwv_{t-1}, tcw_{t-1}, stl2_{t-1}, mn2t_{t-1}, tsn_{t-1}, stl4_{t-1}, stl1_{t-1}, ro_{t-1}, swvl1_{t-1}, swvl2_{t-1}, swvl3_{t-1}, t2m_{t-1}, swvl4_{t-1}$ |
| 16 | $obs, stl3_{t-1}, d2m_{t-1}, tcwv_{t-1}, tcw_{t-1}, stl2_{t-1}, mn2t_{t-1}, tsn_{t-1}, stl4_{t-1}, stl1_{t-1}, ro_{t-1}, swvl1_{t-1}, swvl2_{t-1}, swvl3_{t-1}, t2m_{t-1}, swvl4_{t-1},$ $mx2t_{t-1}$ |
| 17 | $obs, stl3_{t-1}, d2m_{t-1}, tcwv_{t-1}, tcw_{t-1}, stl2_{t-1}, mn2t_{t-1}, tsn_{t-1}, stl4_{t-1}, stl1_{t-1}, ro_{t-1}, swvl1_{t-1}, swvl2_{t-1}, swvl3_{t-1}, t2m_{t-1}, swvl4_{t-1},$ $mx2t_{t-1}, sf_{t-1}$ |
| 18 | $obs, stl3_{t-1}, d2m_{t-1}, tcwv_{t-1}, tcw_{t-1}, stl2_{t-1}, mn2t_{t-1}, tsn_{t-1}, stl4_{t-1}, stl1_{t-1}, ro_{t-1}, swvl1_{t-1}, swvl2_{t-1}, swvl3_{t-1}, t2m_{t-1}, swvl4_{t-1},$ $mx2t_{t-1}, sf_{t-1}, cp_{t-1}$ |
| 19 | $obs, stl3_{t-1}, d2m_{t-1}, tcwv_{t-1}, tcw_{t-1}, stl2_{t-1}, mn2t_{t-1}, tsn_{t-1}, stl4_{t-1}, stl1_{t-1}, ro_{t-1}, swvl1_{t-1}, swvl2_{t-1}, swvl3_{t-1}, t2m_{t-1}, swvl4_{t-1},$ $mx2t_{t-1}, sf_{t-1}, cp_{t-1}, tp_{t-1}$ |
| 20 | $obs, stl3_{t-1}, d2m_{t-1}, tcwv_{t-1}, tcw_{t-1}, stl2_{t-1}, mn2t_{t-1}, tsn_{t-1}, stl4_{t-1}, stl1_{t-1}, ro_{t-1}, swvl1_{t-1}, swvl2_{t-1}, swvl3_{t-1}, t2m_{t-1}, swvl4_{t-1},$ $mx2t_{t-1}, sf_{t-1}, cp_{t-1}, tp_{t-1}, rsn_{t-1}$ |
| 21 | $obs, stl3_{t-1}, d2m_{t-1}, tcwv_{t-1}, tcw_{t-1}, stl2_{t-1}, mn2t_{t-1}, tsn_{t-1}, stl4_{t-1}, stl1_{t-1}, ro_{t-1}, swvl1_{t-1}, swvl2_{t-1}, swvl3_{t-1}, t2m_{t-1}, swvl4_{t-1},$ $mx2t_{t-1}, sf_{t-1}, cp_{t-1}, tp_{t-1}, rsn_{t-1}, lsp_{t-1}$ |
| 22 | $obs, stl3_{t-1}, d2m_{t-1}, tcwv_{t-1}, tcw_{t-1}, stl2_{t-1}, mn2t_{t-1}, tsn_{t-1}, stl4_{t-1}, stl1_{t-1}, ro_{t-1}, swvl1_{t-1}, swvl2_{t-1}, swvl3_{t-1}, t2m_{t-1}, swvl4_{t-1},$ $mx2t_{t-1}, sf_{t-1}, cp_{t-1}, tp_{t-1}, rsn_{t-1}, lsp_{t-1}, sd_{t-1}$ |
| 23 | $obs, stl3_{t-1}, d2m_{t-1}, tcwv_{t-1}, tcw_{t-1}, stl2_{t-1}, mn2t_{t-1}, tsn_{t-1}, stl4_{t-1}, stl1_{t-1}, ro_{t-1}, swvl1_{t-1}, swvl2_{t-1}, swvl3_{t-1}, t2m_{t-1}, swvl4_{t-1},$ $mx2t_{t-1}, sf_{t-1}, cp_{t-1}, tp_{t-1}, rsn_{t-1}, lsp_{t-1}, sd_{t-1}, smlt_{t-1}$ |
| 24 | $obs, stl3_{t-1}, d2m_{t-1}, tcwv_{t-1}, tcw_{t-1}, stl2_{t-1}, mn2t_{t-1}, tsn_{t-1}, stl4_{t-1}, stl1_{t-1}, ro_{t-1}, swvl1_{t-1}, swvl2_{t-1}, swvl3_{t-1}, t2m_{t-1}, swvl4_{t-1},$ $mx2t_{t-1}, sf_{t-1}, cp_{t-1}, tp_{t-1}, rsn_{t-1}, lsp_{t-1}, sd_{t-1}, smlt_{t-1}, istl1_{t-1}$ |
| 25 | $obs, stl3_{t-1}, d2m_{t-1}, tcwv_{t-1}, tcw_{t-1}, stl2_{t-1}, mn2t_{t-1}, tsn_{t-1}, stl4_{t-1}, stl1_{t-1}, ro_{t-1}, swvl1_{t-1}, swvl2_{t-1}, swvl3_{t-1}, t2m_{t-1}, swvl4_{t-1},$ $mx2t_{t-1}, sf_{t-1}, cp_{t-1}, tp_{t-1}, rsn_{t-1}, lsp_{t-1}, sd_{t-1}, smlt_{t-1}, istl1_{t-1}, istl3_{t-1}$ |
| 26 | $obs, stl3_{t-1}, d2m_{t-1}, tcwv_{t-1}, tcw_{t-1}, stl2_{t-1}, mn2t_{t-1}, tsn_{t-1}, stl4_{t-1}, stl1_{t-1}, ro_{t-1}, swvl1_{t-1}, swvl2_{t-1}, swvl3_{t-1}, t2m_{t-1}, swvl4_{t-1},$ $mx2t_{t-1}, sf_{t-1}, cp_{t-1}, tp_{t-1}, rsn_{t-1}, lsp_{t-1}, sd_{t-1}, smlt_{t-1}, istl1_{t-1}, istl3_{t-1}, istl2_{t-1}$ |

*Note: obs* represents the selected observed optimal input set, $obs = \{Q_{t-1}, Q_{t-4}, R_{t-1}, R_{t-2}, R_{t-3}, R_{t-4}, R_{t-5}, R_{t-6}\}$

**Table 3.** List of inputs of GBRT-MIC. There are of two types, observed and reanalysis variables. The reanalysis variables are available two time a day at 00:00 UTC and 12:00 UTC. The cumulative variable (e.g., Total column water) is the sum of two periods and the instantaneous variable (e.g. 2 meter dewpoint temperature) is the mean of two periods.

| Number | Description | Index | Unit | MIC | Type |
|---|---|---|---|---|---|
| 1 | Inflow at day t – 1 | $Q_{t-1}$ | m³ s⁻¹ | - | Obs. |
| 2 | Inflow at day t – 2 | $Q_{t-4}$ | m³ s⁻¹ | - | Obs. |
| 3 | Rainfall at day t – 1 | $R_{t-1}$ | mm | - | Obs. |
| 4 | Rainfall at day t – 2 | $R_{t-2}$ | mm | - | Obs. |
| 5 | Rainfall at day t – 3 | $R_{t-3}$ | mm | - | Obs. |
| 6 | Rainfall at day t – 4 | $R_{t-4}$ | mm | - | Obs. |
| 7 | Rainfall at day t – 5 | $R_{t-5}$ | mm | - | Obs. |
| 8 | Rainfall at day t – 6 | $R_{t-6}$ | mm | - | Obs. |
| 9 | Soil temperature level 3 | $stl3_{t-1}$ | K | 0.847 | ERA-I |
| 10 | 2 meter dewpoint temperature | $2d_{t-1}$ | K | 0.781 | ERA-I |
| 11 | Total column water vapour | $tcwv_{t-1}$ | kg m⁻² | 0.699 | ERA-I |
| 12 | Total column water | $tcw_{t-1}$ | kg m⁻² | 0.699 | ERA-I |
| 13 | Soil temperature level 2 | $stl2_{t-1}$ | K | 0.689 | ERA-I |
| 14 | Minimum temperature at 2 meters | $mn2t_{t-1}$ | K | 0.684 | ERA-I |
| 15 | Temperature of snow layer | $tsn_{t-1}$ | K | 0.664 | ERA-I |
| 16 | Soil temperature level 4 | $stl4_{t-1}$ | K | 0.643 | ERA-I |

**Table 4.** Four commonly used activation functions for ANN-MIC.

| Name | Functional expression |
|---|---|
| Logistic | $f(x) = \dfrac{1}{1 + e^{-x}}$ |
| Tanh | $f(x) = \dfrac{e^x - e^{-x}}{e^x + e^{-x}}$ |
| Identity | $f(x) = x$ |
| Relu | $f(x) = max(0, x)$ |

**Table 5.** Tuning parameters of ANN-MIC and SVR-MIC.

| Model | Tuning parameter | Tuning range | 1 | 2 | 3 | 4 | 5 | 6 | 7 | 8 | 9 | 10 |
|---|---|---|---|---|---|---|---|---|---|---|---|---|
| ANN-MIC | Structure | / | 19-4-1 | 19-2-1 | 19-3-1 | 19-2-1 | 19-2-1 | 19-2-1 | 19-2-1 | 19-2-1 | 19-2-1 | 19-2-1 |
| | Activate function | / | Tanh | tanh | logistic | logistic | logistic | logistic | logistic | logistic | tanh | tanh |
| SVR-MIC | C | **(1, 100, 20)** | 6.2105 | 1.0000 | 1.0000 | 1.0000 | 11.4211 | 1.0000 | 1.0000 | 6.2105 | 1.0000 | 6.2105 |
| | epsilon | **(0.001, 0.1, 20)** | 0.0069 | 0.0084 | 0.0017 | 0.0079 | 0.0017 | 0.0001 | 0.0022 | 0.0006 | 0.0048 | 0.0043 |
| | gamma | **(0.001, 0.1, 20)** | 0323 | 0.0583 | 0.0844 | 0.0271 | 0.0062 | 0.0218 | 0.0375 | 0.0166 | 0.0687 | 0.0166 |

Note: The bold parts, (min, max, step) represent [ $min + \dfrac{max - min}{step - 1} \times 0$ , $min + \dfrac{max - min}{step - 1} \times 1$ , …, $min + \dfrac{max - min}{step - 1} \times (step - 1)$ ].

**Table 6.** Tuning parameters of GBRT and GBRT-MIC.

| Tuning parameter | Tuning range | Optimal parameters (the lead times of 1-10 days) | |
|---|---|---|---|
| | | GBRT | GBRT-MIC |
| max_leaf_nodes | [2, 4, 6, …, 40] | 8, 4, 4, 4, 4, 2, 4, 2, 2, 2 | 7, 9, 13, 7, 15, 4, 5, 4, 4, 17 |
| min_samples_leaf | [1, 6, 11, …, 46] | 6, 31, 1, 1, 1, 31, 6, 1, 6, 1 | 2, 7, 2, 4, 2, 1, 10, 10, 8, 1 |
| max_depth | [1, 2, 3, …, 10] | 3, 2, 2, 2, 3, 1, 3, 1, 1, 1 | 4, 6, 8, 5, 9, 9, 2, 2, 7, 2 |
| min_samples_split | [2, 4, 6, …, 40] | 18, 2, 16, 16, 24, 2, 16, 2, 2, 2 | 18, 15, 12, 13, 8, 3, 19, 3, 19, 8 |
| n_estimators | [100, 200, 300, …, 4000] | 1100, 900, 1200, 700, 700, 1200, 600, 1100, 900, 900 | 3800, 2700, 1300, 900, 1000, 700, 1400, 2000, 1300, 1200 |

**Table 7.** Performance indices of the training set.

| Indice | Model | 1 | 2 | 3 | 4 | 5 | 6 | 7 | 8 | 9 | 10 |
|---|---|---|---|---|---|---|---|---|---|---|---|
| MAE (m³/s) | GBRT-MIC | **56** | **63** | **78** | **122** | **89** | **163** | **161** | **155** | **161** | **172** |
| | SVR-MIC | 98 | 126 | 144 | 162 | 173 | 183 | 188 | 194 | 197 | 203 |
| | ANN-MIC | 99 | 129 | 148 | 162 | 172 | 184 | 192 | 196 | 203 | 205 |
| | MLR-MIC | 103 | 136 | 159 | 175 | 187 | 198 | 207 | 215 | 221 | 228 |
| RMSE (m³/s) | GBRT-MIC | **77** | **87** | **107** | **185** | **124** | **257** | **255** | **245** | **254** | **278** |
| | SVR-MIC | 153 | 212 | 247 | 280 | 300 | 319 | 329 | 337 | 344 | 353 |
| | ANN-MIC | 151 | 206 | 240 | 264 | 284 | 304 | 318 | 328 | 334 | 339 |
| | MLR-MIC | 157 | 214 | 250 | 275 | 295 | 315 | 330 | 342 | 352 | 361 |
| CORR | GBRT-MIC | **0.9952** | **0.9940** | **0.9908** | **0.9724** | **0.9877** | **0.9464** | **0.9468** | **0.9510** | **0.9476** | **0.9366** |
| | SVR-MIC | 0.9811 | 0.9641 | 0.9511 | 0.9380 | 0.9286 | 0.9186 | 0.9126 | 0.9073 | 0.9039 | 0.8977 |
| | ANN-MIC | 0.9815 | 0.9653 | 0.9528 | 0.9424 | 0.9331 | 0.9232 | 0.9156 | 0.9101 | 0.9066 | 0.9036 |
| | MLR-MIC | 0.9801 | 0.9628 | 0.9485 | 0.9376 | 0.9278 | 0.9172 | 0.9090 | 0.9019 | 0.8959 | 0.8900 |
| KGE | GBRT-MIC | **0.9884** | **0.9827** | **0.9738** | **0.9439** | **0.9642** | **0.9009** | **0.9069** | **0.9099** | **0.9002** | **0.8877** |
| | SVR-MIC | 0.9618 | 0.9207 | 0.8982 | 0.8613 | 0.8445 | 0.8266 | 0.8223 | 0.8247 | 0.8149 | 0.8103 |
| | ANN-MIC | 0.9735 | 0.9508 | 0.9325 | 0.9177 | 0.9048 | 0.8907 | 0.8800 | 0.8724 | 0.8668 | 0.8611 |
| | MLR-MIC | 0.9718 | 0.9473 | 0.9272 | 0.9117 | 0.8979 | 0.8829 | 0.8713 | 0.8613 | 0.8528 | 0.8444 |
| BHV (%) | GBRT-MIC | -0.3025 | -0.6382 | -0.8986 | -1.3422 | -1.4019 | -1.5485 | -1.7486 | -1.7692 | -2.6647 | -3.0375 |
| | SVR-MIC | -1.3488 | -3.3959 | -4.0686 | -6.9058 | -7.5421 | -8.2216 | -6.9950 | -6.1996 | -6.2406 | -5.6687 |
| | ANN-MIC | **-0.1814** | **-0.2586** | **-0.7710** | **-0.7723** | **-0.6249** | **-0.6815** | **-0.6878** | **-0.8821** | **-0.6487** | **-0.1239** |
| | MLR-MIC | -0.4668 | -1.0527 | -1.5863 | -1.9709 | -1.9477 | -2.1634 | -2.0182 | -1.8074 | -2.0454 | -1.7473 |
| IA | GBRT-MIC | **0.9976** | **0.9969** | **0.9952** | **0.9854** | **0.9935** | **0.9706** | **0.9712** | **0.9734** | **0.9712** | **0.9650** |
| | SVR-MIC | 0.9902 | 0.9804 | 0.9727 | 0.9636 | 0.9574 | 0.9506 | 0.9472 | 0.9449 | 0.9421 | 0.9386 |
| | ANN-MIC | 0.9906 | 0.9820 | 0.9752 | 0.9695 | 0.9643 | 0.9586 | 0.9541 | 0.9509 | 0.9487 | 0.9468 |
| | MLR-MIC | 0.9898 | 0.9807 | 0.9729 | 0.9668 | 0.9613 | 0.9551 | 0.9502 | 0.9460 | 0.9423 | 0.9387 |

*Note :* The bold numbers represent the values of performance criterion for the best fitted models.

635

**Table 8.** Performance indices of the testing set.

| Indice | Model | 1 | 2 | 3 | 4 | 5 | 6 | 7 | 8 | 9 | 10 |
|---|---|---|---|---|---|---|---|---|---|---|---|
| MAE (m³/s) | GBRT-MIC | 137 | 172 | 185 | **188** | **202** | **211** | **219** | **222** | **230** | 237 |
| | SVR-MIC | **131** | 164 | 182 | 197 | 212 | 223 | 227 | 231 | 233 | **237** |
| | ANN-MIC | 132 | **163** | **182** | 198 | 211 | 221 | 225 | 230 | 239 | 238 |
| | MLR-MIC | 138 | 173 | 195 | 213 | 230 | 244 | 248 | 252 | 259 | 263 |
| RMSE (m³/s) | GBRT-MIC | 211 | 274 | **295** | 304 | 319 | 336 | 347 | 359 | 374 | 396 |
| | SVR-MIC | 200 | 263 | 303 | 342 | 366 | 387 | 395 | 403 | 407 | 413 |
| | ANN-MIC | **199** | **258** | 296 | 324 | 341 | 362 | 376 | 391 | 402 | 399 |
| | MLR-MIC | 205 | 268 | 314 | 347 | 369 | 391 | 404 | 413 | 423 | 429 |
| CORR | GBRT-MIC | 0.9722 | 0.9526 | **0.9449** | **0.9414** | **0.9354** | **0.9285** | **0.9236** | **0.9181** | **0.9112** | **0.8997** |
| | SVR-MIC | 0.9751 | 0.9575 | 0.9434 | 0.9300 | 0.9196 | 0.9099 | 0.9058 | 0.8999 | 0.8993 | 0.8950 |
| | ANN-MIC | **0.9752** | **0.9580** | 0.9444 | 0.9333 | 0.9257 | 0.9163 | 0.9091 | 0.9017 | 0.8956 | 0.8975 |
| | MLR-MIC | 0.9738 | 0.9545 | 0.9374 | 0.9231 | 0.9126 | 0.9012 | 0.8940 | 0.8893 | 0.8834 | 0.8802 |
| KGE | GBRT-MIC | 0.9550 | **0.9367** | **0.9244** | **0.9092** | **0.9200** | **0.8769** | **0.8693** | **0.8580** | **0.8417** | **0.8317** |
| | SVR-MIC | 0.9520 | 0.9055 | 0.8797 | 0.8347 | 0.8158 | 0.7950 | 0.7915 | 0.7941 | 0.7822 | 0.7786 |
| | ANN-MIC | **0.9625** | 0.9352 | 0.9115 | 0.8953 | 0.8808 | 0.8658 | 0.8530 | 0.8440 | 0.8371 | 0.8313 |
| | MLR-MIC | 0.9605 | 0.9284 | 0.9011 | 0.8800 | 0.8620 | 0.8452 | 0.8319 | 0.8232 | 0.8137 | 0.8054 |
| BHV (%) | GBRT-MIC | -0.3826 | **0.3880** | -0.2319 | -0.9629 | 0.6566 | -2.2766 | -2.7422 | -3.1924 | -4.3363 | -4.5040 |
| | SVR-MIC | -1.3382 | -4.0253 | -5.3037 | -8.2410 | -9.4167 | -10.0357 | -9.6049 | -8.9452 | -9.6886 | -10.1058 |
| | ANN-MIC | **-0.1228** | -0.9608 | -1.8150 | -2.0839 | -2.7642 | -3.3509 | -4.4831 | -4.7424 | -5.1999 | -5.5886 |
| | MLR-MIC | -0.8093 | -2.3244 | -3.4945 | -4.4210 | -4.8268 | -5.5955 | -6.5914 | -6.6302 | -6.8944 | -7.3080 |
| IA | GBRT-MIC | 0.9856 | 0.9753 | **0.9710** | **0.9686** | **0.9661** | **0.9601** | **0.9571** | **0.9535** | **0.9485** | **0.9419** |
| | SVR-MIC | 0.9869 | 0.9763 | 0.9676 | 0.9568 | 0.9495 | 0.9421 | 0.9396 | 0.9372 | 0.9351 | 0.9326 |
| | ANN-MIC | **0.9872** | **0.9779** | 0.9701 | 0.9637 | 0.9590 | 0.9532 | 0.9486 | 0.9442 | 0.9405 | 0.9408 |
| | MLR-MIC | 0.9865 | 0.9759 | 0.9661 | 0.9577 | 0.9511 | 0.9441 | 0.9392 | 0.9360 | 0.9320 | 0.9295 |

*Note :* The bold numbers represent the values of performance criterion for the best fitted models.