# Peer review of "Multi-step ahead daily inflow forecasting using ERA-Interim reanalysis dataset based on gradient boosting regression trees"

_Hydrology and Earth System Sciences, 2019_

## Referee Comment (RC1) · Anonymous Referee #1 · 21 Dec 2019

1. The manuscript presents multi-step ahead daily inflow forecasting using ERA-Interim reanalysis dataset based on gradient boosting regression trees, which is interesting. It is relevant and within the scope of the journal. 2. However, the manuscript, in its present form, contains several weaknesses. Appropriate revisions to the following points should be undertaken in order to justify recommendation for publication. 3. Full names should be shown for all abbreviations in their first occurrence in texts. For example, ERA in p.1, ECMWF in p.3, etc. 4. For readers to quickly catch your contribution, it would be better to highlight major difficulties and challenges, and your original achievements to overcome them, in a clearer way in abstract and introduction. 5. It is mentioned in p.1 that ERA-Interim reanalysis data is adopted as input. What are

other feasible alternatives? What are the advantages of adopting this particular data over others in this case? How will this affect the results? The authors should provide more details on this. 6. It is mentioned in p.1 that gradient boosting regression tree is adopted as inflow forecast framework. What are the advantages of adopting this particular soft computing technique over others in this case? How will this affect the results? The authors should provide more details on this. 7. It is mentioned in p.1 that artificial neural networks, support vector regression and multiple linear regression models are adopted as benchmark for comparison. What are the other feasible alternatives? What are the advantages of adopting these particular models over others in this case? How will this affect the results? More details should be furnished. 8. It is mentioned in p.3 that the Xiaowan Hydropower Station is adopted as the case study. What are other feasible alternatives? What are the advantages of adopting this particular case study over others in this case? How will this affect the results? The authors should provide more details on this. 9. It is mentioned in p.4 that the maximum information coefficient is adopted to select inputs from 79 potential predictors from reanalysis data. What are the advantages of adopting this particular approach over others in this case? How will this affect the results? The authors should provide more details on this. 10. It is mentioned in p.4 that autocorrelation function is adopted to identify observed inflow and rainfall lags. What are other feasible alternatives? What are the advantages of adopting this particular approach over others in this case? How will this affect the results? The authors should provide more details on this. 11. It is mentioned in p.6 that four evaluation criteria are adopted to evaluate the performance of the models. What are the other feasible alternatives? What are the advantages of adopting these particular evaluation criteria over others in this case? How will this affect the results? More details should be furnished. 12. It is mentioned in p.7 that a grid search algorithm is adopted to optimization model parameters. What are other feasible alternatives? What are the advantages of adopting this particular algorithm over others in this case? How will this affect the results? The authors should provide more details on this. 13. It is mentioned in p.9 that grid searching is adopted to tune the hyperparameters of GBRT,

GBRT-MIC, ANN-MIC. What are other feasible alternatives? What are the advantages of adopting this particular approach over others in this case? How will this affect the results? The authors should provide more details on this. 14. It is mentioned in p.9 that Bayesian optimization (Snoek et al., 2012) is adopted to tune the hyperparameters of SVR-MIC. What are other feasible alternatives? What are the advantages of adopting this particular approach over others in this case? How will this affect the results? The authors should provide more details on this. 15. It is mentioned in p.9 that Python is adopted to perform all computations. What are other feasible alternatives? What are the advantages of adopting this particular software over others in this case? How will this affect the results? The authors should provide more details on this. 16. Some key parameters are not mentioned. The rationale on the choice of the particular set of parameters should be explained with more details. Have the authors experimented with other sets of values? What are the sensitivities of these parameters on the results? 17. Some assumptions are stated in various sections. Justifications should be provided on these assumptions. Evaluation on how they will affect the results should be made. 18. The discussion section in the present form is relatively weak and should be strengthened with more details and justifications. 19. Moreover, the manuscript could be substantially improved by relying and citing more on recent literatures about contemporary real-life case studies of soft computing techniques in hydrological prediction such as the followings: ïĄň Yaseen, Z.M., et al., "An enhanced extreme learning machine model for river flow forecasting: state-of-the-art, practical applications in water resource engineering area and future research direction," Journal of Hydrology 569: 387-408 2019. ïĄň Fotovatikhah, F., et al., "Survey of Computational Intelligence as Basis to Big Flood Management: Challenges, research directions and Future Work," Engineering Applications of Computational Fluid Mechanics 12 (1): 411-437 2018. ïĄň Mosavi, A., et al., "Flood Prediction Using Machine Learning Models: Literature Review," Water 10 (11): article no. 1536 2018. ïĄň Moazenzadeh, R., et al., "Coupling a firefly algorithm with support vector regression to predict evaporation in northern Iran," Engineering Applications of Computational Fluid Mechanics 12 (1): 584-597 2018.

ïĄň Ghorbani, M.A., et al., "Forecasting pan evaporation with an integrated Artificial Neural Network Quantum-behaved Particle Swarm Optimization model: a case study in Talesh, Northern Iran," Engineering Applications of Computational Fluid Mechanics 12 (1): 724-737 2018. ïĄň Chau, K.W., et al., "Use of Meta-Heuristic Techniques in Rainfall-Runoff Modelling" Water 9(3): article no. 186, 6p 2017. 20. Some inconsistencies and minor errors that needed attention are: ïĄň Replace ". . .was supply to depict. . ." with ". . .was supplied to depict. . ." in line 86 of p.3 ïĄň Replace ". . .into train set, validation set, and test set. . ." with ". . .into training set, validation set, and testing set. . ." in lines 206-207 of p.7 ïĄň Replace ". . .test set. . ." with ". . .testing set. . ." in line 209 of p.7 ïĄň Replace ". . .more accuracy inflow forecasting. . ." with ". . .more accurate inflow forecasting. . ." in line 283 of p.10 ïĄň Replace ". . .arisen in in areas. . ." with ". . .arisen in areas. . ." in line 288 of p.10 ïĄň Replace ". . .for train, validation and test set. . ." with ". . .for training, validation and testing set. . ." in line 294 of p.10 ïĄň Replace ". . .According to compare the forecasted results of. . ." with ". . .According to the comparison of forecasted results of. . ." in line 330 of p.11 21. In the conclusion section, the limitations of this study, suggested improvements of this work and future directions should be highlighted.

---

## Referee Comment (RC2) · Shengli Liao et al. · 27 Jan 2020

Dear Editor/Authors I went through the manuscript. Generally, the manuscript has been well organized. However, it needs substantial revisions to be considered for publications in the journal. My main concerns can be summarized as follows: 1- P1, L6, In abstract the authors stated that "The impacts of climate change and human activities make accurate inflow prediction increasingly difficult, especially for longer lead times". As far as I know, the climate change deals with long term trends, say the climate variation over 20 years. I cannot understand relevance of the abovementioned with climate change impacts and human activities. 2- The authors have to clearly indicate which model was developed for the inflow forecasting. At first they have to demonstrate if they used conceptual models or data driven models. What is the advantage of the

developed model? 3- The input selection for multi-day ahead forecasting should be discussed according to available literature. It is essential why the input structure of the longer period is not updated following the earlier stage forecasts. 4- Literature should be updated discussing on more papers addressing multi-step ahead forecasting. 5- The authors employed gradient boosting regression trees as an ensemble framework. More explanations required about ensemble members. 6- Uncertainty analysis should be carried out to show how much the predictions are confident. As the lead time increases, the metrics reveal errors are increasing drastically. Moreover, high uncertainties are expected to associate with such models. Please discuss this issues accordingly. 7- Concerning inflow predictions, please indicate efficiency of the proposed model to simulate and predict extreme values which are of great importance.

---

## Referee Comment (RC3) · Anonymous Referee #3 · 29 Jan 2020

In this manuscript, the authors compared several data-driven models for multi-step forecasting of inflow. The employed models include gradient boosting regression trees (GBRT), artificial neural networks (ANN), support vector regression (SVR), and multiple linear regression (MLR) models. The models were developed by considering (1) streamflow and rainfall record, and (2) ERA-Interim reanalysis data. Further, the maximum information coefficient and autocorrelation functions were utilized to construct the input structures of the models. The authors concluded that the developed methodology that considers ERA-Interim reanalysis data considerably gives better results in the forecasting of inflows at lead times of 5-10 days. The manuscript is well written and organized. However, there is not a significant novelty in the manuscript except using

ERA-Interim dataset. Further, there are severe weaknesses in the developing of the model input structures. Please see my comments below.

1. The authors made a significant mistake in using the autocorrelation function (ACF) in determining the model structures. They should have employed cross-correlation and partial autocorrelation functions (or other measures) to establish the relationship between the observed records and inflow. The ACF only measures the dependency or relationship of observed value with lagged observations of a considered variable. In a long-dependent series such as inflow time series, the ACF will decay slowly. Therefore, statistically significant relationships between the observed and lagged values could not be determined. To determine the significant relationships, the authors employed user-defined threshold value. The obtained inflow and rainfall values for the input structures of the models include only three lagged-day values as could be expected. This number could be higher based on the selected threshold. However, this finding does not convey any meaningful relationship between the observed records (i.e. inflow and rainfall) and the inflow values. The PACF should have been used for determining the lagged relationships of inflows since the inflow time series mainly shows the long-memory feature where the correlation decays after a long observation period. Further, all statistically significant lagged variables should have been included in the model structures found in PACF. Using a user-defined threshold value is a serious mistake in this situation. 2. The authors claimed that the proposed methodology "significantly" improves the accuracy of inflow prediction for longer lead times. However, I do not agree with this comment. Because, as the authors mentioned, there is only about 1% and 5% improvement in two-day and 10-day ahead forecasting. Therefore, the results do not seem convincing about the superiority of ERA-Interim dataset over the common dataset, especially ill-conditioned input structures with conventional observed inflow and rainfall dataset. 3. The authors found that three-day lagged values of inflow and rainfall have less impact on 10-day ahead forecasting of inflow in Section 4.5. This is a clue that more lagged values of input variables should have been included in the models' structure. 4. The employed performance indices, specifically the coefficient of determination, seems

[Figure]

insufficient to compare several model performances. More distinctive performance indices such as degree of agreement and Kling-Gupta efficiency metrics should have been used. 5. It is not clear how the multi-step forecasting scheme (i.e., recursive or static) was employed? Please give more details about this issue. 6. The selected ranges of the model parameters seem highly subjective. Please justify the selected ranges of the model parameters, especially in Section 4.2. 7. The range for the number of hidden neurons (i.e. 2–20) seems too high. Please justify this from a hydrological perspective. Because using a high number of hidden neurons could lead to overfitting that resulted in a poor performance in multi-step forecasting. 8. The discussion of the obtained results should be improved with more details, especially giving necessary citations to previous studies. 9. It is not clear how Fig. 1 was obtained. Please give the necessary information about this figure. 10. Please give more details on the Lines 78–82. 11. Please give the definitions and meanings of the variables in the ERA-Interim dataset in the Appendix. 12. Please justify using the feature scaling in Line 108. 13. What do you mean with "invalid variables" in Line 116? 14. Please prefer "maximal" or "maximum" information criterion throughout the manuscript. 15. Please check the term $MI^*(D,X,Y)$ in Eq. (5) since you defined $MI^*(D,x,y)$ in Line 130. 16. The definition of $B(n)$ was given in Line 133; however it is not clear where this parameter is used. 17. Please check the terms in Eq. (7). Will they be $R1(i,s)$ or $R1(j,s)$? 18. Please check the notations in Line 144; n features with N samples or n samples with N features according to the given definition. 19. There is little information about the structure of ERA-Interim dataset. Please give more details about this dataset. 20. There is not any information about grid searching methodology. 21. Please add "activation function" after "relu" in Line 248. 22. The comments in Lines 278–280 are vague. 23. The authors did not discuss the reasons why NSE values for lead times of 6-7-8-9-day is worse than the value of lead time of 10-day. 24. It is not clear how top k features were selected according to the chosen threshold value. Did the authors employ several threshold values? Please give more details on this issue.

---

## Author Comment (AC1) · 6 Feb 2020

Thank you very much for your time and for your thoughtful and constructive review. The following are our point-by-point responses to your comments.

*1. The manuscript presents multi-step ahead daily inflow forecasting using ERA-Interim reanalysis dataset based on gradient boosting regression trees, which is interesting. It is relevant and within the scope of the journal.*

**Response:** Thank you very much for your positive comments.

**Proposed changes to manuscript:** N/A

*2. Full names should be shown for all abbreviations in their first occurrence in texts. For example, ERA in page 1, ECMWF in page 3, etc.*

**Response:** Thank you for your carefulness.

**Proposed changes to manuscript:** We will show full names for all abbreviations in their first occurrence in the revised manuscript.

*3. For readers to quickly catch your contribution, it would be better to highlight major difficulties and challenges, and your original achievements to overcome them, in a clearer way in abstract and introduction.*

**Response:** Thank you for your suggestion. The major difficulties and challenges are high precision model input related to inflow of longer lead times and effective prediction model. This paper proposed a new hybrid inflow forecast framework with ERA-Interim (European Centre for Medium-Range Weather Forecasts (ECMWF) Re-Analysis Interim) data as input, adopting gradient boosting regression trees (GBRT) and the maximum information coefficient (MIC) was developed for multi-step ahead daily inflow forecasting. The proposed method overcomes the difficulties from three aspects. Firstly, the ERA-Interim dataset provides enough information for the framework to discover inflow for longer lead times. Secondly, MIC can identify effective feature subset from massive features that significantly affects inflow so that the framework can avoid over-fitting, distinguish key attributes with unimportant ones and provide a concise understanding of inflow. Lastly, the GBRT is a prediction model in the form

of an ensemble of decision trees and has a strong ability to capture nonlinear relationships between input and output in long lead times more fully.

**Proposed changes to manuscript:** We will make careful modifications in Section "Abstract" and "Introduction" of the revised manuscript.

*4. It is mentioned in page 1 that ERA-Interim reanalysis data is adopted as input. What are other feasible alternatives? What are the advantages of adopting this particular data over others in this case? How will this affect the results? The authors should provide more details on this.*

**Response:** Thank you for your careful review and suggestion. The ERA-Interim data are the result of assimilating observed data with forecast data, which has less error than observed data and forecast data (Balsamo et al., 2015). ERA-Interim shows the results of a global climate reanalysis from 1979 to date, which are produced by a fixed version of a NWP system (Dee et al., 2011). The fixed version ensures there are no spurious trends caused by an evolving NWP system. Therefore, meteorological reanalysis data satisfies the need for long sequences of consistent data and have been used for the prediction of wind speeds (Stopa and Cheung 2014) and solar radiation (Linares-Rodríguez, Ruiz-Arias et al. 2011, Ghimire, Deo et al. 2019). Meanwhile, ERA-Interim was proved to be one of the best reanalysis data describing atmospheric circulation and elements (Kishore et al., 2011).

**Proposed changes to manuscript:** More details about ERA-Interim data will be given in Section "Appendix" of the revised manuscript for readers to quickly catch the contribution.

*5. It is mentioned in page 1 that gradient boosting regression tree is adopted as inflow forecast framework. What are the advantages of adopting this particular soft computing technique over others in this case? How will this affect the results? The authors should provide more details on this.*

**Response:** Thank you for your careful review and suggestion. The gradient boosting regression trees (GBRT) (Friedman 2001, Fienen, Nolan et al. 2018), a nonparametric machine learning method based on a boosting strategy and decision trees, was developed and had been used in traffic (Zhan, Zhang et al. 2019) and environmental (Wei, Meng et al. 2019) field and proved to alleviate the problems of being trapped by local minima, over-fitting problems and reduced generalizing performance.

**Proposed changes to manuscript:** More details about GBRT and other soft computing techniques will be given in Section 3 of the revised manuscript according to your suggestion.

*6. It is mentioned in page 1 that artificial neural networks, support vector regression and multiple linear regression models are adopted as benchmark for comparison. What are the other feasible alternatives? What are the advantages of adopting these particular models over others in this case? How will this affect the results? More details should be furnished.*

**Response:** Thank you for your careful review and suggestion. The several studies had shown that artificial neural networks (ANN) (Rasouli et al., 2012; Cheng et al., 2015; El-Shafie and Noureldin, 2011; Chau, 2006; Ali Ghorbani et al., 2018) and support vector regression (SVR) (Tongal and Booij, 2018; Luo et al., 2019; Moazenzadeh et al., 2018) are the two powerful models for inflow predicting. They are widely used and very mature algorithms, which are scientific and reasonable compared with them.

**Proposed changes to manuscript:** More details about compared model will be given in Section "Introduction" of the revised manuscript.

*7. It is mentioned in page 3 that the Xiaowan Hydropower Station is adopted as the case study. What are other feasible alternatives? What are the advantages of adopting this particular case study over others in this case? How will this affect the results? The authors should provide more details on this.*

**Response:** Thank you for your careful review and suggestion. The Xiaowan Hydropower Station in the lower reaches of the Lancang River, which is the longest river with most discard water in Yunnan Province, was chosen as the study site (as shown in Fig. 2). The Lancang River is approximately 2000 km long and has a drainage area of 113300 km$^2$ above the Xiaowan Hydropower Station. Thus, the Xiaowan Hydropower Station is the main control hydropower station in the Lancang River and it is very significant to adopte the Xiaowan Hydropower Station as the case study.

**Proposed changes to manuscript:** More details about case study will be given in Section 2.1 of the revised manuscript according to your suggestion.

*8. It is mentioned in page 4 that the maximum information coefficient is adopted to select inputs from 79 potential predictors from reanalysis data. What are the advantages of*

*adopting this particular approach over others in this case? How will this affect the results? The authors should provide more details on this.*

**Response:** Thank you for your careful review and suggestion. The maximal information coefficient (MIC) (Reshef et al., 2011) is a robust measure of the degree of correlation between two variables and has attracted a lot attention from academia (Zhao et al., 2013; Ge et al., 2016; Lyu et al., 2017; Sun et al., 2018), which can select effective input factors accurately and quickly.

**Proposed changes to manuscript:** More details about inputs selection will be given in Section 4.1 of the revised manuscript.

*9. It is mentioned in page 4 that autocorrelation function is adopted to identify observed inflow and rainfall lags. What are other feasible alternatives? What are the advantages of adopting this particular approach over others in this case? How will this affect the results? The authors should provide more details on this.*

**Response:** Thank you for your careful review and suggestion. The autocorrelation function (ACF) measures the dependency or relationship of observed value with lagged observations of a considered variable. In a long-dependent series such as inflow time series, the ACF will decay slowly. The partial autocorrelation function (PACF) and cross-correlation function (CCF) are two other feasible alternatives. We use the PACF and CCF in these days for modeling, calculation and analysis in these days according to Referee(#3)'s suggestion. We agree to try PACF and CCF replace ACF to determine the model structure.

**Proposed changes to manuscript:** In the revised manuscript, PACF and CCF will be adopted to determining the model structures for inflow and rainfall, respectively. Hypothesis test is used to determine the significant relationships replacing user-defined threshold value.

*10. It is mentioned in page 6 that four evaluation criteria are adopted to evaluate the performance of the models. What are the other feasible alternatives? What are the advantages of adopting these particular evaluation criteria over others in this case? How will this affect the results? More details should be furnished.*

**Response:** Thank you for your careful review and suggestion. The root mean square error (RMSE) and mean absolute error (MAE) are the most commonly used criteria to assess model

performance (Luo et al., 2019; Chau, 2005; Chau, 2006). The Nash-Sutcliffe efficiency coefficient (NSE) (Nash and Sutcliffe, 1970) is commonly for evaluating the performance of hydrological models and it is one of the best performance metrics for reflecting the overall fit of a hydrograph. The Pearson correlation coefficient (CORR) is a good measurement of the average error. Peak flow criterion, degree of agreement and Kling-Gupta efficiency metrics are three other feasible alternatives.

**Proposed changes to manuscript:** Peak flow criterion, degree of agreement and Kling-Gupta efficiency metrics will be added to compare several model performances in Section 3.3 of the revised manuscript.

*11. It is mentioned in page 7 that a grid search algorithm is adopted to optimization model parameters. What are other feasible alternatives? What are the advantages of adopting this particular algorithm over others in this case? How will this affect the results? The authors should provide more details on this.*

**Response:** Thank you for your careful review and suggestion. Grid search is considered as an effective parameter search method, which is widely used (Fienen et al., 2018). Two of other feasible alternatives are randomized search and Bayesian optimization. Bayesian optimization (Snoek et al., 2012) was employed to tune the hyperparameters of support vector regression (SVR) in this paper. We have performed some numerical experiments to compare grid search and randomized search and grid search can obtain more reasonable and stable hyperparameter combination.

**Proposed changes to manuscript:** More details about grid search will be given in Section 4.2 of the revised paper.

*12. It is mentioned in page 9 that grid searching is adopted to tune the hyperparameters of GBRT, GBRT-MIC, ANN-MIC. What are other feasible alternatives? What are the advantages of adopting this particular approach over others in this case? How will this affect the results? The authors should provide more details on this.*

**Response:** Thank you for your careful review and suggestion. Same as question 11, grid search is considered as an effective parameter search method, which is widely used (Fienen et al., 2018). We have performed some numerical experiments to compare grid search and

randomized search and grid search can obtain more reasonable and stable hyperparameter combination.

**Proposed changes to manuscript:** More details about grid search will be given in Section 4.2 of the revised paper.

*13. It is mentioned in page 9 that Bayesian optimization (Snoek et al., 2012) is adopted to tune the hyperparameters of SVR-MIC. What are other feasible alternatives? What are the advantages of adopting this particular approach over others in this case? How will this affect the results? The authors should provide more details on this.*

**Response:** Thank you for your careful review and suggestion. Bayesian optimization (Snoek et al., 2012) is proved as an effective parameter search method, especially for wide domain space.

**Proposed changes to manuscript:** More details about Bayesian optimization will be given in Section 4.2 of the revised paper.

*14. It is mentioned in page 9 that Python is adopted to perform all computations. What are other feasible alternatives? What are the advantages of adopting this particular software over others in this case? How will this affect the results? The authors should provide more details on this.*

**Response:** Thank you for your careful review and suggestion. Python is an important tool for scientific computing and data analysis. It has many open source libraries and is easy to implement.

**Proposed changes to manuscript:** Brief introduction about Python will be given in Section 4.1 of the revised manuscript.

*15. Some key parameters are not mentioned. The rationale on the choice of the particular set of parameters should be explained with more details. Have the authors experimented with other sets of values? What are the sensitivities of these parameters on the results?*

**Response:** Thank you for your careful review and suggestion. For ANN, A range of 2-20 neurons and four activation functions (Table 3) are selected by trail-and-error. The sensitivities of these parameters have been analyzed.

**Proposed changes to manuscript:** We will use more wide selected ranges of the model parameters in Section 4.2 of the revised manuscript.

*16. Some assumptions are stated in various sections. Justifications should be provided on these assumptions. Evaluation on how they will affect the results should be made.*

**Response:** Thank you for your careful review and suggestion. The comparison of different models is based on the basic assumption that parameters are optimal. In the original manuscript, grid searching and Bayesian optimization were employed to tune the hyperparameters of model. Large models for each lead time were developed to find as possible as optimal parameters.

**Proposed changes to manuscript:** More details about the assumption of optimal parameters will be given in Section 4.2 of the revised paper.

*17. The discussion section in the present form is relatively weak and should be strengthened with more details and justifications.*

**Response:** Thank you for your careful review and suggestion. Peak flow criterion, degree of agreement and Kling-Gupta efficiency metrics will be added to compare several model performances and more details about the discussion of the obtained results will be discussed.

**Proposed changes to manuscript:** The discussion of the obtained results will be enriched in Section 4 of the revised manuscript.

*18. Moreover, the manuscript could be substantially improved by relying and citing more on recent literatures about contemporary real-life case studies of soft computing techniques in hydrological prediction such as the followings: ïAn Yaseen, Z.M., et al., "An enhanced extreme learning machine model for river flow forecasting: state-of-the-art, practical applications in water resource engineering area and future research direction," Journal of Hydrology 569: 387-408 2019. ïAn Fotovatikhah, F., et al., "Survey of Computational Intelligence as ˘Basis to Big Flood Management: Challenges, research directions and Future Work," Engineering Applications of Computational Fluid Mechanics 12 (1): 411-437 2018. ïAn Mosavi, A., et al., "Flood Prediction Using Machine Learning Models: Literature Review," Water 10 (11): article no. 1536 2018. ïAn Moazenzadeh, R., et al., "Coupling a ˘firefly algorithm with support vector regression to predict evaporation in northern Iran," Engineering Applications of Computational Fluid Mechanics 12 (1): 584-597 2018. ïAn*

*Ghorbani, M.A., et al., "Forecasting pan evaporation with an integrated Artificial Neural Network Quantum-behaved Particle Swarm Optimization model: a case study in Talesh, Northern Iran," Engineering Applications of Computational Fluid Mechanics 12 (1): 724-737 2018. ïAn Chau, K.W., et al., "Use of Meta-Heuristic Techniques in Rainfall-Runoff Modelling" Water 9(3): article no. 186, 6p 2017.*

**Response:** Thanks.

**Proposed changes to manuscript:** We have carefully looked up the mentioned literature, which has been cited in the paper, and we have also extensively looked up other literatures from HESS, JH and other relative journals, added some necessary literatures.

*19. Some inconsistencies and minor errors that needed attention are: ïAn Replace "... was supply to depict…" with "…was supplied to depict ..." in line 86 of page 3 ïAn Replace "... into train set, validation set, and test set..." with "...into training set, validation set, and testing set..." in line 206-207 of page 7 ïAn Replace "... test set..." with "... testing set..." in line 209 of page 7 ïAn Replace "... more accuracy inflow forecasting..." with "... more accurate inflow forecasting ..." in line 283 of p10 ïAn Replace "... arisen in in areas..." with "... arisen in areas..." in line 288 of page 10 ïAn Replace "... for train, validation and test set ..." with "... for training, validation and testing set ..." in line 294 of page 10 ïAn Replace "... According to compare the forecasted results of …" with "... According to the comparison of forecasted results of ... " in line 330 of page 11*

**Response:** Thank you for your careful review. We agree with all of the minor changes above, and we will go through carefully the manuscript to check and correct any errors.

**Proposed changes to manuscript:** Those typos have been corrected in the revision.

*20. In the conclusion section, the limitations of this study, suggested improvements of this work and future directions should be highlighted.*

**Response:** Thank you for your careful review and suggestion.

**Proposed changes to manuscript:** We have carefully checked the conclusion of the article and will add the limitations of this study, improvements of this work and future directions in Section 5 of the revised manuscript.

---

## Author Comment (AC2) · 6 Feb 2020

*Reply to Anonymous Referee #2*

*I went through the manuscript. Generally, the manuscript has been well organized.*

**Response:** We thank you very much for reviewing the manuscript and giving the positive comment. The following are our point‑by‑point responses to your comments.

*1. P1, L6, In abstract the authors stated that "The impacts of climate change and human activities make accurate inflow prediction increasingly difficult, especially for longer lead times". As far as I know, the climate change deals with long term trends, say the climate variation over 20 years. I cannot understand relevance of the abovementioned with climate change impacts and human activities.*

**Response:** Thank you for your careful review. Climate variation affects the streamflow directly. For example, changing precipitation patterns and intensity, together with changing temperatures, will greatly modify the streamflow. Human activities such as land use change, water withdrawal, and hydraulic structures have substantial impacts on streamflow. We fully approve that climate variation and human activities can generate large effect to streamflow of medium and long term. For short term streamflow forecasting, climate variation and human activities also have some effect, so we need to calibration parameter according to meteorological factors.

**Proposed changes to manuscript:** N/A.

*2. The authors have to clearly indicate which model was developed for the inflow forecasting. At first they have to demonstrate if they used conceptual models or data driven models. What is the advantage of the developed model?*

**Response:** Thank you for your careful review. The gradient boosting regression trees is used to forecast daily streamflow. The model is a data driven model. GBRT has two other advantages. Firstly, GBRT can rank features according to their contribution to model scores, which is of great significance for reducing the complexity of the model. Secondly, GBRT is a white box model and can be easily interpreted.

**Proposed changes to manuscript:** The more details and advantages of model developed will be given in Section 3 of the revised manuscript according to your suggestion.

*3. The input selection for multi-day ahead forecasting should be discussed according to available literature. It is essential why the input structure of the longer period is not updated following literature the earlier stage forecasts.*

**Response:** Thank you for your careful review and suggestion. We have carefully reviewed more literatures about the input selection for multi-day ahead forecasting. There are mainly two strategies that you can use for multi-step forecasting, Static (Direct) multi-step forecast and Recursive multi-step forecast (Taieb et al., 2012). Recursive forecast strategy is biased when the underlying model is nonlinear and is sensitive to the estimation error, since estimated values, instead of actual ones, are more and more used when we get further in the future (Bontempi et al., 2012). Thus, the Static multi-step forecasting strategy is employed in this paper. Since the Static strategy does not use any approximated values to compute the forecasts, it is not prone to any accumulation of errors. The model structure of one-step and two-step forecasting of Static strategy is listed below (as shown in Section 3.4) which has different model parameters.

$$prediction(t + 1) = model1(obs(t - 1), obs(t - 2), ..., obs(t - n))$$
$$prediction(t + 2) = model2(obs(t - 1), obs(t - 2), ..., obs(t - n))$$

where $obs(t - 1)$ is the observation value at the $t - 1$ period and $prediction(t + 1)$ is the predicted value of one-step at the $t$ period.

**Proposed changes to manuscript:** More details about multi-step forecasting will be shown in Section 3.4 of the revised manuscript.

*4. Literature should be updated discussing on more papers addressing multi-step ahead forecasting.*

**Response:** Thank you for your careful review. We have carefully reviewed more literatures from HESS, JH and relative journals about multi-step ahead forecasting.

**Proposed changes to manuscript:** We will review multi-step ahead forecasting in Section "Introduction" of the revised manuscript and the more references about multi-step ahead forecasting will be updated in Section "Reference" of the revised manuscript.

*5. The authors employed gradient boosting regression trees as an ensemble framework. More explanations required about ensemble members.*

**Response:** Thank you for your careful review and suggestion. The ensemble member is the decision tree model.

**Proposed changes to manuscript:** More details about ensemble members will be given in the revised manuscript.

*6. Uncertainty analysis should be carried out to show how much the predictions are confident. As the lead time increases, the metrics reveal errors are increasing drastically. Moreover, high uncertainties are expected to associate with such models. Please discuss this issues accordingly.*

**Response:** Thank you for your careful review. We agree that uncertainty analysis in predictions is significant. As far as we know, medium and long-term forecasting is more uncertainty, for example, monthly or yearly. This paper focuses on improving prediction accuracy by developing new model and importing ERA-Interim reanalysis data and aims to providing reference for reducing discard water. The uncertainty analysis of medium and long-term inflow forecasting will be further studied in the next study.

**Proposed changes to manuscript:** N/A.

*7. Concerning inflow predictions, please indicate efficiency of the proposed model to simulate and predict extreme values which are of great importance.*

**Response:** Thank you for your careful review and suggestion.

**Proposed changes to manuscript:** We introduce peak flow criterion to evaluate the performance of catching extreme values for developed model and more details will be given in Section 3.4 of the revised manuscript.

---

## Author Comment (AC3) · 6 Feb 2020

*Reply to Anonymous Referee #3*

*In this manuscript, the authors compared several data-driven models for multi-step forecasting of inflow. The employed models include gradient boosting regression trees (GBRT), artificial neural networks (ANN), support vector regression (SVR), and multiple linear regression (MLR) models. The models were developed by considering (1) streamflow and rainfall record, and (2) ERA-Interim reanalysis data. Further, the maximum information coefficient and autocorrelation functions were utilized to construct the input structures of the models. The authors concluded that the developed methodology that considers ERA-Interim reanalysis data considerably gives better results in the forecasting of inflows at lead times of 5-10 days. The manuscript is well written and organized. However, there is not a significant novelty in the manuscript except using ERA-Interim dataset. Further, there are severe weaknesses in the developing of the model input structures.*

**Response:** Thank you very much for your time and for your thoughtful and constructive review, and also thank you for giving some positive comments. This paper focuses on improving prediction accuracy by developing new model and importing ERA-Interim reanalysis data and aims to providing reference for reducing discard water. The following are our point-by-point responses to your comments.

**Proposed changes to manuscript**: N/A.

*1. The authors made a significant mistake in using the autocorrelation function (ACF) in determining the model structures. They should have employed cross-correlation and partial autocorrelation functions (or other measures) to establish the relationship between the observed records and inflow. The ACF only measures the dependency or relationship of observed value with lagged observations of a considered variable. In a long-dependent series such as inflow time series, the ACF will decay slowly. Therefore, statistically significant relationships between the observed and lagged values could not be determined. To determine the significant relationships, the authors employed user-defined threshold value. The obtained inflow and rainfall values for the input structures of the models include only three lagged-day values as could be expected. This number could be higher based on the selected threshold. However, this finding does not convey any meaningful relationship between the observed records (i.e. inflow and rainfall) and the inflow values. The PACF should have been used for determining the lagged relationships of inflows since the inflow time series mainly shows the long-memory feature where the correlation decays after a long observation period. Further,*

*all statistically significant lagged variables should have been included in the model structures found in PACF. Using a user-defined threshold value is a serious mistake in this situation.*

**Response:** Thank you for your careful review and nice comments. According to your suggestions, we use the partial autocorrelation function (PACF) and cross correlation function (CCF) in these days for modeling, calculation and analysis, and find that your suggestions are effective. We agree to replace the autocorrelation function (ACF) to determine the model structure and confidence interval obtained by hypothesis test is used to replace user-defined threshold value to determine the significant relationships. The all calculation results will be updated accordingly.

**Proposed changes to manuscript**: In Section 4.1 of the revised manuscript, PACF and CCF to determining the model structures for inflow and rainfall, respectively. Hypothesis test is used to determine the significant relationships replacing user-defined threshold value. The all calculation results will be updated accordingly in the revised manuscript.

*2. The authors claimed that the proposed methodology "significantly" improves the accuracy of inflow prediction for longer lead times. However, I do not agree with this comment. Because, as the authors mentioned, there is only about 1% and 5% improvement in two-day and 10-day ahead forecasting. Therefore, the results do not seem convincing about the superiority of ERA-Interim dataset over the common dataset, especially ill-conditioned input structures with conventional observed inflow and rainfall dataset.*

**Response:** Thank you for your careful review. There is about 1% and 5% improvement in two-day and ten-day ahead forecasting according to NSE and there is about 2.3% and 10.7% improvement in two-day and ten-day ahead forecasting according to MAE. Considering the inflow forecasting is good for reducing discard water, we think the improvement also is very valuable.

**Proposed changes to manuscript:** Revised input structures are used to compare with developed model with ERA-Interim dataset. More discussion about results of models will be given in Section 4.4 of the revised manuscript.

*3. The authors found that three-day lagged values of inflow and rainfall have less impact on 10-day ahead forecasting of inflow in Section 4.5. This is a clue that more lagged values of input variables should have been included in the models' structure.*

**Response:** Thank you for your careful review and suggestion. According to your suggestion, PACF and CCF are used to determining the model structures for inflow and rainfall, respectively, in these days. Numerical experiment results indicate that more lagged values of input variables will be included in the model's structure.

**Proposed changes to manuscript:** PACF and CCF will be used to determining the model structures for inflow and rainfall, respectively. More lagged values of input variables will be included in the model's structure in the revised manuscript.

*4. The employed performance indices, specifically the coefficient of determination, seems insufficient to compare several model performances. More distinctive performance indices such as degree of agreement and Kling-Gupta efficiency metrics should have been used.*

**Response:** Thanks. The Nash-Sutcliffe efficiency coefficient (NSE) (Nash and Sutcliffe, 1970) is commonly for evaluating the performance of hydrological models and it is one of the best performance metrics for reflecting the overall fit of a hydrograph. The Pearson correlation coefficient (CORR) is a good measurement of the average error.

**Proposed changes to manuscript:** Peak flow criterion, degree of agreement and Kling-Gupta efficiency metrics will be added to compare several model performances in Section 3.3 of the revised manuscript.

*5. It is not clear how the multi-step forecasting scheme (i.e., recursive or static) was employed? Please give more details about this issue.*

**Response:** Thank you for your careful review and suggestion. Recursive forecast strategy is biased when the underlying model is nonlinear and is sensitive to the estimation error, since estimated values, instead of actual ones, are more and more used when we get further in the future (Bontempi et al., 2012). Thus, the Static multi-step forecasting strategy was employed and the models of different lead times have different model parameters. The model structure of one-step and two-step forecasting of Static strategy is listed below which has different model parameters.

$$prediction(t + 1) = model1\big(obs(t - 1), obs(t - 2), \ldots, obs(t - n)\big)$$

$$prediction(t + 2) = model2(obs(t - 1), obs(t - 2), \ldots, obs(t - n))$$

where $obs(t - 1)$ is the observation value at the $t - 1$ period and $prediction(t + 1)$ is the predicted value of one-step at the $t$ period.

**Proposed changes to manuscript:** More details about multi-step forecasting will be shown in Section 3.4 of the revised manuscript.

*6. The selected ranges of the model parameters seem highly subjective. Please justify the selected ranges of the model parameters, especially in Section 4.2.*

**Response:** Thank you for your careful review and suggestion. Specifying the selected ranges of the model parameters is the trickiest part of hyperparameter optimization. For gradient boosting regression trees model, we refer to (Fienen et al., 2018; Friedman, 2001; Pedregosa et al., 2011) to inform our choices of hyperparameter distributions by placing greater probability where we think the best values are. It can be difficult to figure out the interaction between hyperparameters. In cases where we aren't sure about the best values, and let the Bayesian algorithm do the reasoning for us.

**Proposed changes to manuscript:** We can use wide selected ranges of the model parameters have been justified in Section 4.2 of the revised manuscript.

*7. The range for the number of hidden neurons (i.e. 2–20) seems too high. Please justify this from a hydrological perspective. Because using a high number of hidden neurons could lead to overfitting that resulted in a poor performance in multi-step forecasting.*

**Response:** Thank you. Specifying the number of hidden neurons is a difficult task (Badrzadeh et al., 2013) and the number of hidden neurons is determined by trial and error procedure in the original paper. In cases where we aren't sure about the best number of hidden neurons, we can use wide ranges and let the trial and error procedure do the reasoning for us. It is found that the optimal number of neurons is 2 or 3.

**Proposed changes to manuscript:** The number of hidden neurons will be justified in Section 4.2 of the revised manuscript.

*8. The discussion of the obtained results should be improved with more details, especially giving necessary citations to previous studies.*

**Response:** Thank you for your careful review and suggestion. Peak flow criterion, degree of agreement and Kling-Gupta efficiency metrics will be added to compare several model performances and more details about the discussion of the obtained results will be discussed.

**Proposed changes to manuscript:** The discussion of the obtained results will be enriched and some necessary citations to previous studies will be discussed.

*9. It is not clear how Fig. 1 was obtained. Please give the necessary information about this figure.*

**Response:** Thank you for your careful review. We cooperate with production unit for a long time, and the data of Fig. 1 from production unit has been obtained from public website.

**Proposed changes to manuscript:** We give the source link of the data in the revised manuscript.

*10. Please give more details on the Lines 78–82.*

**Response:** Thanks.

**Proposed changes to manuscript:** More details about ERA-Interim dataset have been introduced in the revised manuscript.

*11. Please give the definitions and meanings of the variables in the ERA-Interim dataset in the Appendix.*

**Response:** Thank you for your suggestion.

**Proposed changes to manuscript:** The definitions and meanings of the variables in the ERA-Interim dataset will be given in Section "Appendix" of the revised manuscript.

*12. Please justify using the feature scaling in Line 108.*

**Response:** Thank you for your suggestion.

**Proposed changes to manuscript:** The "data scaling" has been replaced by "feature scaling".

*13. What do you mean with "invalid variables" in Line 116?*

**Response:** Thanks. The "invalid variables" in Line 116 mainly demonstrate the weak-correlated variables which has a weak correlation and cannot interpret inflow very well.

**Proposed changes to manuscript:** The "invalid variables" will be modified to "weak-correlated variables" in the revised manuscript.

*14. Please prefer "maximal" or "maximum" information criterion throughout the manuscript.*

**Response:** Thank you for your careful review.

**Proposed changes to manuscript:** All "maximum" information criterion in the original manuscript will be modified to "maximal" information criterion in the revised manuscript.

*15. Please check the term MI\*(D,X,Y) in Eq. (5) since you defined MI\*(D,x,y) in Line 130.*

**Response:** Thank you for your careful review.

**Proposed changes to manuscript:** The term MI\*(D,X,Y) has been modified to MI\*(D,x,y).

*16. The definition of B(n) was given in Line 133; however it is not clear where this parameter is used.*

**Response:** Thank you for your careful review. $B(n)$ is the maximal grid size which is a function of sample size and we usually set $B = n^{0.6}$.

**Proposed changes to manuscript:** Some details about B(n) will be added in the revised manuscript.

*17. Please check the terms in Eq. (7). Will they be R1(i,s) or R1(j,s)?*

**Response:** Thank you for your careful review.

**Proposed changes to manuscript:** R1(i,s) and R2(i,s) in Eq. (7) will be modified to R1(j,s) and R2(j,s) in the revised manuscript.

*18. Please check the notations in Line 144; n features with N samples or n samples with N features according to the given definition.*

**Response:** Thanks. The notations in Line 144 shows n features with N samples.

**Proposed changes to manuscript:** N/A

*19. There is little information about the structure of ERA-Interim dataset. Please give more details about this dataset.*

**Response:** Thank you for your careful review. There are detailed introductions for ERA-Interim dataset in the https://www.ecmwf.int/en/forecasts/datasets/reanalysis-datasets/era-interim. According to your suggestion, more detailed information about variables of ERA-Interim dataset will be added in the revised manuscript.

**Proposed changes to manuscript:** More details about ERA-Interim dataset will be given in Section "Appendix" of the revised manuscript.

*20. There is not any information about grid searching methodology.*

**Response:** Thank you for your careful review and suggestion. Grid search is considered as an effective parameter search method, which is widely used (Fienen et al., 2018).

**Proposed changes to manuscript:** The grid searching methodology will be introduced in detail in Section 4.2 of the revised manuscript.

*21. Please add "activation function" after "relu" in Line 248.*

**Response:** Thank you for your suggestion.

**Proposed changes to manuscript:** The "activation function" has been added in the revised manuscript.

*22. The comments in Lines 278–280 are vague.*

**Response:** Thank you for your careful review. The comments in Lines 278–280 indicate the relationship between performance indices and lead times in the test set (2017-2018). We mainly discuss the trend of performance indices as the lead time increases.

**Proposed changes to manuscript:** The comments about the relationship between performance indices and lead times will be given more details in Section 4.3 and 4.4.

*23. The authors did not discuss the reasons why NSE values for lead times of 6-7-8-9-day is worse than the value of lead time of 10-day.*

**Response:** Thank you for your careful review. It should be noted that NSE values for lead times of 6-7-8-9-day is worse than the value of lead time of 10-day in the train set and validation set. We consider the possible reasons are parameter optimization and model structure. We have done a lot of numerical experiments in these days and the question will be discussed in Section 4 of the revised manuscript.

**Proposed changes to manuscript:** More discussion about why NSE values for lead times of 6-7-8-9-day is worse than the value of lead time of 10-day in the train set and validation set will be added in Section 4 of the revised manuscript.

*24. It is not clear how top k features were selected according to the chosen threshold value. Did the authors employ several threshold values? Please give more details on this issue.*

**Response:** Thank you for your careful review and suggestion. The original manuscript totally employs three threshold values. Two of these thresholds were used to determine the model input structures with inflow and rainfall. Another threshold value was used to determine the model input structures with ERA-Interim dataset. Further, we perform input selection in two steps via the maximal information coefficient (MIC). First, compute MIC value of each reanalysis factors and observed inflow. Then, sort features based on MIC in a descending order. And then, select the top k features whose mic are greater than or equal to the set threshold. The selected k features from reanalysis data are used as part of input to the model.

**Proposed changes to manuscript:** Consider the subjectivity of user-defined thresholds, the three threshold values will be modified by significance test and trail-and-error to determine input structures of model in Section 3.1 of the revised manuscript.

---

## Author Comment (AC4) · 16 Feb 2020

*Reply to Anonymous Referee #1*

Thank you very much for your time and for your thoughtful and constructive review. The following are our supplementary reply for a lot of research has been done to your comments.

*8. It is mentioned in page 4 that the maximum information coefficient is adopted to select inputs from 79 potential predictors from reanalysis data. What are the advantages of adopting this particular approach over others in this case? How will this affect the results? The authors should provide more details on this.*

**Response:** Thank you for your careful review and suggestion. The maximal information coefficient (MIC) (Reshef et al., 2011) is a robust measure of the degree of correlation between two variables and has attracted a lot attention from academia (Zhao et al., 2013; Ge et al., 2016; Lyu et al., 2017; Sun et al., 2018), which can select effective input factors accurately and quickly. According to your suggestion, we adjusted the selection procedure of reanalysis variables. We perform feature selection in two steps via MIC. First, compute MIC value of each reanalysis factors and observed inflow. Then, sort features based on MIC in a descending order and determine the optimum inputs using trail-and-error method, i.e. starting from the top one feature and then modifying the external input feature by successively adding one more feature into model input (Moosavi et al., 2013; Shoaib et al., 2015). Finally, the top 14 reanalysis variables are selected as the input (Table 1). More details will be given in the revised version.

Table 1: Selected reanalysis variables

| No. | Variable | Index | Unit | MIC |
|-----|----------|-------|------|-----|
| 1 | Forecast albedo | $fal\_t$ | - | 0.865 |
| 2 | Soil temperature level 3 | $stl3_t$ | $K$ | 0.846 |
| 3 | 2 meter dewpoint temperature | $d2m_t$ | $K$ | 0.781 |
| 4 | Total column water vapour | $tcwv_t$ | $kg \cdot m^{-2}$ | 0.699 |
| 5 | Total column water | $tcw_t$ | $kg \cdot m^{-2}$ | 0.699 |
| 6 | Soil temperature level 2 | $stl2_t$ | $K$ | 0.689 |
| 7 | Minimum temperature at 2 meters | $mn2t_t$ | $K$ | 0.683 |
| 8 | Surface thermal radiation downwards | $strd_t$ | $J \cdot m^{-2}$ | 0.669 |
| 9 | Temperature of snow layer | $tsn_t$ | $K$ | 0.664 |
| 10 | Soil temperature level 4 | $stl4_t$ | $K$ | 0.642 |

| 11 | Soil temperature level 1 | $stl1_t$ | $K$ | 0.631 |
|---|---|---|---|---|
| 12 | Surface net thermal radiation, clear sky | $strc_t$ | $J \cdot m^{-2}$ | 0.620 |
| 13 | Runoff | $ro_t$ | $m$ | 0.619 |
| 14 | Volumetric soil water layer 1 | $swvl1_t$ | $m^3 \cdot m^{-3}$ | 0.614 |

*9. It is mentioned in page 4 that autocorrelation function is adopted to identify observed inflow and rainfall lags. What are other feasible alternatives? What are the advantages of adopting this particular approach over others in this case? How will this affect the results? The authors should provide more details on this.*

**Response:** Thank you for your careful review and suggestion. We use the partial autocorrelation function (PACF) and cross-correlation function (CCF) in these days for modeling, calculation and analysis in these days according to Referee (#3)'s suggestion. Figure 1 shows the PACF, CCF and the corresponding 95% confidence bands from lag 1 to lag 10.

[Figure]

Figure 1: PACF of Xiaowan daily inflow and CCF of rainfall (2011-2014).

The PACF show significant autocorrelation at lag one and lag four, respectively. Therefore, one-day and four-day lag can be selected as input of the model. According to CCF of Xiaowan daily inflow and rainfall, ten-day lag all are significant. are selected as the input. And thus, trail-and-error method is used to determine the optimum inputs. The following five inputs are used as the model input successively.

1. $Q_{t-1}, Q_{t-4}$

2. $Q_{t-1}, Q_{t-4}, R_{t-1}$

3. $Q_{t-1}, Q_{t-4}, R_{t-1}, R_{t-2}$

4. $Q_{t-1}, Q_{t-4}, R_{t-1}, R_{t-2}, R_{t-3}$

5. $Q_{t-1}, Q_{t-4}, R_{t-1}, R_{t-2}, R_{t-3}, R_{t-4}$

Finally, the fourth input is selected as the model input. More details will be given in the revised version.

*10. It is mentioned in page 6 that four evaluation criteria are adopted to evaluate the performance of the models. What are the other feasible alternatives? What are the advantages of adopting these particular evaluation criteria over others in this case? How will this affect the results? More details should be furnished.*

**Response:** Thank you for your careful review and suggestion. The root mean square error (RMSE) and mean absolute error (MAE) are the most commonly used criteria to assess model performance (Luo et al., 2019; Chau, 2005; Chau, 2006). The Pearson correlation coefficient (CORR) is a good measurement of the average error. According to Referee (#2) and (#3)'s suggestions, Kling-Gupta efficiency metrics (KGE), the percent bias in flow duration curve high-segment volume (BHV) and the Index of Agreement (IA) are introduced as supplements. Kling–Gupta efficiency scores (KGE) (Knoben et al., 2019) is also a widely used evaluation index. It can be provided as following Eq. (1) and (2).

$$KGE = 1 - \sqrt{(CORR - 1)^2 + \left(\frac{\hat{\sigma}}{\sigma} - 1\right)^2 + \left(\frac{\bar{\hat{Q}}}{\bar{Q}} - 1\right)^2} \qquad (1)$$

$$CORR = \frac{\sum_{i=1}^{n}(Q_i - \bar{Q})\left(\widehat{Q_i} - \bar{\hat{Q}}\right)}{\sqrt{\sum_{i=1}^{n}(Q_i - \bar{Q})^2}\sqrt{\sum_{i=1}^{n}\left(\widehat{Q_i} - \bar{\hat{Q}}\right)^2}}, \hat{\sigma} = \sqrt{\frac{1}{n}\sum_{i=1}^{n}\left(\widehat{Q_i} - \bar{\hat{Q}}\right)^2}, \sigma = \sqrt{\frac{1}{n}\sum_{i=1}^{n}(Q_i - \bar{Q})^2} \qquad (2)$$

where $\widehat{Q_i}$ and $Q_i$ are the inflow estimation and observed value at time *i*, respectively and *n* is the number of samples. $\bar{\hat{Q}}$ is the mean of the estimation values. $\sigma$ is the standard deviation of the observed values, $\hat{\sigma}$ is the standard deviation of the inflow estimation.

The percent bias in flow duration curve high-segment volume (BHV) (Yilmaz et al., 2008; Vogel and Fennessey, 1994) is used to evaluate performance of peak inflow forecasting. It can be provided as following Eq. (3).

$$BHV = \frac{\sum_{h=1}^{H}(\hat{Q}_h - Q_h)}{\sum_{h=1}^{H} Q_h} \times 100 \qquad (3)$$

where h = 1, 2,. . .H are the flow indices for flows with exceedance probabilities lower than 0.02.

The Index of Agreement (IA) (Willmott, 1981) plays a significant role in evaluating the degree of the agreement between observed values and inflow estimation. It is given by Eq. (4).

$$IA = 1 - \frac{\sum_{i=1}^{n}(\hat{Q}_i - Q_i)^2}{\sum_{i=1}^{n}(|\hat{Q}_i - \bar{Q}| + |Q_i - \bar{Q}|)^2} \qquad (4)$$

More details will be given in the revised version.

*12. It is mentioned in page 9 that grid searching is adopted to tune the hyperparameters of GBRT, GBRT-MIC, ANN-MIC. What are other feasible alternatives? What are the advantages of adopting this particular approach over others in this case? How will this affect the results? The authors should provide more details on this.*

**Response:** Thank you for your careful review and suggestion. The grid search is considered as an effective parameter search method, which is widely used (Fienen et al., 2018). We have performed some numerical experiments to compare grid search and randomized search and grid search can obtain more reasonable and stable hyperparameter combination. In addition, more wide range of parameters has been performed according to Referee (#3)'s suggestion. For artificial neural networks-maximal information coefficient (ANN-MIC), a range of 2-20 neurons and four activation functions are selected by grid searching. Table 2 shows results of parameter optimization of ANN-MIC. Table 3 and Table 4 show results of parameter optimization of support vector regression-maximal information coefficient (SVR-MIC) and gradient boosting regression trees-maximal information coefficient (GBRT-MIC). More details will be given in the revised version.

Table 2: Tuning parameters of ANN-MIC

| Model | Tuning parameter | 1 | 2 | 3 | 4 | 5 | 6 | 7 | 8 | 9 | 10 |
|---|---|---|---|---|---|---|---|---|---|---|---|
| ANN-MIC | Structure | 19-5-1 | 19-2-1 | 19-3-1 | 19-2-1 | 19-2-1 | 19-2-1 | 19-2-1 | 19-2-1 | 19-2-1 | 19-2-1 |
| | Activate function | tanh | logistic | tanh | logistic | logistic | logistic | logistic | logistic | logistic | tanh |

Table 3: Tuning parameters of SVR-MIC.

| Model | Tuning parameter | Tuning range | 1 | 2 | 3 | 4 | 5 | 6 | 7 | 8 | 9 | 10 |
|---|---|---|---|---|---|---|---|---|---|---|---|---|
| SVR-MIC | C | **(1, 100, 20)** | 8.7368 | 50.0000 | 29.3684 | 19.0526 | 21.6316 | 6.1579 | 16.4737 | 6.1579 | 8.7368 | 3.5789 |
| | epsilon | **(0.001, 0.1, 20)** | 0.00048 | 0.00953 | 0.00032 | 0.00011 | 0.00001 | 0.00001 | 0.00032 | 0.00022 | 0.00058 | 0.00001 |
| | gamma | **(0.0001, 0.01, 20)** | 0.0100 | 0.0095 | 0.0019 | 0.0072 | 0.0019 | 0.0067 | 0.0038 | 0.0057 | 0.0072 | 0.0067 |

*Note*: The bold parts, (min, max, step) represent $[min + \frac{max-min}{step-1} \times 0, min + \frac{max-min}{step-1} \times 1, ..., min + \frac{max-min}{step-1} \times (step-1)]$.

Table 4: Tuning parameters of GBRT-MIC

| Tuning parameter | Tuning range | Optimal parameters (the lead times of 1-10 days) | |
|---|---|---|---|
| | | GBRT | GBRT-MIC |
| max_leaf_nodes | [2, 3, …, 20] | 7,3,3,3,3,2,3,3,3,3 | 10,11,16,11,12,10,6,6,4,4 |
| min_samples_leaf | [1, 2, …, 20] | 2,3,1,1,10,1,2,2,4,1 | 5,9,1,5,6,9,4,6,6,10 |
| max_depth | [1, 2, ..., 20] | 3,2,2,4,2,1,2,2,2,10 | 4,4,8,7,10,6,6,4,5,3 |
| min_samples_split | [2, 3, …, 20] | 9,14,13,20,11,3,6,2,3,4 | 16,20,19,16,20,20,17,17,20,3 |
| n_estimators | [500,550, …, 4000] | 1000,1000,1000,1500,1500,2500,1500,3500,2500,2500 | 2500,1000,1500,2500,1500,2500,1000,1000,1500,300 |
| learning_rate | [0.001,0.0025,0.005,0.0075,0.01,0.025,0.05,0.075,0.1] | 0.01,0.01,0.01,0.005,0.005,0.005,0.005,0.0025,0.0025,0.0025 | 0.0075,0.01,0.01,0.0025,0.01,0.0025,0.01,0.01,0.01,0.01 |

*13. It is mentioned in page 9 that Bayesian optimization (Snoek et al., 2012) is adopted to tune the hyperparameters of SVR-MIC. What are other feasible alternatives? What are the advantages of adopting this particular approach over others in this case? How will this affect the results? The authors should provide more details on this.*

**Response:** Thank you for your careful review and suggestion. Bayesian optimization (Snoek et al., 2012) is proved as an effective parameter search method, especially for wide domain space. According to your suggestion, grid search has been used to replace Bayesian optimization for optimize parameters of SVR-MIC. The grid search spends much more time for hyperparameter optimization but and the result of optimization are more stable. And thus, grid search has been used for optimize parameters of SVR-MIC. In addition, for time consuming of grid search, will it is our next research direction to use heuristic algorithms, such as particle swarm optimization, genetic algorithm and gray wolf algorithm to optimize model parameters. More details will be given in the revised version.

---

## Author Comment (AC5) · 16 Feb 2020

We thank you very much for reviewing the manuscript. The following are our supplementary reply for a lot of research has been done to your comments.

*7. Concerning inflow predictions, please indicate efficiency of the proposed model to simulate and predict extreme values which are of great importance.*

**Response:** Thank you for your careful review and suggestion. The root mean square error (RMSE) and mean absolute error (MAE) are the most commonly used criteria to assess model performance (Luo et al., 2019; Chau, 2005; Chau, 2006). The Pearson correlation coefficient (CORR) is a good measurement of the average error. According to Referee (#2) and (#3)'s suggestions, Kling-Gupta efficiency metrics (KGE), the percent bias in flow duration curve high-segment volume (BHV) and the Index of Agreement (IA) are introduced as supplements. Kling–Gupta efficiency scores (KGE) (Knoben et al., 2019) is also a widely used evaluation index. It can be provided as following Eq. (1) and (2).

$$KGE = 1 - \sqrt{(CORR - 1)^2 + \left(\frac{\hat{\sigma}}{\sigma} - 1\right)^2 + \left(\frac{\bar{\hat{Q}}}{\bar{Q}} - 1\right)^2} \tag{1}$$

$$\text{CORR} = \frac{\sum_{i=1}^{n}(Q_i - \bar{Q})(\hat{Q}_i - \bar{\hat{Q}})}{\sqrt{\sum_{i=1}^{n}(Q_i - \bar{Q})^2}\sqrt{\sum_{i=1}^{n}(\hat{Q}_i - \bar{\hat{Q}})^2}}, \hat{\sigma} = \sqrt{\frac{1}{n}\sum_{i=1}^{n}\left(\hat{Q}_i - \bar{\hat{Q}}\right)^2}, \sigma = \sqrt{\frac{1}{n}\sum_{i=1}^{n}(Q_i - \bar{Q})^2} \tag{2}$$

where $\hat{Q}_i$ and $Q_i$ are the inflow estimation and observed value at time *i*, respectively and *n* is the number of samples. $\bar{\hat{Q}}$ is the mean of the estimation values. $\sigma$ is the standard deviation of the observed values, $\hat{\sigma}$ is the standard deviation of the inflow estimation.

The percent bias in flow duration curve high-segment volume (BHV) (Yilmaz et al., 2008; Vogel and Fennessey, 1994) is used to evaluate performance of peak inflow forecasting. It can be provided as following Eq. (3).

$$\text{BHV} = \frac{\sum_{h=1}^{H}(\hat{Q}_h - Q_h)}{\sum_{h=1}^{H} Q_h} \times 100 \tag{3}$$

where h = 1, 2,. . .H are the flow indices for flows with exceedance probabilities lower than 0.02.

The Index of Agreement (IA) (Willmott, 1981) plays a significant role in evaluating the degree of the agreement between observed values and inflow estimation. It is given by Eq. (4).

$$IA = 1 - \frac{\sum_{i=1}^{n}(\widehat{Q_i} - Q_i)^2}{\sum_{i=1}^{n}(|\widehat{Q}_i - \bar{Q}| + |Q_i - \bar{Q}|)^2} \tag{4}$$

More details will be given in the revised version.

---

## Author Comment (AC6) · 16 Feb 2020

*Reply to Anonymous Referee #3*

We thank you very much for reviewing the manuscript. The following are our supplementary reply for a lot of research has been done to your comments.

*In this manuscript, the authors compared several data-driven models for multi-step forecasting of inflow. The employed models include gradient boosting regression trees (GBRT), artificial neural networks (ANN), support vector regression (SVR), and multiple linear regression (MLR) models. The models were developed by considering (1) streamflow and rainfall record, and (2) ERA-Interim reanalysis data. Further, the maximum information coefficient and autocorrelation functions were utilized to construct the input structures of the models. The authors concluded that the developed methodology that considers ERA-Interim reanalysis data considerably gives better results in the forecasting of inflows at lead times of 5-10 days. The manuscript is well written and organized. However, there is not a significant novelty in the manuscript except using ERA-Interim dataset. Further, there are severe weaknesses in the developing of the model input structures.*

**Response:** Thank you very much for your time and for your thoughtful and constructive review, and also thank you for giving some positive comments. This paper focuses on improving prediction accuracy by three significant measures. Firstly, ERA-Interim reanalysis data are introduced to provide enough information for the model to discover inflow for longer lead times. Secondly, gradient boosting regression trees (GBRT) is adopted to implement inflow forecasting and GBRT has been used to achieve multi-step inflow forecasting. Thirdly, most widely used models are developed to compare with GBRT for multi-step inflow forecasting which demonstrates that developed model improves inflow forecasting accuracy. In order to make it easier for the author to grasp the innovation of this paper, we will modify the "Abstract" and "Introduction" carefully to make the innovation more prominent. More details will be given in the revised version.

*1. The authors made a significant mistake in using the autocorrelation function (ACF) in determining the model structures. They should have employed cross-correlation and partial autocorrelation functions (or other measures) to establish the relationship between the observed records and inflow. The ACF only measures the dependency or relationship of observed value with lagged observations of a considered variable. In a long-dependent series such as inflow time series, the ACF will decay slowly. Therefore, statistically significant relationships between the observed and lagged values could not be determined. To determine*

*the significant relationships, the authors employed user-defined threshold value. The obtained inflow and rainfall values for the input structures of the models include only three lagged-day values as could be expected. This number could be higher based on the selected threshold. However, this finding does not convey any meaningful relationship between the observed records (i.e. inflow and rainfall) and the inflow values. The PACF should have been used for determining the lagged relationships of inflows since the inflow time series mainly shows the long-memory feature where the correlation decays after a long observation period. Further, all statistically significant lagged variables should have been included in the model structures found in PACF. Using a user-defined threshold value is a serious mistake in this situation.*

**Response:** Thank you for your careful review and nice comments. We use the partial autocorrelation function (PACF) and cross-correlation function (CCF) in these days for modeling, calculation and analysis in these days according to your suggestion. Figure 1 shows the PACF, CCF and the corresponding 95% confidence bands from lag 1 to lag 10.

[Figure]

Figure 1: PACF of Xiaowan daily inflow and CCF of rainfall (2011-2014).

The PACF show significant autocorrelation at lag one and lag four, respectively. Therefore, one-day and four-day lag can be selected as input of the model. According to CCF of Xiaowan daily inflow and rainfall, ten-day lag all are significant. are selected as the input. And thus, trail-and-error method is used to determine the optimum inputs. The following five inputs are used as the model input successively.

1. $Q_{t-1}, Q_{t-4}$

2. $Q_{t-1}, Q_{t-4}, R_{t-1}$

3. $Q_{t-1}, Q_{t-4}, R_{t-1}, R_{t-2}$

4. $Q_{t-1}, Q_{t-4}, R_{t-1}, R_{t-2}, R_{t-3}$

5. $Q_{t-1}, Q_{t-4}, R_{t-1}, R_{t-2}, R_{t-3}, R_{t-4}$

Finally, the fourth input is selected as the model input. More details will be given in the revised version.

*2. The authors claimed that the proposed methodology "significantly" improves the accuracy of inflow prediction for longer lead times. However, I do not agree with this comment. Because, as the authors mentioned, there is only about 1% and 5% improvement in two-day and 10-day ahead forecasting. Therefore, the results do not seem convincing about the superiority of ERA-Interim dataset over the common dataset, especially ill-conditioned input structures with conventional observed inflow and rainfall dataset.*

**Response:** Thank you for your careful review. Revised input structures are used to compare with developed model with ERA-Interim dataset. Table 1 shows performance indices of model in the test set. The experimental results indicate that the developed method generally performs better than other models and improves the accuracy of inflow forecasting about 1% and 14% in two-day and 10-day ahead forecasting. Especially for 5-10 day lead times, GBRT-MIC could be used for more accurate and reliable inflow forecasting. More details will be given in the revised version.

Table 1: Performance indices of the test set.

| Indice | Model | 1 | 2 | 3 | 4 | 5 | 6 | 7 | 8 | 9 | 10 |
|--------|-------|---|---|---|---|---|---|---|---|---|----|
| MAE | GBRT-MIC | 141 | **158** | **174** | **181** | **188** | **191** | **194** | **201** | **208** | **210** |
| | SVR-MIC | 132 | 161 | 183 | 191 | 212 | 219 | 226 | 231 | 235 | 239 |
| | ANN-MIC | **132** | 163 | 184 | 197 | 213 | 224 | 228 | 233 | 240 | 244 |
| | MLR-MIC | 136 | 170 | 191 | 209 | 226 | 240 | 243 | 249 | 254 | 258 |
| RMSE | GBRT-MIC | 219 | **245** | **271** | **284** | **297** | **304** | **311** | **321** | **332** | **335** |
| | SVR-MIC | 200 | 256 | 305 | 332 | 362 | 375 | 392 | 400 | 409 | 413 |
| | ANN-MIC | **198** | 254 | 292 | 317 | 338 | 357 | 371 | 385 | 398 | 403 |
| | MLR-MIC | 203 | 263 | 306 | 338 | 361 | 382 | 394 | 402 | 412 | 417 |
| CORR | GBRT-MIC | 0.9701 | **0.9620** | **0.9539** | **0.9492** | **0.9445** | **0.9416** | **0.9386** | **0.9344** | **0.9302** | **0.9284** |
| | SVR-MIC | 0.9753 | 0.9596 | 0.9438 | 0.9340 | 0.9214 | 0.9136 | 0.9055 | 0.9014 | 0.8973 | 0.8959 |
| | ANN-MIC | **0.9756** | 0.9596 | 0.9462 | 0.9363 | 0.9272 | 0.9184 | 0.9114 | 0.9045 | 0.8975 | 0.8948 |
| | MLR-MIC | 0.9744 | 0.9564 | 0.9407 | 0.9273 | 0.9164 | 0.9061 | 0.8998 | 0.8953 | 0.8900 | 0.8870 |
| KGE | GBRT-MIC | 0.9493 | **0.9382** | **0.9270** | **0.9230** | **0.9189** | **0.9155** | **0.9121** | **0.9101** | **0.9081** | **0.9045** |
| | SVR-MIC | 0.9506 | 0.9190 | 0.8700 | 0.8434 | 0.8165 | 0.8201 | 0.8053 | 0.7955 | 0.7817 | 0.7751 |
| | ANN-MIC | **0.9631** | 0.9375 | 0.9154 | 0.9021 | 0.8891 | 0.8791 | 0.8703 | 0.8635 | 0.8599 | 0.8611 |

| | | | | | | | | | | | |
|---|---|---|---|---|---|---|---|---|---|---|---|
| | MLR-MIC | 0.9619 | 0.9318 | 0.9062 | 0.8872 | 0.8701 | 0.8551 | 0.8430 | 0.8344 | 0.8260 | 0.8188 |
| BHV | GBRT-MIC | **-0.1909** | **-0.1909** | **-0.1909** | **-0.1909** | **-0.1909** | **0.0886** | **0.3681** | **0.2720** | **0.1759** | **0.2720** |
| | SVR-MIC | -1.6396 | -3.3890 | -6.4785 | -7.7173 | -9.3569 | -8.1066 | -9.0180 | -9.4042 | -9.9785 | -10.7111 |
| | ANN-MIC | -0.2509 | -0.7876 | -1.2337 | -1.5023 | -1.6509 | -1.8062 | -2.7456 | -3.1596 | -3.1661 | -2.8261 |
| | MLR-MIC | -0.6867 | -2.0428 | -2.9254 | -3.8346 | -4.1555 | -4.4089 | -5.7323 | -5.7912 | -5.8660 | -6.4303 |
| ia | GBRT-MIC | 0.9844 | **0.9800** | **0.9756** | **0.9731** | **0.9705** | **0.9688** | **0.9671** | **0.9651** | **0.9631** | **0.9619** |
| | SVR-MIC | 0.9870 | 0.9780 | 0.9670 | 0.9598 | 0.9506 | 0.9473 | 0.9414 | 0.9381 | 0.9342 | 0.9325 |
| | ANN-MIC | **0.9874** | 0.9788 | 0.9713 | 0.9657 | 0.9603 | 0.9552 | 0.9511 | 0.9473 | 0.9435 | 0.9424 |
| | MLR-MIC | 0.9868 | 0.9770 | 0.9680 | 0.9603 | 0.9537 | 0.9474 | 0.9432 | 0.9402 | 0.9367 | 0.9345 |

*None :* The bold numbers represent the values of performance criterion for the best fitted models.

*3. The authors found that three-day lagged values of inflow and rainfall have less impact on 10-day ahead forecasting of inflow in Section 4.5. This is a clue that more lagged values of input variables should have been included in the models' structure.*

**Response:** Thank you for your careful review and suggestion. According to your suggestion, PACF and CCF are used to determining the model structures for inflow and rainfall, respectively, in these days (see Question 2). More details will be given in the revised version.

*4. The employed performance indices, specifically the coefficient of determination, seems insufficient to compare several model performances. More distinctive performance indices such as degree of agreement and Kling-Gupta efficiency metrics should have been used.*

**Response:** Thanks. The Pearson correlation coefficient (CORR) is a good measurement of the average error. The root mean square error (RMSE) and mean absolute error (MAE) are the most commonly used criteria to assess model performance (Luo et al., 2019; Chau, 2005; Chau, 2006). The Pearson correlation coefficient (CORR) is a good measurement of the average error. According to Referee (#2)'s and your suggestions, Nash-Sutcliffe efficiency coefficient (NSE) is removed, Kling-Gupta efficiency metrics (KGE), the percent bias in flow duration curve high-segment volume (BHV) and the Index of Agreement (IA) are introduced as supplements. Kling–Gupta efficiency scores (KGE) (Knoben et al., 2019) is also a widely used evaluation index. It can be provided as following Eq. (1) and (2).

$$KGE = 1 - \sqrt{(CORR - 1)^2 + \left(\frac{\hat{\sigma}}{\sigma} - 1\right)^2 + \left(\frac{\bar{Q}}{\bar{Q}} - 1\right)^2} \tag{1}$$

$$CORR = \frac{\sum_{i=1}^{n}(Q_i - \bar{Q})(\widehat{Q_i} - \overline{\hat{Q}})}{\sqrt{\sum_{i=1}^{n}(Q_i - \bar{Q})^2}\sqrt{\sum_{i=1}^{n}(\widehat{Q_i} - \overline{\hat{Q}})^2}}, \hat{\sigma} = \sqrt{\frac{1}{n}\sum_{i=1}^{n}\left(\widehat{Q_i} - \overline{\hat{Q}}\right)^2}, \sigma = \sqrt{\frac{1}{n}\sum_{i=1}^{n}(Q_i - \bar{Q})^2} \tag{2}$$

where $\widehat{Q_i}$ and $Q_i$ are the inflow estimation and observed value at time $i$, respectively and $n$ is the number of samples. $\overline{Q}$ is the mean of the estimation values. $\sigma$ is the standard deviation of the observed values, $\widehat{\sigma}$ is the standard deviation of the inflow estimation.

The percent bias in flow duration curve high-segment volume (BHV) (Yilmaz et al., 2008; Vogel and Fennessey, 1994) is used to evaluate performance of peak inflow forecasting. It can be provided as following Eq. (3).

$$BHV = \frac{\sum_{h=1}^{H}(\widehat{Q}_h - Q_h)}{\sum_{h=1}^{H} Q_h} \times 100 \tag{3}$$

where $h = 1, 2,...H$ are the flow indices for flows with exceedance probabilities lower than 0.02.

The Index of Agreement (IA) (Willmott, 1981) plays a significant role in evaluating the degree of the agreement between observed values and inflow estimation. It is given by Eq. (4).

$$IA = 1 - \frac{\sum_{i=1}^{n}(\widehat{Q_i} - Q_i)^2}{\sum_{i=1}^{n}(|\widehat{Q}_i - \bar{Q}| + |Q_i - \bar{Q}|)^2} \tag{4}$$

More details will be given in the revised version.

*6. The selected ranges of the model parameters seem highly subjective. Please justify the selected ranges of the model parameters, especially in Section 4.2.*

**Response:** Thank you for your careful review and suggestion. Specifying the selected ranges of the model parameters is the trickiest part of hyperparameter optimization. For gradient boosting regression trees model, we refer to (Fienen et al., 2018; Friedman, 2001; Pedregosa et al., 2011) to inform our choices of hyperparameter distributions by placing greater probability where we think the best values are. It can be difficult to figure out the interaction between hyperparameters. grid search is considered as an effective parameter search method, which is widely used (Fienen et al., 2018). In addition, more wide range of parameters has been performed according to your suggestion. For artificial neural networks-maximal information coefficient (ANN-MIC), a range of 2-20 neurons and four activation functions are selected by grid searching. Table 2 shows results of parameter optimization of ANN-MIC. Table 3 and Table 4 show results of parameter optimization of support vector regression-maximal

information coefficient (SVR-MIC) and gradient boosting regression trees-maximal information coefficient (GBRT-MIC). However, comparing with selected ranges of the model parameters in the original manuscript, the new optimization to max_leaf_nodes, min_samples_leaf, max_depth and min_samples_split of GBRT-MIC generates 36100 models which spends about 200 minutes using 12 cores for parallel computing in each leadtime. All computations of this paper are performed on a ThinkPad P1 workstation containing an Intel Core i7-9850H CPU with 2.60 GHz and 16.0 GB of RAM, using the version 3.7.10 of Python and scikit-learn package (Pedregosa et al., 2011). How to decrease time consuming of grid search will be our next research direction.

Table 2: Tuning parameters of ANN-MIC

| Model | Tuning parameter | 1 | 2 | 3 | 4 | 5 | 6 | 7 | 8 | 9 | 10 |
|---|---|---|---|---|---|---|---|---|---|---|---|
| ANN-MIC | Structure | 19-5-1 | 19-2-1 | 19-3-1 | 19-2-1 | 19-2-1 | 19-2-1 | 19-2-1 | 19-2-1 | 19-2-1 | 19-2-1 |
| | Activate function | tanh | logistic | tanh | logistic | logistic | logistic | logistic | logistic | logistic | tanh |

Table 3: Tuning parameters of SVR-MIC.

| Model | Tuning parameter | Tuning range | 1 | 2 | 3 | 4 | 5 | 6 | 7 | 8 | 9 | 10 |
|---|---|---|---|---|---|---|---|---|---|---|---|---|
| SVR-MIC | C | **(1, 100, 20)** | 8.7368 | 50.0000 | 29.3684 | 19.0526 | 21.6316 | 6.1579 | 16.4737 | 6.1579 | 8.7368 | 3.5789 |
| | epsilon | **(0.001, 0.1, 20)** | 0.00048 | 0.00953 | 0.00032 | 0.00011 | 0.00001 | 0.00001 | 0.00032 | 0.00022 | 0.00058 | 0.00001 |
| | gamma | **(0.0001, 0.01, 20)** | 0.0100 | 0.0095 | 0.0019 | 0.0072 | 0.0019 | 0.0067 | 0.0038 | 0.0057 | 0.0072 | 0.0067 |

*Note*: The bold parts, (min, max, step) represent $[min + \frac{max-min}{step-1} \times 0, min + \frac{max-min}{step-1} \times 1, ..., min + \frac{max-min}{step-1} \times (step - 1)]$.

Table 4: Tuning parameters of GBRT-MIC

| Tuning parameter | Tuning range | Optimal parameters (the lead times of 1-10 days) | |
|---|---|---|---|
| | | GBRT | GBRT-MIC |
| max_leaf_nodes | [2, 3, …, 20] | 7,3,3,3,3,2,3,3,3,3 | 10,11,16,11,12,10,6,6,4,4 |
| min_samples_leaf | [1, 2, …, 10] | 2,3,1,1,10,1,2,2,4,1 | 5,9,1,5,6,9,4,6,6,10 |
| max_depth | [1, 2, ..., 10] | 3,2,2,4,2,1,2,2,2,10 | 4,4,8,7,10,6,6,4,5,3 |
| min_samples_split | [2, 3, …, 20] | 9,14,13,20,11,3,6,2,3,4 | 16,20,19,16,20,20,17,17,20,3 |
| n_estimators | [500,550, …, 4000] | 1000,1000,1000,1500,1500,2500,1500,3500,2500,2500 | 2500,1000,1500,2500,1500,2500,1000,1000,1500,300 |
| learning_rate | [0.001,0.0025,0.005,0.0075,0.01,0.025,0.05, 0.075,0.1] | 0.01,0.01,0.01,0.005,0.005,0.005,0.005,0.0025,0.0025,0.0025 | 0.0075,0.01,0.01,0.0025,0.01,0.0025,0.01,0.01,0.01,0.01 |

7. The range for the number of hidden neurons (i.e. 2–20) seems too high. Please justify this from a hydrological perspective. Because using a high number of hidden neurons could lead to overfitting that resulted in a poor performance in multi-step forecasting.

**Response:** Thank you. Specifying the number of hidden neurons is a difficult task (Badrzadeh et al., 2013) and the number of hidden neurons is determined by trial and error procedure in the original paper. In cases where we aren't sure about the best number of hidden neurons, we can use wide ranges and let the trial and error procedure do the reasoning for us. It is found that the optimal number of neurons is 2, 3 or 5 (see Table 4). More details will be given in the revised version.

[Figure]

Figure 2: Sensitivity of the number of nodes and activation function in the hidden layer on the MAE of ANN-MIC, the shadow part is 95% confidence interval obtained by bootstrap of 50 trials. (a) One-day-ahead (b) Ten-day-ahead.

*23. The authors did not discuss the reasons why NSE values for lead times of 6-7-8-9-day is worse than the value of lead time of 10-day.*

**Response:** Thank you for your careful review. It should be noted that NSE values for lead times of 6-7-8-9-day is worse than the value of lead time of 10-day in the train set and validation set. We consider the possible reasons are parameter optimization and model structure. According to Referee (#2)'s and your suggestions, NSE is removed and KGE, BHV and IA are introduced as supplements. Table 1 and Figure 3 shows performance indices of model in the test set. More details will be given in the revised version.

[Figure]

Figure 3: Performance of GBRT-MIC, SVR-MIC, ANN-MIC and MLR-MIC for the test set (2017-2018). (a) MAE (b) RMSE (c) CORR (d) KGE (e) BHV(f)IA.

*24. It is not clear how top k features were selected according to the chosen threshold value. Did the authors employ several threshold values? Please give more details on this issue.*

**Response:** Thank you for your careful review and suggestion. The original manuscript totally employs three threshold values. Two of these thresholds were used to determine the model input structures with inflow and rainfall (See Question 1 for details). Another threshold value was used to determine the model input structures with ERA-Interim dataset. Further, we perform input selection in two steps via the maximal information coefficient (MIC). First, compute MIC value of each reanalysis factors and observed inflow. Then, sort features based on MIC in a descending order and determine the optimum inputs using trail-and-error method, i.e. starting from the top one feature and then modifying the external input feature by

successively adding one more feature into model input (Moosavi et al., 2013; Shoaib et al., 2015). Finally, the top 14 reanalysis variables are selected as the input (Table 5, No.6-19). More details will be given in the revised version.

Table 5: List of inputs of GBRT-MIC. There are of two types, observed and reanalysis variables. The reanalysis variables are available four time a day at 00:00 UTC, 06:00 UTC, 12:00 UTC and 18:00 UTC. The cumulative variable (e.g., Total column water) is the sum of four periods and the instantaneous variable (e.g. 2 meter dewpoint temperature) is the mean of four periods.

| No. | Variable | Index | Unit | MIC | Type |
|---|---|---|---|---|---|
| 1 | Inflow at day $t-1$ | $Q_{t-1}$ | $m^3 \cdot s^{-1}$ | - | Obs. |
| 2 | Inflow at day $t-2$ | $Q_{t-4}$ | $m^3 \cdot s^{-1}$ | - | Obs. |
| 3 | Rainfall at day $t-1$ | $R_{t-1}$ | $mm$ | - | Obs. |
| 4 | Rainfall at day $t-2$ | $R_{t-2}$ | $mm$ | - | Obs. |
| 5 | Rainfall at day $t-3$ | $R_{t-3}$ | $mm$ | - | Obs. |
| 6 | Forecast albedo | $fal\_t$ | - | 0.865 | ERA-I |
| 7 | Soil temperature level 3 | $stl3_t$ | $K$ | 0.846 | ERA-I |
| 8 | 2 meter dewpoint temperature | $d2m_t$ | $K$ | 0.781 | ERA-I |
| 9 | Total column water vapour | $tcwv_t$ | $kg \cdot m^{-2}$ | 0.699 | ERA-I |
| 10 | Total column water | $tcw_t$ | $kg \cdot m^{-2}$ | 0.699 | ERA-I |
| 11 | Soil temperature level 2 | $stl2_t$ | $K$ | 0.689 | ERA-I |
| 12 | Minimum temperature at 2 meters | $mn2t_t$ | $K$ | 0.683 | ERA-I |
| 13 | Surface thermal radiation downwards | $strd_t$ | $J \cdot m^{-2}$ | 0.669 | ERA-I |
| 14 | Temperature of snow layer | $tsn_t$ | $K$ | 0.664 | ERA-I |
| 15 | Soil temperature level 4 | $stl4_t$ | $K$ | 0.642 | ERA-I |
| 16 | Soil temperature level 1 | $stl1_t$ | $K$ | 0.631 | ERA-I |
| 17 | Surface net thermal radiation, clear sky | $strc_t$ | $J \cdot m^{-2}$ | 0.620 | ERA-I |
| 18 | Runoff | $ro_t$ | $m$ | 0.619 | ERA-I |
| 19 | Volumetric soil water layer 1 | $swvl1_t$ | $m^3 \cdot m^{-3}$ | 0.614 | ERA-I |

---

## Author Response (AR1)

Dear Editor and Referees,

Thank you for your time and for your thoughtful and constructive review; they have greatly improved the manuscript. Below, we address the points risen by the three anonymous reviewers and state how we would like to address them in a revised version of the manuscript. Our replies to the reviewers' comments are written in blue and
5   normal font. We believe we have substantially addressed all of the outstanding comments and issues, and we look forward to your second review of the work. A marked-up version of the manuscript can be found right after the point-by-point answers at the end of this document.

On the behalf of all co-authors,

Yours sincerely,

10   Zhanwei Liu

Thank you very much for your time and for your thoughtful and constructive review. The following are our point-by-point responses to your comments.

15 *1. The manuscript presents multi-step ahead daily inflow forecasting using ERA-Interim reanalysis dataset based on gradient boosting regression trees, which is interesting. It is relevant and within the scope of the journal.*

**Response:** Thank you very much for your positive comments.

*2. Full names should be shown for all abbreviations in their first occurrence in texts. For example, ERA in page 1, ECMWF in page 3, etc.*

20 **Response:** Thank you for your carefulness. We have shown full names for all abbreviations in their first occurrence in the revised manuscript. Please see Section Introduction for more details.

*3. For readers to quickly catch your contribution, it would be better to highlight major difficulties and challenges, and your original achievements to overcome them, in a clearer way in abstract and introduction.*

**Response:** Thank you for your suggestion. The major difficulties and challenges are selection of appropriate input
25 variables related to inflow of longer lead times and effective prediction model. This paper proposed a new hybrid inflow forecast framework with ERA-Interim (European Centre for Medium-Range Weather Forecasts (ECMWF) Re-Analysis Interim) data as input, adopting gradient boosting regression trees (GBRT) and the maximal information coefficient (MIC) for multi-step ahead daily inflow forecasting. The proposed inflow forecast framework has three advantages. Firstly, the ERA-Interim dataset provides enough information for the framework
30 to discover inflow for longer lead times. Secondly, MIC can identify effective feature subset from massive features that significantly affects inflow so that the framework can reduce computational burden, distinguish key attributes with unimportant ones and provide a concise understanding of inflow. Lastly, the GBRT is a prediction model in the form of an ensemble of decision trees and has a strong ability to capture nonlinear relationships between input and output in longer lead times more fully. We have made careful modifications in Section Abstract and
35 Introduction of the revised manuscript.

*4. It is mentioned in page 1 that ERA-Interim reanalysis data is adopted as input. What are other feasible alternatives? What are the advantages of adopting this particular data over others in this case? How will this affect the results? The authors should provide more details on this.*

**Response:** Thank you for your careful review and suggestion. The ERA-Interim data is the result of assimilating
40 observed data with forecast data, which has less error than observed data and forecast data (Balsamo et al., 2015). ERA-Interim data is produced by a fixed version of numerical weather prediction (NWP) system (Dee et al., 2011). The fixed version ensures there are no spurious trends caused by an evolving NWP system. Therefore, meteorological reanalysis data satisfies the need for long sequences of consistent data and have been used for the prediction of wind speeds (Stopa and Cheung 2014) and solar radiation (Linares-Rodríguez, Ruiz-Arias et al. 2011,

Ghimire, Deo et al. 2019). Meanwhile, ERA-Interim was proved to be one of the best reanalysis data describing atmospheric circulation and elements (Kishore et al., 2011). More details about ERA-Interim data are given in Section Appendix A of the revised manuscript.

*5. It is mentioned in page 1 that gradient boosting regression tree is adopted as inflow forecast framework. What are the advantages of adopting this particular soft computing technique over others in this case? How will this affect the results? The authors should provide more details on this.*

**Response:** Thank you for your careful review and suggestion. The gradient boosting regression trees (GBRT) (Friedman 2001, Fienen, Nolan et al. 2018), is a nonparametric machine learning method based on a boosting strategy and the decision trees model. The decision tree robust to outliers is used as a primitive model and boosting algorithm as integration rule is used to improve inflow forecasting accuracy. GBRT was developed and had been used in traffic (Zhan, Zhang et al., 2019) and environmental (Wei, Meng et al., 2019) field and proved to alleviate the problems of being trapped by local minima, over-fitting problems and reduced generalizing performance. More details about GBRT is given in Section 3.2 of the revised manuscript.

*6. It is mentioned in page 1 that artificial neural networks, support vector regression and multiple linear regression models are adopted as benchmark for comparison. What are the other feasible alternatives? What are the advantages of adopting these particular models over others in this case? How will this affect the results? More details should be furnished.*

**Response:** Thank you for your careful review and suggestion. The several studies had shown that artificial neural networks (ANN) (Rasouli et al., 2012; Cheng et al., 2015; El-Shafie and Noureldin, 2011; Chau, 2006; Ali Ghorbani et al., 2018) and support vector regression (SVR) (Tongal and Booij, 2018; Luo et al., 2019; Moazenzadeh et al., 2018) are the two powerful models for inflow predicting. They are widely used and very mature algorithms, which are scientific and reasonable compared with them. Please see Section Introduction for more details about compared model.

*7. It is mentioned in page 3 that the Xiaowan Hydropower Station is adopted as the case study. What are other feasible alternatives? What are the advantages of adopting this particular case study over others in this case? How will this affect the results? The authors should provide more details on this.*

**Response:** Thank you for your careful review and suggestion. The Xiaowan Hydropower Station in the lower reaches of the Lancang River, which is the longest river with most discard water in Yunnan Province, is chosen as the study site (as shown in Fig. 2 in the revised manuscript). The Xiaowan Hydropower Station is the main controlling hydropower station in the Lancang River and it is very meaningful to adopt the Xiaowan Hydropower Station as the case study. Please see Section 2.1 of the revised manuscript for more details about case study.

*8. It is mentioned in page 4 that the maximum information coefficient is adopted to select inputs from 79 potential predictors from reanalysis data. What are the advantages of adopting this particular approach over others in this case? How will this affect the results? The authors should provide more details on this.*

**Response:** Thank you for your careful review and suggestion. The maximal information coefficient (MIC) (Reshef et al., 2011) is a robust measure of the degree of correlation between two variables and has attracted a lot attention from academia (Zhao et al., 2013; Ge et al., 2016; Lyu et al., 2017; Sun et al., 2018), which can identify effective feature subset from massive features that significantly affects inflow so that the framework can reduce computational burden, distinguish key attributes with unimportant ones and provide a concise understanding of inflow. Please see Section 3.1 of the revised manuscript for more details about inputs selection.

*9. It is mentioned in page 4 that autocorrelation function is adopted to identify observed inflow and rainfall lags. What are other feasible alternatives? What are the advantages of adopting this particular approach over others in this case? How will this affect the results? The authors should provide more details on this.*

**Response:** Thank you for your careful review and suggestion. The autocorrelation function (ACF) measures the dependency or relationship of observed value with lagged observations of a considered variable. In a long memory time series such as inflow time series, the ACF declines slowly (as shown in Fig. R1). The partial autocorrelation function (PACF) and cross-correlation function (CCF) (as shown in Fig. 6 in the revised manuscript) are two other feasible alternatives. We use the PACF and CCF for modeling, calculation and analysis according to Referee(#3)'s suggestion. We agree to use PACF and CCF replace ACF to determine the model structure. In the revised manuscript, PACF and CCF are adopted to determining the model structures for inflow and rainfall, respectively. 95% confidence interval is used to determine the significant relationships replacing user-defined threshold value. Please see Section 2.2 and Section 4.1 of the revised manuscript for more details.

[Figure]

Figure R1. ACF plots of Xiaowan inflow time series.

*10. It is mentioned in page 6 that four evaluation criteria are adopted to evaluate the performance of the models. What are the other feasible alternatives? What are the advantages of adopting these particular evaluation criteria over others in this case? How will this affect the results? More details should be furnished.*

**Response:** Thank you for your careful review and suggestion. The root mean squared error (RMSE) and mean absolute error (MAE) are the most commonly used criteria to assess model performance (Luo et al., 2019; Chau, 2005; Chau, 2006). The Pearson correlation coefficient (CORR) is a measure of the strength of the association between observed inflow series and forecasted inflow series. Nash-Sutcliffe efficiency coefficient (NSE) is

replaced by Kling-Gupta efficiency metrics (KGE) and the Index of Agreement (IA) according to results of trials and Referee(#3)'s suggestion. The percent bias in flow duration curve high-segment volume (BHV) is introduced to evaluate the performance of forecasting extreme values for developed model according to results of trials and Referee(#2)'s suggestion. KGE, IA and BHV and are added to compare several model performances in Section 3.3 of the revised manuscript.

*11. It is mentioned in page 7 that a grid search algorithm is adopted to optimization model parameters. What are other feasible alternatives? What are the advantages of adopting this particular algorithm over others in this case? How will this affect the results? The authors should provide more details on this.*

**Response:** Thank you for your careful review and suggestion. The grid search algorithm, which is an exhaustive search all candidate parameter combination method, is guided to optimizing model parameters by evaluation of validation set for each lead time (Chicco and Davide, 2017). Grid search is considered as an effective parameter search method, which is widely used (Fienen et al., 2018). Two of other feasible alternatives are randomized search and Bayesian optimization. We have performed some numerical trials to compare grid search, randomized search and Bayesian optimization, and grid search can obtain more reasonable and stable hyperparameter combination. More details about grid search are given in Section 4.2 of the revised paper.

*12. It is mentioned in page 9 that grid searching is adopted to tune the hyperparameters of GBRT, GBRT-MIC, ANN-MIC. What are other feasible alternatives? What are the advantages of adopting this particular approach over others in this case? How will this affect the results? The authors should provide more details on this.*

**Response:** Thank you for your careful review and suggestion. Same as question 11, more details about grid search are given in Section 4.2 of the revised paper.

*13. It is mentioned in page 9 that Bayesian optimization (Snoek et al., 2012) is adopted to tune the hyperparameters of SVR-MIC. What are other feasible alternatives? What are the advantages of adopting this particular approach over others in this case? How will this affect the results? The authors should provide more details on this.*

**Response:** Thank you for your careful review and suggestion. At present, there are three commonly used methods of hyperparameter selection: grid search, random search and Bayesian optimization (Snoek et al., 2012). We have performed some numerical trials to compare grid search, randomized search and Bayesian optimization, and grid search can obtain more reasonable and stable hyperparameter combination. Bayesian optimization has been replaced by grid search method. Please see Section 3.4 for more details.

*14. It is mentioned in page 9 that Python is adopted to perform all computations. What are other feasible alternatives? What are the advantages of adopting this particular software over others in this case? How will this affect the results? The authors should provide more details on this.*

**Response:** Thank you for your careful review and suggestion. Python is an important tool for scientific computing and data analysis, which is powerful, fast and open. Brief introduction about Python is given in Section 4 of the revised manuscript.

*15. Some key parameters are not mentioned. The rationale on the choice of the particular set of parameters should be explained with more details. Have the authors experimented with other sets of values? What are the sensitivities of these parameters on the results?*

**Response:** Thank you for your careful review and suggestion. For ANN, A range of 2-20 neurons and four activation functions (as shown in Table 4 in the revised manuscript) are selected by a trail-and-error procedure. The sensitivities of these parameters have been analyzed by trying different parameter combinations (as shown in Fig. 7 in the revised manuscript). Meanwhile, referring to (Fienen et al., 2018; Friedman, 2001; Pedregosa et al., 2011), more wide ranges of the model parameters are used for grid search in Section 4.2 of the revised manuscript.

*16. Some assumptions are stated in various sections. Justifications should be provided on these assumptions. Evaluation on how they will affect the results should be made.*

**Response:** Thank you for your careful review and suggestion. The comparison of different models is based on the basic assumption that parameters are optimal. In the revised manuscript, grid search is employed to tune the hyperparameters of model. A lot of models for each lead time are developed to find as possible as optimal parameters and this assumption of optimal parameters can be satisfied. More details about the assumption of optimal parameters are given in Section 4.2 of the revised paper.

*17. The discussion section in the present form is relatively weak and should be strengthened with more details and justifications.*

**Response:** Thank you for your careful review and suggestion. KGE, IA and BHV are added to compare several model performances and more details about the discussion of the obtained results have been added. The discussion of the obtained results is enriched in Section 4 of the revised manuscript.

*18. Moreover, the manuscript could be substantially improved by relying and citing more on recent literatures about contemporary real-life case studies of soft computing techniques in hydrological prediction such as the followings: ïAn Yaseen, Z.M., et al., "An enhanced extreme learning machine model for river flow forecasting: state-of-the-art, practical applications in water resource engineering area and future research direction," Journal of Hydrology 569: 387-408 2019. ïAn Fotovatikhah, F., et al., "Survey of Computational Intelligence as ̆Basis to Big Flood Management: Challenges, research directions and Future Work," Engineering Applications of Computational Fluid Mechanics 12 (1): 411-437 2018. ïAn Mosavi, A., et al., "Flood Prediction Using Machine Learning Models: Literature Review," Water 10 (11): article no. 1536 2018. ïAn Moazenzadeh, R., et al., "Coupling a ̆firefly algorithm with support vector regression to predict evaporation in northern Iran," Engineering Applications of Computational Fluid Mechanics 12 (1): 584-597 2018. ïAn Ghorbani, M.A., et al., "Forecasting pan evaporation with an integrated Artificial Neural Network Quantum-behaved Particle Swarm*

*Optimization model: a case study in Talesh, Northern Iran," Engineering Applications of Computational Fluid Mechanics 12 (1): 724-737 2018. ïAn Chau, K.W., et al., "Use of Meta-Heuristic Techniques in Rainfall-Runoff Modelling" Water 9(3): article no. 186, 6p 2017.*

**Response:** Thanks. We have carefully looked up the mentioned literature, which has been cited in the paper, and we have also extensively looked up other literatures from HESS, JH and other relative journals, added some necessary literatures. Please see Section References.

*19. Some inconsistencies and minor errors that needed attention are: ïAn Replace "... was supply to depict…" with "…was supplied to depict ..." in line 86 of page 3 ïAn Replace "... into train set, validation set, and test set..." with "...into training set, validation set, and testing set..." in line 206-207 of page 7 ïAn Replace "... test set..." with "... testing set..." in line 209 of page 7 ïAn Replace "… more accuracy inflow forecasting..." with "... more accurate inflow forecasting ..." in line 283 of p10 ïAn Replace "... arisen in in areas..." with "... arisen in areas..." in line 288 of page 10 ïAn Replace "... for train, validation and test set ..." with "... for training, validation and testing set ..." in line 294 of page 10 ïAn Replace "... According to compare the forecasted results of …" with "... According to the comparison of forecasted results of ... " in line 330 of page 11*

**Response:** Thank you for your careful review. We agree with all of the minor changes above, and we will go through carefully the manuscript to check and correct any errors. Those typos have been corrected in the revision.

*20. In the conclusion section, the limitations of this study, suggested improvements of this work and future directions should be highlighted.*

**Response:** Thank you for your careful review and suggestion. We have carefully checked the conclusion of the article and added the limitations and future directions of this study in the revised manuscript. Please see page Section 5 for more details.

*I went through the manuscript. Generally, the manuscript has been well organized.*

195   **Response:** We thank you very much for reviewing the manuscript and giving the positive comment. The following are our point‑by‑point responses to your comments.

*1. P1, L6, In abstract the authors stated that "The impacts of climate change and human activities make accurate inflow prediction increasingly difficult, especially for longer lead times". As far as I know, the climate change deals with long term trends, say the climate variation over 20 years. I cannot understand relevance of the*
200   *abovementioned with climate change impacts and human activities.*

   **Response:** Thank you for your careful review. Climate variation affects the streamflow directly. For example, changing precipitation patterns and intensity, together with changing temperatures, will greatly modify the streamflow. Human activities such as land use change, water withdrawal, and hydraulic structures have substantial impacts on streamflow. We fully approve that climate variation and human activities can generate large effect to
205   streamflow of medium and long term. For short term streamflow forecasting, climate variation and human activities also have some effect, so we need to calibration parameter according to meteorological factors. Please see Section Introduction for more details.

*2. The authors have to clearly indicate which model was developed for the inflow forecasting. At first they have to demonstrate if they used conceptual models or data driven models. What is the advantage of the developed*
210   *model?*

   **Response:** Thank you for your careful review. The gradient boosting regression trees (GBRT) is used to forecast daily streamflow. The model is a data-driven model. Compared with artificial neural network (ANN), GBRT has two main advantages. Firstly, GBRT can rank features according to their contribution to model scores, which is of great significance for reducing the complexity of the model. Secondly, GBRT is a white box model and can be
215   easily interpreted. The more details and advantages of model developed are given in Section 3.2 of the revised manuscript according to your suggestion.

*3. The input selection for multi-day ahead forecasting should be discussed according to available literature. It is essential why the input structure of the longer period is not updated following literature the earlier stage forecasts.*

220   **Response:** Thank you for your careful review and suggestion. We have carefully reviewed more literatures about the input selection for multi-day ahead forecasting. There are mainly two strategies that you can use for multi-step forecasting for single-output, namely, Static (Direct) multi-step forecast and Recursive multi-step forecast (Gianluca et al., 2013; Taieb et al., 2012). Recursive forecast strategy is biased when the underlying model is nonlinear; it is sensitive to the estimation error instead of actual ones since estimated values are more and more
225   used when we get further in the future (Bontempi et al., 2012). Thus, the Static multi-step forecasting strategy is

employed in this paper. Since the Static strategy does not use any approximated values to compute the forecasts, it is not prone to any accumulation of errors. The model structure of one-step and two-step forecasting of Static strategy is listed below (as shown in Section 3.4) which has different model parameters.

$$prediction(t + 1) = model1\big(obs(t - 1), obs(t - 2), \ldots, obs(t - n)\big)$$

$$prediction(t + 2) = model2\big(obs(t - 1), obs(t - 2), \ldots, obs(t - n)\big)$$

where $obs(t - 1)$ is the observed value at the $t - 1$ period and $prediction(t + 1)$ is the predicted value of one-step at the $t$ period. More details about multi-step forecasting are shown in Section 3.4 of the revised manuscript.

*4. Literature should be updated discussing on more papers addressing multi-step ahead forecasting.*

**Response:** Thank you for your careful review. We have carefully reviewed more literatures from HESS, JH and relative journals about multi-step ahead forecasting and the more references about multi-step ahead forecasting have been updated in Section Reference of the revised manuscript.

*5. The authors employed gradient boosting regression trees as an ensemble framework. More explanations required about ensemble members.*

**Response:** Thank you for your careful review and suggestion. The ensemble member is the decision tree model. More details about decision tree model are given in Section 3.3.1 in the revised manuscript.

*6. Uncertainty analysis should be carried out to show how much the predictions are confident. As the lead time increases, the metrics reveal errors are increasing drastically. Moreover, high uncertainties are expected to associate with such models. Please discuss this issues accordingly.*

**Response:** Thank you for your careful review. We agree that uncertainty analysis in predictions is significant. As far as we know, medium and long-term forecasting is more uncertainty, for example, monthly or yearly. This paper focuses on improving inflow prediction accuracy of lead times of one to ten days by developing new model and importing ERA-Interim reanalysis data, which aims to providing reference for reducing discard water. As time goes on, model often needs to be rebuilt and parameters of model need to be recalibrated according to the actual flow and meteorological data. The uncertainty analysis of medium and long-term inflow forecasting will be further studied in the next study.

*7. Concerning inflow predictions, please indicate efficiency of the proposed model to simulate and predict extreme values which are of great importance.*

**Response:** Thank you for your careful review and suggestion. We introduce the percent bias in flow duration curve high-segment volume (BHV) to evaluate the performance of catching extreme values for developed model and more details have been given in Section 3.3 of the revised manuscript.

We thank you very much for reviewing the manuscript and giving the positive comment. The following are our

260 point-by-point responses to your comments.

*1. In this manuscript, the authors compared several data-driven models for multi-step forecasting of inflow. The employed models include gradient boosting regression trees (GBRT), artificial neural networks (ANN), support vector regression (SVR), and multiple linear regression (MLR) models. The models were developed by considering (1) streamflow and rainfall record, and (2) ERA-Interim reanalysis data. Further, the maximum*

265 *information coefficient and autocorrelation functions were utilized to construct the input structures of the models. The authors concluded that the developed methodology that considers ERA-Interim reanalysis data considerably gives better results in the forecasting of inflows at lead times of 5-10 days. The manuscript is well written and organized. However, there is not a significant novelty in the manuscript except using ERA-Interim dataset. Further, there are severe weaknesses in the developing of the model input structures.*

270 **Response:** Thank you very much for your time and for your thoughtful and constructive review, and also thank you for giving some positive comments. We have made careful modifications in Abstract and Introduction of the revised manuscript for readers to quickly catch our contribution. This paper focuses on improving prediction accuracy by three significant measures. Firstly, ERA-Interim reanalysis data is introduced to provide enough information for the model to discover inflow for longer lead times. Secondly, gradient boosting regression trees

275 (GBRT) is adopted to implement inflow forecasting and GBRT has been used to achieve multi-step inflow forecasting. Thirdly, most widely used models are developed to compare with GBRT for multi-step inflow forecasting, which demonstrates that developed model improves inflow forecasting accuracy. In order to make it easier for the reader to grasp the innovation of this paper, we have modified the Abstract and Introduction carefully to make the innovation more prominent. More details are given in the revised version.

280 *2. The authors made a significant mistake in using the autocorrelation function (ACF) in determining the model structures. They should have employed cross-correlation and partial autocorrelation functions (or other measures) to establish the relationship between the observed records and inflow. The ACF only measures the dependency or relationship of observed value with lagged observations of a considered variable. In a long-dependent series such as inflow time series, the ACF will decay slowly. Therefore, statistically significant*

285 *relationships between the observed and lagged values could not be determined. To determine the significant relationships, the authors employed user-defined threshold value. The obtained inflow and rainfall values for the input structures of the models include only three lagged-day values as could be expected. This number could be higher based on the selected threshold. However, this finding does not convey any meaningful relationship between the observed records (i.e. inflow and rainfall) and the inflow values. The PACF should have been used*

290 *for determining the lagged relationships of inflows since the inflow time series mainly shows the long-memory feature where the correlation decays after a long observation period. Further, all statistically significant lagged variables should have been included in the model structures found in PACF. Using a user-defined threshold value is a serious mistake in this situation.*

**Response:** Thank you for your careful review and nice comments. According to your suggestions, we use the partial autocorrelation function (PACF) and cross correlation function (CCF) for modeling, calculation and analysis, and find that your suggestions are effective. We agree to replace the autocorrelation function (ACF) to determine the model structure and 95% confidence levels obtained by hypothesis test is used to replace user-defined threshold value to determine the significant relationships (as shown in Fig. 6 in the revised manuscript). The all calculation results have been updated accordingly (as shown in Table 3 in the revised manuscript). In Section 4.1 of the revised manuscript, PACF and CCF to determining the model structures for inflow and rainfall, respectively.

*3. The authors claimed that the proposed methodology "significantly" improves the accuracy of inflow prediction for longer lead times. However, I do not agree with this comment. Because, as the authors mentioned, there is only about 1% and 5% improvement in two-day and 10-day ahead forecasting. Therefore, the results do not seem convincing about the superiority of ERA-Interim dataset over the common dataset, especially ill-conditioned input structures with conventional observed inflow and rainfall dataset.*

**Response:** Thank you for your careful review. Revised input structures are used to compare with developed model with ERA-Interim dataset. Table 8 of the revised paper shows performance indices of model in the testing set. The results indicate that the developed method generally performs better than other models and improves the accuracy of inflow forecasting in four and ten-day ahead forecasting. It should be noted that the results of "Supplementary response to Referee 3" has some mistakes because of a tight deadline; they have been corrected in the revised version of manuscript. More discussion about results of models are given in Section 4.4 of the revised manuscript.

*4. The authors found that three-day lagged values of inflow and rainfall have less impact on 10-day ahead forecasting of inflow in Section 4.5. This is a clue that more lagged values of input variables should have been included in the models' structure.*

**Response:** Thank you for your careful review and suggestion. According to your suggestion, PACF and CCF are used to determining the model structures for inflow and rainfall, respectively. The results indicate that one-day and four-day lagged values of inflow and one to six-day lagged values of rainfall are included in the model's structure in the revised manuscript. Please see Section 4.1 of the revised manuscript for more details.

*5. The employed performance indices, specifically the coefficient of determination, seems insufficient to compare several model performances. More distinctive performance indices such as degree of agreement and Kling-Gupta efficiency metrics should have been used.*

**Response:** Thanks. The Pearson correlation coefficient (CORR) is a measure of the strength of the association between observed inflow series and forecasted inflow series. The root mean squared error (RMSE) and mean absolute error (MAE) are the most commonly used criteria to assess model performance (Luo et al., 2019; Chau, 2005; Chau, 2006). According to Referee (#2)'s and your suggestions, Nash-Sutcliffe efficiency coefficient (NSE) is removed, Kling-Gupta efficiency metrics (KGE), the percent bias in flow duration curve high-segment volume

(BHV) and the Index of Agreement (IA) are introduced as supplements. Please see Section 3.3 of the revised manuscript for more details.

330 *6. It is not clear how the multi-step forecasting scheme (i.e., recursive or static) was employed? Please give more details about this issue.*

**Response:** Thank you for your careful review and suggestion. Recursive forecast strategy is biased when the underlying model is nonlinear and is sensitive to the estimation error, since estimated values, instead of actual ones, are more and more used when we get further in the future (Bontempi et al., 2012). Thus, the Static multi-step

335 forecasting strategy is employed and the models of different lead times have different model parameters. The model structure of one-step and two-step forecasting of Static strategy is listed below which has different model parameters.

$$prediction(t + 1) = model1\big(obs(t - 1), obs(t - 2), \dots, obs(t - n)\big)$$
$$prediction(t + 2) = model2\big(obs(t - 1), obs(t - 2), \dots, obs(t - n)\big)$$

340 where $obs(t - 1)$ is the observation value at the $t - 1$ period and $prediction(t + 1)$ is the predicted value of one-step at the $t$ period. More details about multi-step forecasting are shown in Section 3.4 of the revised manuscript.

*7. The selected ranges of the model parameters seem highly subjective. Please justify the selected ranges of the model parameters, especially in Section 4.2.*

345 **Response:** Thank you for your careful review and suggestion. Specifying the selected ranges of the model parameters is the trickiest part of hyperparameter optimization. For gradient boosting regression trees (GBRT), we refer to (Fienen et al., 2018; Friedman, 2001; Pedregosa et al., 2011) to inform our choices of hyperparameter distributions. It can be difficult to figure out the interaction between hyperparameters. We have used wide ranges of the model parameters (as shown in Table 5 and Table 6 in the revised manuscript) and the model parameters have

350 been justified in Section 4.2 of the revised manuscript.

*8. The range for the number of hidden neurons (i.e. 2–20) seems too high. Please justify this from a hydrological perspective. Because using a high number of hidden neurons could lead to overfitting that resulted in a poor performance in multi-step forecasting.*

**Response:** Thank you. Specifying the number of hidden nodes is a difficult task (Badrzadeh et al., 2013) and the

355 number of hidden nodes is determined by a trial-and-error procedure in the original paper. In cases where we are not sure about the best number of hidden nodes, we use wide ranges and let the trial-and-error procedure do the reasoning for us. It is found that the optimal number of neurons is 2, 3 or 4. Please see Section 4.2 for more details.

*9. The discussion of the obtained results should be improved with more details, especially giving necessary*
360 *citations to previous studies.*

**Response:** Thank you for your careful review and suggestion. KGE, IA and BHV have been added to compare several model performances and more details about the discussion of the obtained results has been added. In addition, we have carefully looked up the related literature from HESS, JH and other relative journals and some necessary citations to previous studies are discussed. Please see Section 5 and Reference for more details.

365 *10. It is not clear how Fig. 1 was obtained. Please give the necessary information about this figure.*

**Response:** Thank you for your careful review. We cooperate with production unit for a long time and the data of Fig. 1 from production unit has been obtained from public website. We give the source link of the data in the revised manuscript. Please see Section Introduction for more details.

*11. Please give more details on the Lines 78–82.*

370 **Response:** Thanks. More details about ERA-Interim dataset have been introduced in Section Appendix A the revised manuscript.

*12. Please give the definitions and meanings of the variables in the ERA-Interim dataset in the Appendix.*

**Response:** Thank you for your suggestion. The definitions and meanings of the variables in the ERA-Interim dataset are given in Section Appendix A of the revised manuscript.

375 *13. Please justify using the feature scaling in Line 108.*

**Response:** Thank you for your suggestion. The "data scaling" has been replaced by "feature scaling". Please see Section 2.2 for more details.

*14. What do you mean with "invalid variables" in Line 116?*

**Response:** Thanks. The "invalid variables" in Line 116 mainly demonstrate the weak-correlated variables which 380 has a weak correlation and cannot interpret inflow very well. The "invalid variables" has been modified to "redundant feature information" in the revised manuscript. Please see Section 2.2 for more details.

*15. Please prefer "maximal" or "maximum" information criterion throughout the manuscript.*

**Response:** Thank you for your careful review. All "maximum" information criterion in the original manuscript has been modified to "maximal" information criterion in the revised manuscript. Please see Section Abstract for 385 more details.

*16. Please check the term MI\*(D,X,Y) in Eq. (5) since you defined MI\*(D,x,y) in Line 130.*

**Response:** Thank you for your careful review. The term MI\*(D,X,Y) has been modified to MI\*(D,x,y). Please see Eq. (5) in Section 3.1.

*17. The definition of B(n) was given in Line 133; however it is not clear where this parameter is used.*

390 **Response:** Thank you for your careful review. $B(n)$ is the maximal grid size which is a function of sample size and we usually set $B = n^{0.6}$. Some details about $B(n)$ are added in the revised manuscript. Please see Section 3.1.

*18. Please check the terms in Eq. (7). Will they be R1(i,s) or R1(j,s)?*

**Response:** Thank you for your careful review. R1(i,s) and R2(i,s) in Eq. (7) has been modified to R1(j,s) and R2(j,s) in the revised manuscript. Please see Section 3.1.

395 *19. Please check the notations in Line 144; n features with N samples or n samples with N features according to the given definition.*

**Response:** Thanks. To avoid ambiguity, the sentence has been corrected in the revised version of the manuscript. The notation shows N features with n samples. Please see Section 3.1.

*20. There is little information about the structure of ERA-Interim dataset. Please give more details about this*
400 *dataset.*

**Response:** Thank you for your careful review. There are detailed introductions for ERA-Interim dataset in the https://www.ecmwf.int/en/forecasts/datasets/reanalysis-datasets/ era-interim. According to your suggestion, more detailed information about variables of ERA-Interim dataset has been added in the revised manuscript. More details about ERA-Interim dataset are given in Section Appendix A of the revised manuscript.

405 *21. There is not any information about grid searching methodology.*

**Response:** Thank you for your careful review and suggestion. Grid search is considered as an effective parameter search method, which is widely used (Fienen et al., 2018). The grid search methodology is introduced in detail in Section 4.2 of the revised manuscript.

*22. Please add "activation function" after "relu" in Line 248.*

410 **Response:** Thank you for your suggestion. In order to show the results more clearly, this sentence has been replaced with "The results of the trials show *tanh* and *logistic* function are two more robust activation function and ANN with fewer nodes is inclined to obtain lower error" in the revised manuscript. Please see Section 4.2.

*23. The comments in Lines 278–280 are vague.*

**Response:** Thank you for your careful review. The comments in Lines 278–280 indicate the relationship between
415 performance indices and lead times in the testing set (2017-2018). We mainly discuss the trend of performance indices as increasing lead time. The comments about the relationship between performance indices and lead times are given more details in Section 4.3 and Section 4.4.

*24. The authors did not discuss the reasons why NSE values for lead times of 6-7-8-9-day is worse than the value of lead time of 10-day.*

**Response:** Thank you for your careful review. In original manuscript, it should be noted that NSE values for lead times of 6-7-8-9-day is worse than the value of lead time of 10-day in the validation set. We consider the possible reasons are inadequate parameter optimization and different model structure. MAE is the objective function of parameter optimization instead of NSE. According to the results of trials and your suggestion, we replace NSE with KGE and more discussion are added in Section 4 of the revised manuscript. BHV and KGE have no consistent trend as increasing lead times. Please see Section 4.4 for more details.

*25. It is not clear how top k features were selected according to the chosen threshold value. Did the authors employ several threshold values? Please give more details on this issue.*

**Response:** Thank you for your careful review and suggestion. The original manuscript totally employs three threshold values. Two of these thresholds are used to determine the model input structures with inflow and rainfall. Another threshold value is used to determine the model input structures with ERA-Interim dataset.

In the revised manuscript, MIC is employed to select inputs from 26 candidate predictors from reanalysis data; observed inflow and rainfall lags are identified by partial autocorrelation function (PACF) and cross-correlation function (CCF) of the inflow time series. The corresponding 95% confidence interval is used to identify significant correlations. Furthermore, when correlation coefficient slowly declines and cannot fall into confidence interval, a trial-and-error procedure is used to determine the optimum lag, i.e., starting from one-lag and then modifying the external inputs by successively adding one more lagged time series into inputs (Amiri 2015; Shoaib et al., 2015). Consider the subjectivity of user-defined thresholds, the three threshold values are modified by hypothesis test and the trail-and-error procedure to determine input structures of model. Please see Section 2.2 and Section 4.1 for more details.

[revised manuscript text omitted]

Table

**4.** Four commonly used activation functions for ANN-MIC.

| Name | Functional expression |
|---|---|
| Logistic | $f(x) = \dfrac{1}{1 + e^{-x}}$ |
| Tanh | $f(x) = \dfrac{e^x - e^{-x}}{e^x + e^{-x}}$ |
| Identity | $f(x) = x$ |
| Relu | $f(x) = max(0, x)$ |

1140

Table

5. Tuning parameters of ANN-MIC and SVR-MIC.

| Model | Tuning parameter / Tuning range | 1 | 2 | 3 | 4 | 5 | 6 | 7 | 8 | 9 | 10 |
|---|---|---|---|---|---|---|---|---|---|---|---|
|  ANN-MIC | Structu / Structure | 19-4-1 | 19-2-1 | 19-3-1 | 19-2-1 | 19-2-1 | 19-2-1 | 19-2-1 | 19-2-1 | 19-2-1 | 19-2-1 |
|  |  | Tanh | 17-3-1 | 17-2-1 tanh | 17-2-2 logistic | 17-2-4  logistic | 17-2  logistic | 17-2-7 logistic | 17-2-8 logistic | 17-2-9 tanh | tanh |
| SVR-MIC | | 8.9693(  6.  105 | 53.1206 1.  | 9.8590 1.  | 44.2134 1.0000 | 81.4581 11.4211 | 120.2356 1.  | 0.2587 1.  2105 | 144.4810 6.  | 0.5105 1.  | 6.2105 |
| | epsilon (0.0030  | 0.0007 00 | 0.0028 0080 | 0.0066 00 | 0.0006 0079 | 0.0015 0017 | 0.0004 000 | 0.0012 00  | 0.0027 000  | 0.0085 00 | 0.0043 |
| | gamm a (0.0265  | 0323 | 0.0201 0583 | 0.0037 0844 | 0.0120 0271 | 0.0051 0062 | 0.0012 0218 | 0.0088 0375 | 0.2150 0016 | 0.0067 0687 | 0.1815 0166 |

Note: The bold parts, (min, max, step) represent $[ min + \dfrac{max-min}{step-1} \times 0 ,\ min + \dfrac{max-min}{step-1} \times 1 ,\ \dots,\ min + \dfrac{max-min}{step-1} \times (step-1) ]$.

Table

6. Tuning parameters of GBRT and GBRT-MIC.

| Tuning parameter | Tuning range | Optimal parameters (the lead times of 1-10 days) | |
|---|---|---|---|
| | | GBRT | GBRT-MIC |
| max_leaf_nodes | [2, 4, 6, …, 40] | 8, 4, 4, 4, 4, 2, 4, 2, 2, 2 |  9, 13, 7, 15, 4, 5, 4, 4, 17 |
| min_samples_leaf | [1, 6, 11, …, 46] | 6, 31, 1, 1, 1, 31, 6, 1, 6, 1 |  2, 7, 2, 4, 2, 1, 10, 10, 8, 1 |
| max_depth | [1, 2, 3, …, 10] | 3, 2, 2, 2, 3, 1, 3, 1, 1, 1 | 4, 6, 8, 5, 9, 9, 2, 2, 7, 2 |
| min_samples_split | [2, 4, 6, …, 40] | 18, 2, 16, 16, 24, 2, 16, 2, 2, 2 | 18, 15, 12, 13, 8, 3, 19, 3, 19, 8 |
| n_estimators | [100, 200, 300, …, 4000] | 1100, 900, 1200, 700, 700, 1200, 600, 1100, 900, 900 | 3800, 2700, 1300, 900, 1000, 700, 1400, 2000, 1300, 1200 |

Table 6 7. Performance indices of the train training set.

[revised manuscript text omitted]